# Highly synergistic combinations of nanobodies that target SARS-CoV-2 and are resistant to escape

Fred D Mast[1†], Peter C Fridy[2†], Natalia E Ketaren[2†], Junjie Wang[3†], Erica Y Jacobs[3,4†], Jean Paul Olivier[1†], Tanmoy Sanyal[5], Kelly R Molloy[3], Fabian Schmidt[6], Magdalena Rutkowska[6], Yiska Weisblum[6], Lucille M Rich[7], Elizabeth R Vanderwall[7], Nicholas Dambrauskas[1], Vladimir Vigdorovich[1], Sarah Keegan[8], Jacob B Jiler[2], Milana E Stein[2], Paul Dominic B Olinares[3], Louis Herlands[9], Theodora Hatziioannou[6], D Noah Sather[1,10], Jason S Debley[7,10,11], David Fenyö[8], Andrej Sali[5], Paul D Bieniasz[6,12], John D Aitchison[1,10,13*], Brian T Chait[3*], Michael P Rout[2*]

[1]Center for Global Infectious Disease Research, Seattle Children's Research Institute, Seattle, United States; [2]Laboratory of Cellular and Structural Biology, The Rockefeller University, New York, United States; [3]Laboratory of Mass Spectrometry and Gaseous Ion Chemistry, The Rockefeller University, New York, United States; [4]Department of Chemistry, St. John's University, Queens, United States; [5]Department of Bioengineering and Therapeutic Sciences, Department of Pharmaceutical Chemistry, California Institute for Quantitative Biosciences, University of California, San Francisco, San Francisco, United States; [6]Laboratory of Retrovirology, The Rockefeller University, New York, United States; [7]Center for Immunity and Immunotherapies, Seattle Children's Research Institute, Seattle, United States; [8]Institute for Systems Genetics and Department of Biochemistry and Molecular Pharmacology, NYU Grossman School of Medicine, New York, United States; [9]AbOde Therapeutics Inc, Woods Hole, United States; [10]Department of Pediatrics, University of Washington, Seattle, United States; [11]Division of Pulmonary and Sleep Medicine, Seattle Children's Hospital, Seattle, United States; [12]Howard Hughes Medical Institute, The Rockefeller University, New York, United States; [13]Department of Biochemistry, University of Washington, Seattle, United States

*For correspondence:
John.Aitchison@seattlechildrens.org (JDA);
chait@rockefeller.edu (BTC);
rout@rockefeller.edu (MPR)

†These authors contributed equally to this work

**Abstract** The emergence of SARS-CoV-2 variants threatens current vaccines and therapeutic antibodies and urgently demands powerful new therapeutics that can resist viral escape. We therefore generated a large nanobody repertoire to saturate the distinct and highly conserved available epitope space of SARS-CoV-2 spike, including the S1 receptor binding domain, N-terminal domain, and the S2 subunit, to identify new nanobody binding sites that may reflect novel mechanisms of viral neutralization. Structural mapping and functional assays show that indeed these highly stable monovalent nanobodies potently inhibit SARS-CoV-2 infection, display numerous neutralization mechanisms, are effective against emerging variants of concern, and are resistant to mutational escape. Rational combinations of these nanobodies that bind to distinct sites within and between spike subunits exhibit extraordinary synergy and suggest multiple tailored therapeutic and prophylactic strategies.

## Editor's evaluation

The paper describes an impressive collection of hundreds of new nanobodies binding SARS-CoV-2 spike by combining in vivo antibody affinity maturation and proteomics. It provides a comprehensive characterization of a repertoire of the spike nanobodies and their combinations by complementary biophysical, structural modeling, and functional assays. It also identifies non-receptor binding domain nanobodies, includes extensive bioengineering to substantially improve potency and resistance to escaping variants, and demonstrates synergistic activities using nanobody cocktails. This work thus provides significant impacts on SARS-CoV-2 research and therapeutics.

## Introduction

SARS-CoV-2, the viral causative agent of COVID-19, is estimated to have infected some 10% of the world's population, killing a confirmed ~5 million but likely considerably more. Despite the great promise of vaccines, the pandemic is ongoing; inequities in vaccine distribution, waning immunity, the biological and behavioral diversity of the human population, the emergence of viral variants that compromise monoclonal therapies and vaccine efficacy, all challenge current and future containment (*Diamond et al., 2021*; *Lavine et al., 2021*; *Fraser et al., 2004*; *Wang et al., 2021a*; *Wang et al., 2021b*). Thus, the best we can hope for now is an uneasy truce, in which multipronged containment strategies will be required for many years to keep SARS-CoV-2, future variants, and novel coronaviruses at bay (*Phillips, 2021*; *McKenna, 2021*; *Steenhuysen and Kelland, 2021*; *Weisblum et al., 2020*).

Spike (S), the major surface envelope glycoprotein of the SARS-CoV-2 virion, is key for infection as it attaches the virion to its cognate host surface receptor, angiotensin-converting enzyme 2 (ACE2) protein, and triggers fusion between the host and viral membranes, leading to viral entry into the cytoplasm (*Zhou et al., 2020*; *Wrapp et al., 2020b*; *Walls et al., 2020*). The spike protein monomer is ~140 kDa, or ~180–200 kDa including its extensive glycosylation, and exists as a homotrimer on the viral surface. Spike is highly dynamic and is composed of two domains: S1, which contains the host receptor binding domain (RBD); and S2, which undergoes large conformational changes that enable fusion of the viral membrane with that of its host (*Li et al., 2003*; *Li, 2016*; *Letko et al., 2020*; *Watanabe et al., 2020*; *Hsieh et al., 2020*). Based on its requirement for entry, the major target of immunotherapeutics has been the RBD (*Hartenian et al., 2020*; *Wu et al., 2020*; *Baum et al., 2020*; *Finkelstein et al., 2021*; *Korber et al., 2020*; *Trigueiro-Louro et al., 2020*; *Barnes et al., 2020*).

Major immunotherapeutic strategies to date have focused on immune sera and human monoclonal antibodies; however, these therapies now face the emergence of variants, particularly RBD point mutants, which have evolved to bypass the most potent neutralizing human antibodies (*Wang et al., 2021b*; *Liu et al., 2021a*; *Weisblum et al., 2020*; *Garcia-Beltran et al., 2021*; *Starr et al., 2021*). A specific alternative class of single-chain monoclonal antibodies, commonly called nanobodies, are attractive alternatives to traditional monoclonal antibodies (*Muyldermans, 2013*). Nanobodies are the smallest single-domain antigen binding proteins identified to date, possessing several potential advantages over conventional monoclonal antibodies. Nanobodies are derived from the variable domain ($V_H$H) of variant heavy chain-only IgGs (HCAb) found in camelids (e.g., llamas, alpacas, and camels). They can bind in modes different from typical antibodies, covering more chemical space and binding with very high affinities (comparable to the very best antibodies) (*Jovčevska and Muyldermans, 2020*; *Muyldermans, 2013*). Their small size (~15 kDa) allows them to bind tightly to otherwise inaccessible epitopes that may be obscured by the glycoprotein coat, as well as minimizing issues of steric hindrance of multiple antibodies binding to adjacent epitopes as observed with larger immunoglobulin G molecules (*Corti et al., 2021*). Nanobodies are also highly soluble, very stable, lack glycans, and are readily cloned and produced in bacteria or yeast (*Muyldermans, 2013*). They have low immunogenicity (*Revets et al., 2005*; *Jovčevska and Muyldermans, 2020*; *Bannas et al., 2017*) and can be readily 'humanized' (including Fc addition), modified to alter clearance rates, derivatized, combined for synergistic activity, and multimerized to improve characteristics (*Chanier and Chames, 2019*; *Vincke et al., 2009*; *Duggan, 2018*). In the case of respiratory viruses like SARS-CoV-2, nanobodies' flexibility in drug delivery is a critical advantage. Beyond typical administration methods, a major advantage of nanobodies is their potential for direct delivery by nebulization deep into the lungs (*Wölfel et al., 2020*; *Nambulli et al., 2021*). This route can provide a high local concentration

in the airways and lungs to ensure rapid onset of therapeutic effects, while limiting the potential for unwanted systemic effects (*Erreni et al., 2020*) as exemplified by clinical trials (*Van Heeke et al., 2017*; *Zare et al., 2021*). Moreover, with respect to deployment, nanobodies are relatively inexpensive and easy to reproducibly manufacture, with long shelf-lives and greater inherent stability compared to other biologicals, including monoclonals. Taken together nanobodies have great potential for the development of superior and differentiated therapeutics that would not only serve critically ill hospitalized patients, but also are especially well suited to the developing countries, most of which lack a reliable supply chain, or to stockpiling.

To date, there are 453 nanobodies available against SARS-CoV-2 spike and those that are available primarily recognize regions of RBD with many subject to escape variation (*Niu et al., 2021*; *Raybould et al., 2021*; *Schoof et al., 2020*; *Xiang et al., 2020*; *Koenig et al., 2021*; *Huo et al., 2020a*; *Pymm et al., 2021*; *Hanke et al., 2020*; *Custódio et al., 2020*; *Esparza et al., 2020*; *Wrapp, 2020a*; *Dong et al., 2020*; *Ye et al., 2021*). To address the urgent need for strongly neutralizing and escape resistant nanobodies, we generated a large repertoire of nanobodies that exploit the available epitope and vulnerability landscape of SARS-CoV-2 spike protein. The resulting repertoire provides a plethora of synergistically potent and escape resistant therapeutics.

## Results and discussion
### Maximizing the size and diversity of anti-SARS-CoV-2 spike nanobody repertoire

We sought to isolate a large repertoire of highly diverse nanobodies against SARS-CoV-2 spike protein. Thus, we built on our existing nanobody generation pipeline (*Fridy et al., 2014a*), further optimizing each step, explicitly designing it to yield hundreds of high-quality, highly diverse nanobody candidates (*Figure 1A*). In this way, we took advantage both of the straightforward procedure of llama immunization and the powerful natural affinity maturation processes in vivo (*Thompson et al., 2016*).

To identify $V_HH$ domains that bind spike, we affinity-purified $V_HH$ domains from the immunized animals' sera against spike S1, S2, or RBD domains using independent domains in this purification step to maximize epitope accessibility. In parallel, lymphocyte RNA was taken from bone marrow aspirates and used to amplify $V_HH$ domain sequences by PCR, which were sequenced to generate an in silico library representative of all $V_HH$ sequences expressed in the individual animal. The affinity-purified $V_HH$ fragments were proteolyzed and the resulting peptides analyzed by LC-MS/MS. These data were searched against the $V_HH$ sequence library to identify and rank candidate nanobody sequences using our Llama-Magic software package (*Fridy et al., 2014a*; *Fridy et al., 2014b*) with a series of key improvements (see Materials and methods).

To maximize sequence diversity and thus the paratope space being explored, we clustered CDR sequences, revealing that many of the candidates form clusters likely to have similar antigen binding behavior. Here, partitioning of the clusters was performed by requiring that CDR3s in distinct clusters differ by a distance of more than three Damerau–Levenshtein edit operations (*Bard, 2007*) – that is, each operation being defined by insertion, deletion, or substitution of an amino acid residue, or transposition of two adjacent amino acid residues (*Figure 1B*). This partitioning was found to be effective, in that virtually no overlap was observed between those directed against S1 versus S2 (4 out of 183 clusters show overlap). The lengths of these CDR3 candidates also varied considerably, ranging from 3 to 22 amino acids in length. The use of two animals further expanded the paratope diversity in that only 4 out of 22 possible clusters from the second animal were observed to be shared with the first animal. In addition, we detected relatively little overlap between our CDR3 clusters and those observed by other groups; for example, only 1 out of 109 S1-specific clusters (Damerau–Levenshtein ≤3) were shared by *Xiang et al., 2020* and the present work, indicating that our repertoires sampled extended regions of the available paratope space (see also below).

Of the several hundred positives, 180 high-confidence candidates were selected for expression and screening. Of these, 66 were from S1 affinity purification, 63 from S2, and 51 from RBD, numbered S1-n, S2-n, and S1-RBD-n, respectively. These were then expressed with periplasmic secretion in bacteria, and crude periplasmic fractions were bound in large excess to the corresponding immobilized spike antigen to assay recombinant expression, specific binding, and degree of binding (*Figure 2—figure supplements 1 and 2*). 138 candidates were validated by this screen: 52 against

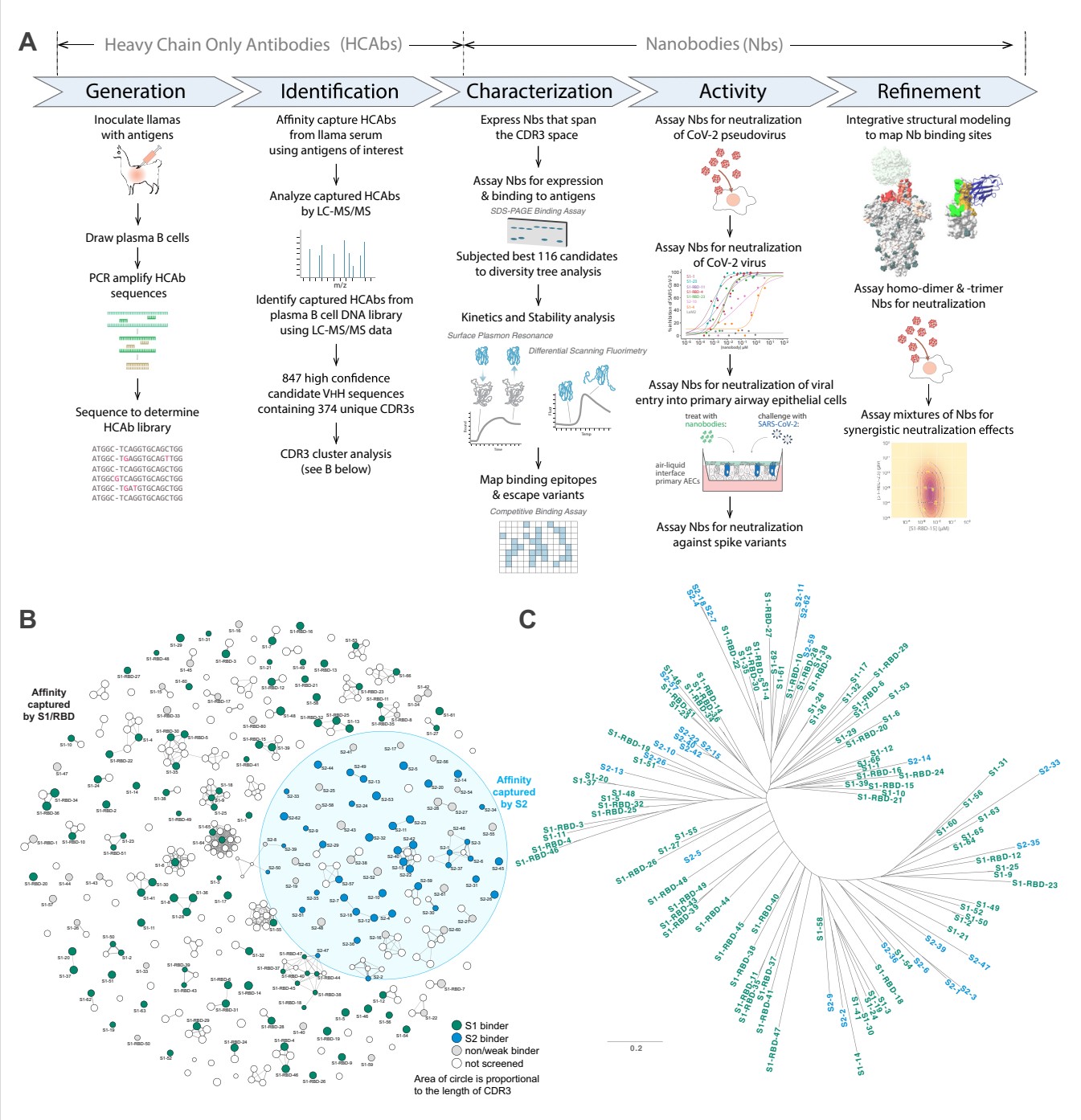

**Figure 1.** Approach. (**A**) Schematic of our strategy for generating, identifying, and characterizing large, diverse repertoires of nanobodies that bind the spike protein of SARS-CoV-2. The highest quality nanobodies were assayed for their ability to neutralize SARS-CoV-2 pseudovirus, SARS-CoV-2 virus, and viral entry into primary human airway epithelial cells. We also measured the activities of homodimers/homotrimers and mixtures. (**B**) A network visualization of 374 high-confidence CDR3 sequences identified from the mass spectrometry workflow. Nodes (CDR3 sequences) were connected by edges defined by a Damerau–Levenshtein distance of no more than 3, forming 183 isolated components. A thicker edge indicates a smaller distance value, that is, a closer relation. (**C**) Dendrogram showing sequence relationships between the 116 selected nanobodies, demonstrating that the repertoire generally retains significant diversity in both anti-S1 (green) and anti-S2 (blue) nanobodies, albeit with a few closely related members. Scale, 0.2 substitutions per residue.

The online version of this article includes the following source data for figure 1:

**Source data 1.** Nanobody sequences.

S1, 42 against S2, and 44 against RBD (*Figure 1B*). To eliminate candidates with weaker expression and binding affinity, only nanobodies in lysates with binding intensity >20% of the observed maximum across all those screened were chosen for follow-up study. This filtering identified the top 116 nanobodies that were purified for further characterization (*Tables 1* and *2*). Note that these selections were designed to provide a strict cutoff in the interests of maximizing the quality of the repertoire selected for thorough characterization, but eliminated many additional nanobodies that nevertheless specifically bind to S1 and S2. While a few of these 116 nanobodies were chosen to share similar paratopes, overall, the group retained a high sequence and paratope diversity (*Figure 1C*).

## High-affinity nanobodies across the entire spike ectodomain that are refractory to common spike escape mutants

Surface plasmon resonance (SPR) was used to detail the kinetic properties and affinities of the selected nanobodies (*Table 1* and *2*). All bound with high affinity, with >60% binding with $K_D$s < 1 nM, and two with single-digit picomolar affinities (*Figure 2*). While most S1-binding nanobodies bind RBD (71 nanobodies), 19 targeted non-RBD regions of S1 and 26 bind S2 (*Figure 2*). The lower number of non-RBD S1 and S2 nanobodies likely reflects the highly antigenic nature of the RBD and the occlusion of non-RBD S1 regions and S2 due to the glycan shield of SARS-CoV-2 spike (*Grant et al., 2020*; *Watanabe et al., 2020*). At the same time, we observed no obvious bias in nanobody affinities for these different domains. While both high on rates and low off rates contributed to these high affinities, kinetic analyses underscore the generally fast association rates (many with $k_{on} \geq 10^{+6}$) of these nanobodies (likely due to their small size and proportionally large paratope surface area), with many surpassing the $k_{on}$ of high-performing monoclonal antibodies ($k_{on} \sim 10^{+5}$) (*Tian et al., 2020*; *Figure 2*), a property that would benefit translation of these nanobodies into rapid therapeutics and diagnostics (*Carter, 2006*). For those nanobodies with apparently homologous paratopes (*Figure 1C*), we found no correlation in their kinetic properties (*Tables 1* and *2*), demonstrating that even small paratope changes can strongly alter behaviors (*Fridy et al., 2014a*).

A worrying development is the continuing emergence of viral variants, including mutations in RBD that minimize or nullify binding of many currently available monoclonal antibodies and nanobodies, which solely target RBD (*Weisblum et al., 2020*; *Wang et al., 2021b*; *Diamond et al., 2021*; *Jangra et al., 2021*; *Garcia-Beltran et al., 2021*; *Liu et al., 2021b*; *Sun et al., 2021*). Indeed, in one study, the efficacy of 14 out of the 17 most potent monoclonal antibodies tested was compromised by such common RBD mutants (*Wang et al., 2021b*). Here, based on the large size of our repertoire and its extensive binding across the available epitope space of spike, nanobodies or combinations thereof show great potential to be particularly resistant to these variants (*Sun et al., 2021*). RBD mutants represent a significant class of escape variants (*Garcia-Beltran et al., 2021*; *Greaney et al., 2021*), leading us to employ two strategies to ensure the generation of numerous nanobodies whose binding (and virus-neutralizing activities) are resistant to emerging variants. First, we isolated a large diversity of high-quality anti-RBD nanobodies to maximize the probability of identifying ones that are refractory to escape. Second, to reveal additional nanobody-neutralizing potential, we deliberately targeted non-RBD regions of spike (see below) (*Elshabrawy et al., 2012*; *Greaney et al., 2021*). To test the first strategy, we sampled RBD-binding nanobodies covering non-overlapping epitopes on RBD (*Figure 3*) and examined their binding to SARS-CoV-2 variants B.1.1.7/20I/501Y.V1/alpha (United Kingdom) and B.1.351/20H/501Y.V2/beta (South Africa) (*Wang et al., 2021a*; *Ho et al., 2021*; *Figure 2*, *Table 3*). Of the seven nanobodies tested, six of these (S1-1, S1-6, S1-RBD-9, S1-RBD-11, S1-RBD-15, and S1-RBD-35) retained their very strong binding to both variants, with only a modest reduction in affinity for S1-RBD-11 binding to variant B.1.351/20H/501Y.V2/beta (20–161 pM). For the seventh nanobody, S1-23, binding to variant B.1.1.7/20I/501Y.V1/alpha was only reduced from a $K_D$ of 17 pM to a still-respectable 230 pM, although its binding to variant B.1.351/20H/501Y.V2/beta was abolished (*Figure 2*). As expected (*VanCott et al., 1994*; *Magnus, 2013*; *Steckbeck et al., 2005*), it is the off rates that are most affected by these variants. Nevertheless, based on epitope mapping (below) and our identification of nanobodies that recognize epitopes not altered in the emerging variant strains, we expect that a high percentage of our nanobodies will remain resistant to these escape mutants; this would now include the B.1.617.2/21A/delta variant (*Campbell et al., 2021*), making our collection a powerful resource for potential prophylactics and therapeutics.

**Table 1.** S1 nanobody characterization; related to *Figures 2 and 4*.

Nanobodies against S1 were determined to bind RBD or non-RBD epitopes by their affinity for recombinant full-length S1 and/or S1 RBD protein. Binding kinetics against these two recombinant proteins were determined by surface plasmon resonance (SPR), with on rates, off rates, and $K_D$s determined by Langmuir fits to binding sensorgrams unless otherwise noted. Nanobody melting temperatures ($T_m$) were determined by differential scanning fluorimetry (DSF). Nanobodies were assayed for neutralization activity against a SARS-CoV-2 spike pseudotyped HIV-1 virus (PSV), with IC50s calculated from neutralization curves. Standard error of the mean (s.e.m.) is reported when available.

| ID | Epitope | S1 $K_{on}$ ($M^{-1} s^{-1}$) | S1 $K_{off}$ ($s^{-1}$) | S1 $K_D$ (M) | RBD $K_{on}$ ($M^{-1} s^{-1}$) | RBD $K_{off}$ ($s^{-1}$) | RBD $K_D$ (M) | $T_m$ (°C) | SARS-CoV-2 PSV IC50 (s.e.m.) (nM) |
|---|---|---|---|---|---|---|---|---|---|
| S1-1 | RBD | 4.14E+05 | 2.98E-05 | 7.20E-11 | 6.50E+05 | 5.98E-07 | 9.20E-12 | 66.5 | 6.7 (1.0) |
| S1-2 | Non-RBD | 1.59E+06 | 1.88E-03 | 1.18E-09 | No interaction detected | | | 66.5 | NA |
| S1-3 | Non-RBD | 5.08E+05 | 4.32E-04 | 8.51E-10 | No interaction detected | | | 64 | 1030 (666) |
| S1-4 | RBD | 1.25E+06 | 1.06E-04 | 8.46E-11 | 1.26E+06 | 1.26E-04 | 1.37E-10 | 66 | 41.5 (3.7) |
| S1-5 | RBD | 1.33E+05 | 1.15E-03 | 8.61E-09 | | – | | 65.25 | NA |
| S1-6 | RBD | 1.02E+06 | 5.75E-04 | 5.65E-10 | 5.92E+05 | 3.69E-04 | 6.22E-10 | 65 | 56.1 (20.7) |
| S1-7 | Non-RBD | 7.59E+05 | 9.90E-04 | 1.30E-09 | No interaction detected | | | 60.5 | NA |
| S1-9 | Non-RBD | 9.51E+05 / 1.25E+05 | 4.28E-07 / 1.57E-04 | 4.50E-13* / 1.25E-09 | | – | | 47.5 | NA |
| S1-10 | Non-RBD | 8.35E+04 | 1.82E-03 | 2.19E-08 | | – | | 64 | NA |
| S1-11 | Non-RBD | | – | | | – | | 60 | NA |
| S1-12 | RBD | 2.90E+05 | 8.92E-04 | 3.07E-09 | 2.33E+05 | 2.24E-04 | 9.63E-10 | 68 | NA |
| S1-14 | RBD | 1.08E+06 | 1.10E-03 | 1.02E-09 | 5.37E+05 | 7.99E-04 | 1.49E-09 | 57.5 | 135.8 (36.4) |
| S1-17 | Non-RBD | | – | | | – | | 65 | 1271 (888) |
| S1-19 | RBD | 1.30E+06 / 3.55E+04 | 8.86E-03 / 2.41E-04 | 6.81E-09* / 6.81E-09 | | – | | 64.5 | 139 (9.6) |
| S1-20 | RBD | 1.48E+07 | 4.37E-03 | 2.95E-10 | | – | | 69 | 51.8 (3.7) |
| S1-21 | RBD | 4.77E+06 | 1.58E-04 | 3.31E-11 | 1.22E+06 | 2.45E-04 | 2.00E-10 | 70.5 | 226 (158) |
| S1-23 | RBD | 2.82E+06 | 4.91E-05 | 1.74E-11 | 1.09E+06 | 1.07E-04 | 9.78E-11 | 64 | 5.7 (2.2) |
| S1-24 | Non-RBD | 6.49E+05 | 2.89E-04 | 4.45E-10 | No interaction detected | | | 71.5 | 724 (144) |
| S1-25 | Non-RBD | 2.15E+05 | 3.39E-05 | 1.57E-10 | No interaction detected | | | 58 | NA |
| S1-27 | RBD | 3.15E+06 | 4.52E-04 | 1.43E-10 | 2.89E+06 | 6.30E-04 | 2.18E-10 | 54 | 19.5 (4.9) |
| S1-28 | RBD | 1.38E+06 | 7.97E-04 | 5.76E-10 | 1.79E+06 | 1.03E-03 | 5.77E-10 | 66 | 66.0 (10.9) |
| S1-29 | RBD | 2.39E+05 | 1.01E-03 | 4.21E-09 | 1.73E+05 | 8.89E+04 | 5.12E-09 | 61.5 | NA |
| S1-30 | Non-RBD | 6.21E+05 | 1.48E-03 | 2.38E-09 | No interaction detected | | | 57 | 717 (388) |
| S1-31 | RBD | 2.17E+06 | 5.63E-04 | 2.59E-10 | 1.94E+06 | 9.37E-04 | 4.84E-10 | 72 | 78.7 (3.5) |

*Table 1 continued on next page*

*Table 1 continued*

| ID | Epitope | S1 $K_{on}$ (M⁻¹ s⁻¹) | S1 $K_{off}$ (s⁻¹) | S1 $K_D$ (M) | RBD $K_{on}$ (M⁻¹ s⁻¹) | RBD $K_{off}$ (s⁻¹) | RBD $K_D$ (M) | $T_m$ (°C) | SARS-CoV-2 PSV IC50 (s.e.m.) (nM) |
|---|---|---|---|---|---|---|---|---|---|
| S1-32 | Non-RBD | 2.73E+05 | 4.66E-04 | 1.71E-09 | No interaction detected | | | 79 | NA |
| S1-35 | RBD | 2.46E+06 | 2.11E-05 | 8.60E-12 | 2.70E+06 | 9.77E-05 | 3.62E-11 | 70.5 | 12.5 (0.1) |
| S1-36 | RBD | 2.28E+06 | 3.92E-04 | 1.72E-10 | 7.87E+06 | 1.72E-03 | 2.18E-10 | 63 | 48.5 (21.1) |
| S1-37 | RBD | 4.03E+06 | 2.75E-04 | 6.82E-11 | 4.14E+06 | 2.09E-04 | 5.05E-11 | 65 | 6.8 (0.7) |
| S1-38 | RBD | 5.34E+06 | 1.12E-03 | 2.10E-10 | | – | – | 64 | 66.1 (2.9) |
| S1-39 | RBD | 2.14E+06 | 8.11E-04 | 3.79E-10 | 1.68E+06 | 1.06E-03 | 6.30E-10 | 55 | 111 (4.0) |
| S1-41 | Non-RBD | 8.73E+05 | 1.38E-03 | 1.58E-09 | No interaction detected | | | 62.5 | 679 (53.4) |
| S1-46 | RBD | 1.68E+05 | 2.94E-04 | 1.75E-09 | 2.22E+05 | 1.70E-04 | 7.66E-10 | 68 | 312 (14.0) |
| S1-48 | RBD | 2.61E+06 | 6.22E-05 | 2.39E-11 | 1.66E+06 | 1.64E-04 | 9.85E-11 | 60.5 | 5.82 (0.5) |
| S1-49 | Non-RBD | 1.94E+06 | 3.63E-03 | 1.87E-09 | | – | – | 49, 74‡ | 356 (32.8) |
| S1-50 | Non-RBD | 3.33E+05 3.34E-03 | 1.39E-02 3.94E-04 | 4.40E-09† | No interaction detected | | | 66 | 13(11) |
| S1-51 | RBD | 9.28E+04 | 4.22E-04 | 4.54E-09 | 3.77E+06 | 2.01E-03 | 5.33E-10 | 56 | 555.8 (52.5) |
| S1-52 | RBD | 4.22E+05 | 3.13E-04 | 7.74E-09 | 4.53E+04 | 1.94E-04 | 4.36E-09 | 57.5 | 3343 (291) |
| S1-53 | RBD | 1.40E+06 2.36E+04 | 8.46E-03 2.19E-04 | 6.05E-09 9.27E-09 | | – | | 51.5 | 2466 (939) |
| S1-54 | RBD | 1.13E+06 | 6.58E-05 | 5.84E-11 | 2.55E+04 | 2.88E-04 | 1.13E-08 | 69 | 1699 (1554) |
| S1-55 | RBD | 3.98E+06 3.53E+04 | 5.41E-03 5.31E-06 | 1.36E-09* 1.51E-10 | 5.03E+05 1.84E+04 | 1.11E-02 1.82E-04 | 2.21E-08* 9.89E-09 | 54.5 | 5725 (3372) |
| S1-56 | RBD | 1.46E+04 4.45E-03 | 2.99E-03 7.90E-05 | 3.57E-09† | 2.21E+03 | 1.05E-04 | 4.73E-08 | 54 | NA |
| S1-58 | Non-RBD | 5.73E+05 | 1.66E-04 | 2.90E-10 | | – | | 53.5 | 940 (795) |
| S1-60 | Non-RBD | 3.30E+05 4.61E+04 | 5.24E-06 3.67E-03 | 1.59E-11* 9.58E-08 | | – | | 62 | NA |
| S1-61 | RBD | 9.87E+05 8.23E+02 | 1.81E-02 1.10E-04 | 1.84E-08* 1.34E-07 | 4.46E+04 | 1.88E-04 | 4.21E-09 | 60 | NA |
| S1-62 | RBD | 2.68E+06 | 9.51E-05 | 3.54E-11 | 3.30E+06 | 6.30E-05 | 2.08E-11 | 71.5 | 3.3 (0.8) |
| S1-63 | RBD | 1.09E+06 3.39E+04 | 1.12E-02 1.67E-04 | 1.02E-08* 4.94E-09 | 5.10E+04 | 2.23E-04 | 4.37E-09 | 65 | NA |
| S1-64 | Non-RBD | 6.97E+05 | 1.58E-04 | 2.26E-10 | | – | | 66 | 16.4 (11.7) |
| S1-65 | Non-RBD | 1.06E+06 | 1.67E-04 | 1.57E-10 | | – | | 60 | 7.3 (6.0) |
| S1-66 | Non-RBD | 4.66E+05 | 2.74E-04 | 5.87E-10 | | – | | 59 | NA |

*Table 1 continued on next page*

Table 1 continued

| ID | Epitope | S1 Kon (M⁻¹ s⁻¹) | S1 Koff (s⁻¹) | S1 KD (M) | RBD Kon (M⁻¹ s⁻¹) | RBD Koff (s⁻¹) | RBD KD (M) | Tm (°C) | SARS-CoV-2 PSV IC50 (s.e.m.) (nM) |
|---|---|---|---|---|---|---|---|---|---|
| S1-RBD-3 | RBD | 8.81E+05 / 7.36E+04 | 1.76E-02 / 1.13E-03 | 2.00E-08* / 1.53E-08 | | – | | 72 | 384 (18.7) |
| S1-RBD-4 | RBD | 2.02E+06 | 1.64E-04 | 8.09E-11 | 2.83E+06 | 8.16E-04 | 2.89E-10 | 64.5 | 17.5 (1.98) |
| S1-RBD-5 | RBD | 1.94E+06 | 1.63E-04 | 8.38E-11 | 7.21E+06 | 1.05E-03 | 1.45E-10 | 64 | 174 (3.3) |
| S1-RBD-6 | RBD | 1.55E+06 | 1.63E-04 | 1.05E-10 | 3.48E+06 | 1.13E-03 | 3.24E-10 | 66.5 | 77.2 (21.8) |
| S1-RBD-9 | RBD | | – | | 2.85E+05 | 1.23E-04 | 4.30E-10 | 69 | 235 (97.5) |
| S1-RBD-10 | RBD | | – | | | – | | – | 52.9 |
| S1-RBD-11 | RBD | 2.22E+07 | 2.94E-04 | 1.32E-11 | 2.06E+07 | 4.06E-04 | 1.97E-11 | 65 | 13.5 (5.5) |
| S1-RBD-12 | RBD | | – | | 1.10E+04 | 3.39E-05 | 3.10E-09 | 67 | NA |
| S1-RBD-14 | RBD | | – | | 1.33E+04 | 3.34E-04 | 2.51E-08 | 65 | NA |
| S1-RBD-15 | RBD | 5.37E+06 | 1.50E-04 | 2.79E-11 | 7.52E+06 | 4.95E-04 | 6.58E-11 | 59.5, 80‡ | 4.6 (1.2) |
| S1-RBD-16 | RBD | | – | | 1.68E+04 | 6.25E-05 | 3.73E-09 | 61 | 79.2 (4.2) |
| S1-RBD-18 | RBD | 2.28E+06 | 6.25E-04 | 2.74E-10 | 4.43E+06 | 1.27E-03 | 2.87E-10 | 69.5 | 67.2 (1.9) |
| S1-RBD-19 | RBD | | – | | | – | | 60 | 2124 (1451) |
| S1-RBD-20 | RBD | 2.37E+06 | 2.23E-04 | 9.43E-11 | 3.05E+06 | 7.91E-04 | 2.59E-10 | 49, 70‡ | 12.4 (1.1) |
| S1-RBD-21 | RBD | 3.50E+06 | 1.31E-03 | 3.73E-10 | 3.15E+06 | 1.71E-03 | 5.45E-10 | 48.5, 70.5‡ | 17.3 (3.1) |
| S1-RBD-22 | RBD | 9.34E+05 | 2.28E-04 | 2.44E-10 | 9.24E+05 | 4.42E-04 | 4.78E-10 | 57.5 | 100 (0.1) |
| S1-RBD-23 | RBD | | – | | 2.89E+06 | 4.61E-05 | 1.59E-11 | 61 | 7.31 (0.4) |
| S1-RBD-24 | RBD | 1.61E+06 | 1.40E-03 | 8.65E-10 | 2.12E+06 | 1.22E-03 | 5.75E-10 | 46, 67‡ | 221 (4) |
| S1-RBD-25 | RBD | | – | | 8.41E+04 | 1.16E-02 | 1.38E-07 | – | NA |
| S1-RBD-26 | RBD | 1.06E+05 | 4.58E-06 | 4.32E-11 | 2.15E+05 | 1.33E-05 | 6.19E-11 | 66 | 241 (81.4) |
| S1-RBD-27 | RBD | | – | | 6.19E+06 | 1.24E-02 | 2.00E-09 | 71 | 163 (71.4) |
| S1-RBD-28 | RBD | 1.80E+06 | 4.27E-04 | 2.38E-10 | 1.80E+06 | 4.27E-04 | 2.38E-10 | 64.5 | 32.7 (3.1) |
| S1-RBD-29 | RBD | | – | | 5.36E+05 | 1.35E-03 | 2.51E-09 | 74 | 9.53 (1.0) |
| S1-RBD-30 | RBD | 2.15E+06 | 6.66E-05 | 3.10E-11 | 3.77E+06 | 4.82E-04 | 1.28E-10 | 65 | 25.0 (3.6) |
| S1-RBD-32 | RBD | | – | | 1.05E+05 | 7.90E-03 | 7.52E-08 | 65 | NA |
| S1-RBD-34 | RBD | | – | | 5.71E+04 | 4.88E-03 | 8.54E-08 | 64 | NA |
| S1-RBD-35 | RBD | 8.01E+05 | 1.68E-04 | 2.10E-10 | 1.33E+06 | 2.50E-04 | 1.88E-10 | 57, 68‡ | 12.3 (2.4) |
| S1-RBD-36 | RBD | | – | | | – | | 71 | NA |
| S1-RBD-37 | RBD | | – | | 3.60E+05 | 8.88E-04 | 2.47E-09 | 71 | 523 (93.4) |

*Table 1 continued on next page*

*Table 1 continued*

| ID | Epitope | S1 $K_{on}$ ($M^{-1}$ $s^{-1}$) | S1 $K_{off}$ ($s^{-1}$) | S1 $K_D$ (M) | RBD $K_{on}$ ($M^{-1}$ $s^{-1}$) | RBD $K_{off}$ ($s^{-1}$) | RBD $K_D$ (M) | $T_m$ (°C) | SARS-CoV-2 PSV IC50 (s.e.m.) (nM) |
|---|---|---|---|---|---|---|---|---|---|
| S1-RBD-38 | RBD | | – | | 1.12E+06 | 9.84E-04 | 8.79E-10 | 68.5 | 84.6 (22.7) |
| S1-RBD-39 | RBD | | – | | 4.92E+05 | 7.77E-05 | 1.58E-10 | 67.5 | 90.4 (8.9) |
| S1-RBD-40 | RBD | | – | | 7.47E+05 | 2.77E-05 | 3.71E-11 | 70 | 25.6 (5.9) |
| S1-RBD-41 | RBD | | – | | 4.37E+05 | 1.39E-04 | 3.17E-10 | – | 17.0 |
| S1-RBD-43 | RBD | | – | | 6.21E+05 | 1.82E-04 | 2.92E-10 | 68 | 33.6 (1.3) |
| S1-RBD-44 | RBD | | – | | 1.91E+05 | 6.97E-05 | 3.65E-10 | 57.5 | 93.4 |
| S1-RBD-45 | RBD | | – | | 4.43E+05 | 4.14E-05 | 9.30E-11 | 53 | 22.6 |
| S1-RBD-46 | RBD | | – | | 4.69E+05 | 5.79E-04 | 1.23E-09 | 75.5 | 48.0 |
| S1-RBD-47 | RBD | | – | | 2.11E+05 | 6.45E-04 | 3.06E-09 | 53.5 | 127 (11.6) |
| S1-RBD-48 | RBD | | – | | 1.05E+05 | 2.42E-04 | 2.30E-09 | 58, 63[‡] | 68.7 (14.2) |
| S1-RBD-49 | RBD | 3.24E+05 | 3.24E-04 | 1.00E-09 | 3.15E+05 | 5.34E-04 | 1.69E-09 | 66.5 | 37.6 (5.6) |
| S1-RBD-51 | RBD | | – | | 3.77E+06 | 2.01E-03 | 5.33E-10 | 52, 61[‡] | 70.9 (29.3) |

*Curves were fit to a heterogeneous ligand model. Respective $K_{on}$, $K_{off}$, and $K_D$ values are shown for each component.

†Curves were fit to a two-state reaction model. Respective $K_{on}$, $K_{off}$, and $K_D$ values are shown for each binding state.

‡Two peaks were observed in the melting curve. $T_m$s for both are reported.

–, not determined; NA, no activity.

**Table 2.** S2 nanobody characterization; related to *Figures 2 and 4*.

Binding kinetics of S2 nanobodies were determined by surface plasmon resonance (SPR) using recombinant S2 protein, with on rates, off rates, and $K_D$s determined by Langmuir fits to binding sensorgrams unless otherwise noted. Nanobody melting temperatures ($T_m$) were determined by differential scanning fluorimetry (DSF). Nanobodies were assayed for neutralization activity against a SARS-CoV-2 or SARS-CoV-1 spike pseudotyped HIV-1 virus (PSV), with IC50s calculated from neutralization curves with standard error of the mean (s.e.m.).

| ID | $K_{on}$ (M$^{-1}$ s$^{-1}$) | $K_{off}$ (s$^{-1}$) | $K_D$ (M) | $T_m$ (°C) | SARS-CoV-2 PSV IC50 (s.e.m.) (nM) |
|---|---|---|---|---|---|
| S2-1 | 6.32E+04 | 3.79E-04 | 6.00E-09 | 65.5 | NA |
| S2-2 | 1.26E+06 | 9.35E-05 | 7.41E-11 | 64.5 | 4460 (901) |
| S2-3 | 2.62E+05 | 7.21E-05 | 2.76E-10 | 65 | 2234 (751) |
| S2-4 | 2.44E+06 | 2.62E-04 | 1.08E-10 | 56 | NA |
| S2-5 | 9.35E+05 | 2.74E-04 | 2.93E-10 | 66 | NA |
| S2-6 | – | – | – | 61.5 | NA |
| S2-7 | 1.66E+06 | 9.36E-05 | 5.62E-11 | 61 | NA |
| S2-9 | 9.29E+05 | 2.32E-04 | 2.50E-10 | 64.5 | NA |
| S2-10 | 9.31E+04 | 3.13E-04 | 3.37E-09 | 59, 64.5* | 5269 (1418) |
| S2-11 | 7.94E+06 | 1.12E-03 | 1.41E-10 | 69.5 | NA |
| S2-13 | 7.02E+05 | 1.05E-04 | 1.49E-10 | 64.5 | NA |
| S2-14 | 3.16E+06 | 1.28E-03 | 4.07E-10 | 72.5 | NA |
| S2-15 | – | – | – | 70 | NA |
| S2-18 | 1.63E+06 | 4.87E-04 | 2.99E-10 | 47, 54.5* | NA |
| S2-22 | – | – | – | – | NA |
| S2-26 | 4.45E+05 | 8.15E-05 | 1.83E-10 | 76.5 | NA |
| S2-33 | 3.68E+05 | 5.58E-05 | 2.33E-10 | 70 | NA |
| S2-35 | 2.36E+05 | 4.72E-05 | 2.00E-10 | 77 | NA |
| S2-36 | 4.39E+06 | 3.69E-04 | 8.41E-11 | 74 | NA |
| S2-39 | – | – | – | 58 | NA |
| S2-40 | 5.08E+04 | 7.16E-05 | 1.41E-09 | 69.5 | 1712 (828) |
| S2-42 | 5.12E+05 | 3.77E-06 | 7.36E-12 | 69 | NA |
| S2-47 | 3.86E+05 | 1.14E-04 | 2.96E-10 | 40, 65* | NA |
| S2-57 | 2.33E+06 | 7.18E-04 | 3.08E-10 | 67 | NA |
| S2-59 | – | – | – | 37.5, 60* | NA |
| S2-62 | 1.65E+06 | 1.12E-04 | 6.81E-11 | 64, 77.5* | 7177 (5801) |

*Two peaks were observed in the melting curve. $T_m$s for both are reported.

–, not determined; NA, no activity.

## The nanobody repertoire has favorable stability properties

A key consideration for possible biological therapeutics and diagnostics for SARS-CoV-2 is their stability under potentially denaturing conditions (*McConnell et al., 2014*). To address this, we performed differential scanning fluorimetry (DSF) experiments to determine the thermal stability ($T_m$) of each of our nanobodies. These studies revealed a thermal stability range between 50 and 80°C, similar to published results of other properly folded nanobodies and indicative of their generally high stability (*Muyldermans, 2013*). In contrast to many conventional antibodies, nanobodies are also reported

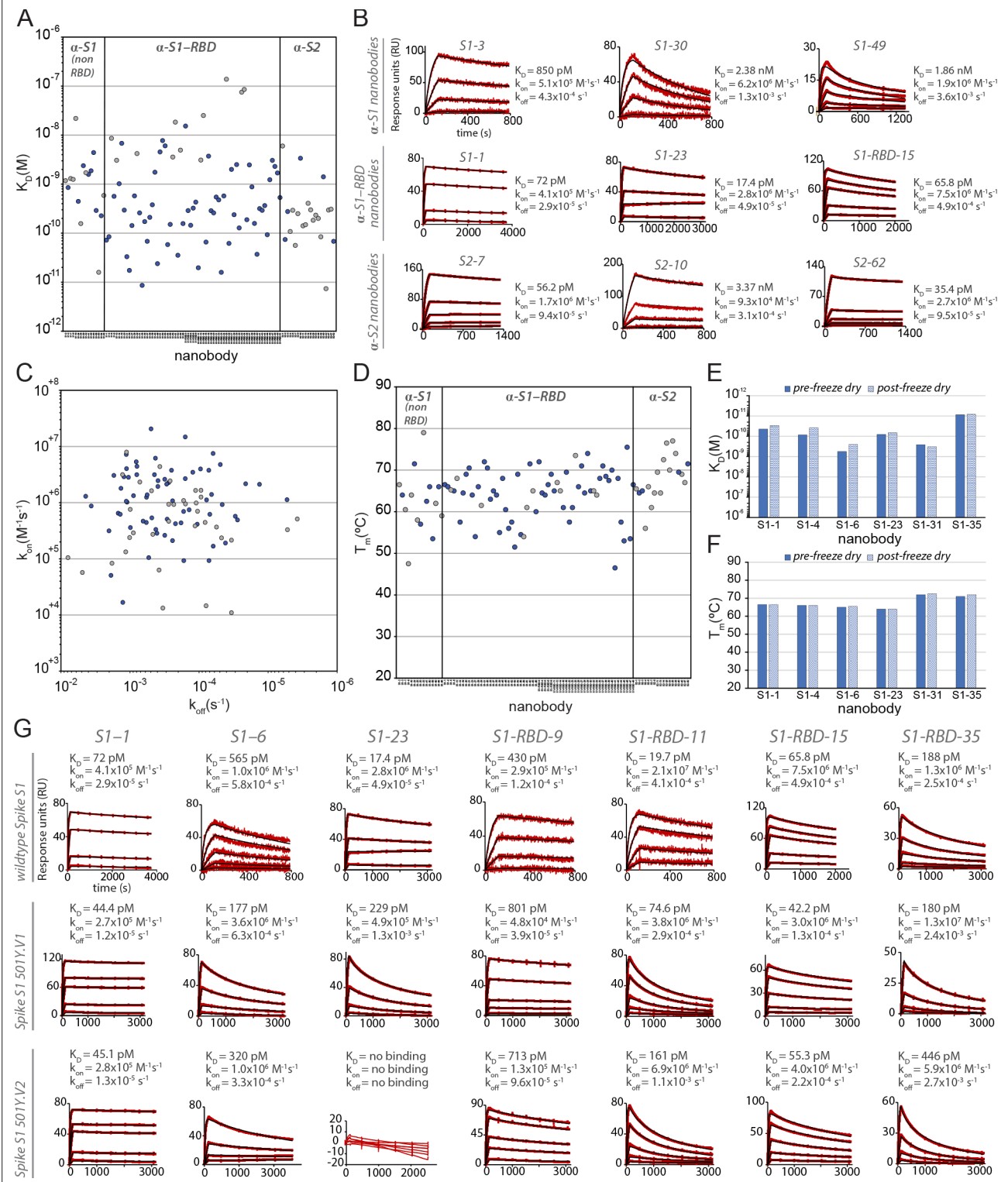

**Figure 2.** Biophysical characterization of anti-SARS-CoV-2 spike nanobodies. (**A**) Each nanobody plotted against their affinity ($K_D$) for their antigen separated into three groups based on their binding region on SARS-CoV-2 spike protein. The data points highlighted in blue correspond to nanobodies that neutralize. The majority of nanobodies have high affinity for their antigen with $K_D$s below 1 nm. 10 nanobodies are not included in this plot as they were unable to be analyzed successfully using surface plasmon resonance (SPR). (**B**) SPR sensorgrams for each of the three targets on SARS-CoV-2 spike protein of our nanobody repertoire, showing three representatives for each binding region. (**C**) The association rate of each nanobody ($k_{on}$) versus the corresponding dissociation rate ($k_{off}$). The majority of our nanobodies have fast association rates (~$10^{+5}$–$10^{+7}$ M$^{-1}$ s$^{-1}$), with many surpassing the $k_{on}$ of

*Figure 2 continued on next page*

*Figure 2 continued*

high-performing monoclonal antibodies (~$10^{+4}$–$10^{+5}$ $M^{-1}$ $s^{-1}$). (**D**) Each nanobody plotted against their $T_m$ as measured by differential scanning fluorimetry (DSF), revealing all but two nanobodies fall within a $T_m$ range between 50 and 80°C, where the bulk of our nanobodies have a $T_m \geq$ 60°C. No data could be collected for two nanobodies, and 10 nanobodies exhibited two dominant peaks in the thermal shift assay and were not included in this plot (a full summary of this data can be seen in *Tables 1–3*). The $K_D$ (**E**) and $T_m$ (**F**) of six nanobodies were assessed pre- and post-freeze-drying, revealing no significant change in affinity or $T_m$ after freeze-drying. (**G**) SPR sensorgrams comparing the kinetic and affinity analysis of seven nanobodies against wild-type spike S1 (Wuhan strain), spike 20I/S1 501Y.V1 (alpha variant), and 20H/spike S1 501Y.V2 (beta variant).

The online version of this article includes the following figure supplement(s) for figure 2:

**Figure supplement 1.** Binding of nanobody candidates to immobilized antigen.

**Figure supplement 2.** Quantified antigen binding of nanobody candidates.

to remain fully active upon reconstitution after lyophilization, particularly in buffers lacking cryoprotectants (*Schoof et al., 2020*; *Xiang et al., 2020*). A representative sample from our repertoire was thus freeze-dried without cryoprotectants, reconstituted, then analyzed via SPR and DSF to determine whether their properties were compromised due to lyophilization. The results revealed no significant effect on stability, kinetics, and affinity (*Figure 2E and F*). Taken together, these data suggest that our nanobodies, like those published in other contexts (*Xiang et al., 2020*; *Schoof et al., 2020*), are able to withstand various temperatures and storage conditions without affecting their stability and binding. These are essential requirements for downstream applications (e.g., use in a nebulizer) and ease of storage – important considerations if these are to be used for mass distribution, including in resource-poor settings (*Peeling and McNerney, 2014*).

## Nanobodies explore the major domains of the spike ectodomain

We applied a multifaceted approach to physically distinguish nanobodies that target common regions on the surface of the RBD. Using an eight-channel biolayer interferometer, we tested for pairwise competitive binding of nanobodies that bind the RBD, as well as for those that bind outside of the RBD (i.e., within the S1 non-RBD and S2 domains) (*Figure 3*). Label-free binding of antibodies to antigens measured in a 'dip-and-read' mode provides a real-time analysis of affinity and the kinetics of the competitive binding of nanobody pairs and can distinguish between those that bind to similar or overlapping epitopes versus distinct, non-overlapping epitopes (*Estep et al., 2013*). 56 anti-RBD nanobodies were screened in pairwise combinations. The response values were used to assist the discovery of nanobody groups that most likely bind non-overlapping epitopes by ensuring that the least response of pairwise nanobodies within the group was maximized. Eleven representative anti-RBD nanobodies were used as a foundation, selecting two or more representative nanobodies from each group to bin the remaining RBD nanobodies in our collection. Overlapping pairs from the foundation group and the remaining RBD binders were used to measure if a nanobody pair behaved similarly against other nanobodies measured in the dataset (*Figure 3A*), to comprehensively map nanobody competition and epitope bins (*Figure 3D*). Pearson's correlation coefficients were derived based on their binding characteristics, and the data were used to hierarchically cluster and group all RBD binders into bins. This approach revealed three large, mostly non-overlapping bins. However, each bin contained smaller, better-correlated clusters of nanobodies, reflected by the dendrogram, indicating the presence of numerous distinct sub-epitope bins present within each larger bin, that is, discrete epitopes that partially overlap with other discrete epitopes in the same bin. We calculated the gap statistic (*Tibshirani et al., 2001*), to estimate the optimal cluster number, discerning at least eight epitope bins (*Figure 3A*).

Nanobodies binding to regions outside of the RBD of S1 were binned in a similar fashion *Figure 3B, C, E, and F*. Using SPR, we binned 16 non-RBD S1-binding nanobodies and 19 S2-binding nanobodies in pairwise competition assays (*Figure 3B and C*). Pearson's correlation coefficients were used to hierarchically cluster these nanobodies, revealing as many as four S1-non-RBD bins and five S2 bins.

The binning data from pairwise combinations suggest numerous epitope bins, and thus it is reasonable to hypothesize that more than two nanobodies can bind a single domain at the same time. To test this hypothesis, we used mass photometry (MP) (*Soltermann et al., 2020*; *Wu and Piszczek, 2021*; *Young et al., 2018*), which can accurately measure multiple binding events to a single antigen. This allowed us to determine which nanobodies share epitope space on spike S1 monomer through detection of additive mass accumulation of a nanobody (or nanobodies) on spike S1 depending on

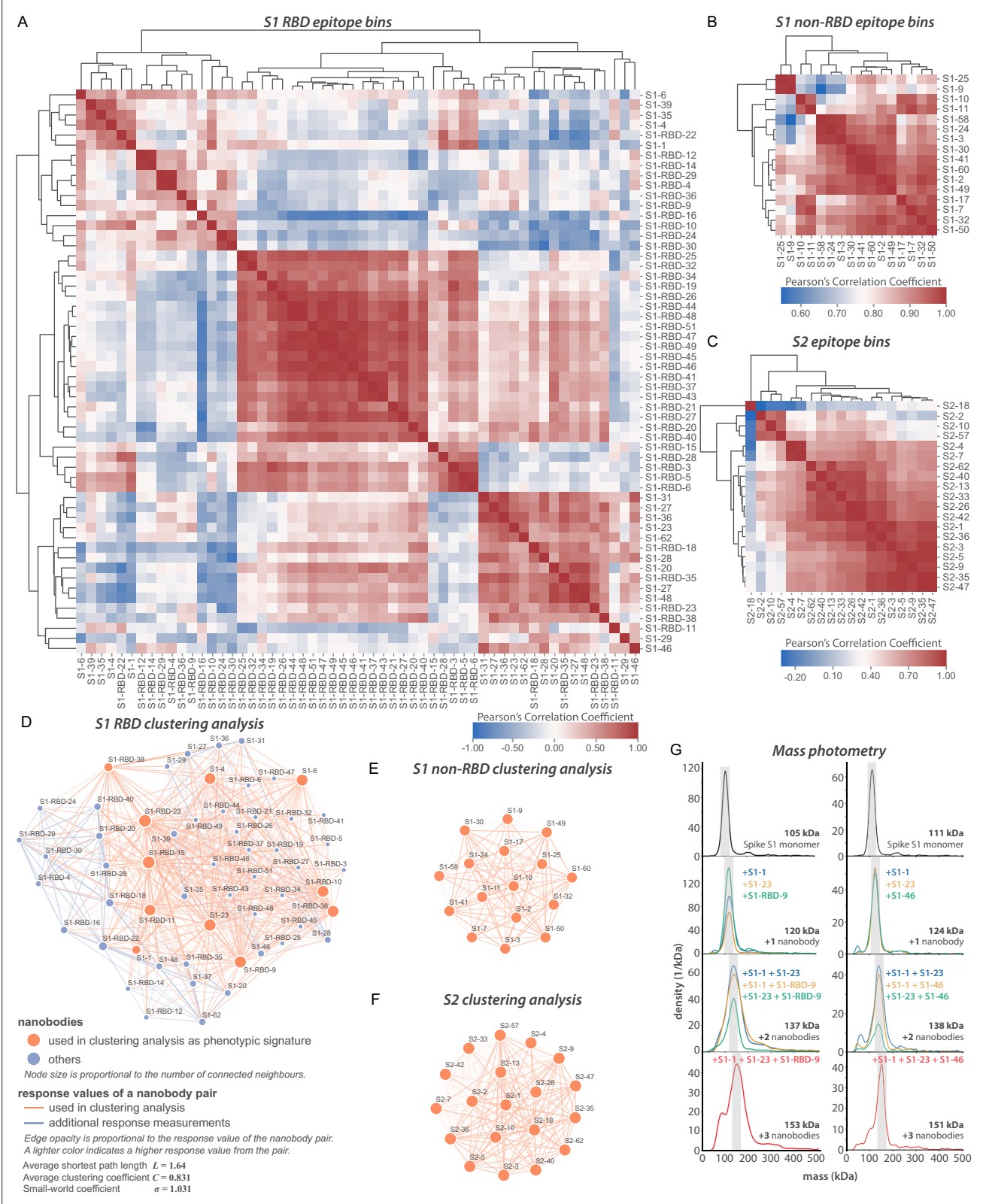

**Figure 3.** Epitope characterization of nanobodies against SARS-CoV-2 spike. (**A**) Major epitope bins are revealed by a clustered heat map of Pearson's correlation coefficients computed from the response values of nanobodies binding to the spike RBD in pairwise cross-competition assays on a biolayer interferometer. Correlated values (red) indicate that the two nanobodies respond similarly when measured against a panel of 11 RBD nanobodies that bind to distinct regions of the RBD. A strong correlation score indicates binding to a similar/overlapping region on the RBD. Anticorrelated values (blue)

*Figure 3 continued on next page*

*Figure 3 continued*

indicate that a nanobody pair responds divergently when measured against nanobodies in the representative panel and indicate binding to distinct or non-overlapping regions on the RBD. (**B**) As in (**A**), but for 16 S1 non-RBD-binding nanobodies. (**C**) As in (**A**), but for 19 S2-binding nanobodies. (**D**) A network visualization of anti-S1-RBD nanobodies. Each node is a nanobody and each edge is a response value measured by biolayer interferometry from pairwise cross-competition assays. Orange nodes represent 11 nanobodies used as a representative panel for clustering analysis in (**A**). Blue nodes represent the other nanobodies in the dataset. The average shortest distance between any nanobody pair in the dataset is 1.64. An average clustering coefficient of 0.831 suggests that the measurements are well distributed across the dataset. The small world coefficient of 1.031 indicates that the network is more connected than to be expected from random, but the average path length is what you would expect from a random network, together indicating that the relationship between nanobody pairs not actually measured can be inferred from the similar/neighboring nanobodies. (**E, F**) As in (**D**) but for S1 non-RBD and S2 nanobodies, respectively. These are complete networks with every nanobody measured against the others in the dataset. (**G**) Mass photometry (MP) analysis of spike S1 monomer incubated with different anti-spike S1 nanobodies. Two examples of an increase in mass as spike S1 monomers (black line) are incubated with 1–3 nanobodies. The accumulation in mass upon addition of each different nanobody on spike S1 monomer is due to each nanobody binding to non-overlapping space on spike S1, an observation consistent with Octet binning data. As a control, using MP, each individual nanobody was shown to bind spike S1 monomers on its own (data not shown).

The online version of this article includes the following source data and figure supplement(s) for figure 3:

**Source data 1.** Normalized response values from epitope binning of nanobodies.

**Figure supplement 1.** Mass photometry (MP) of non-RBD S1 nanobodies.

whether or not nanobodies share epitope space on spike S1. Several representative nanobodies that sample across the epitope space of our nanobody repertoire that bind the RBD were chosen for MP studies based on the epitope binning data (*Figure 3G*). These data confirmed the separation of our major epitope bins, and furthermore demonstrated that we can bind at least three different nanobodies simultaneously to the RBD, contrasting with the much larger conventional immunoglobulins, which may be too large to simultaneously bind either monomer or trimer S protein (*Corti et al., 2021*; *Stewart et al., 1997*; *Xu et al., 2021*). This is a critical consideration for the design of complementary nanobody cocktails and multimers with synergistic-neutralizing activities (see below).

## Anti-RBD nanobodies are highly effective neutralizing agents

We used a SARS-CoV-2 pseudovirus neutralization assay to screen and characterize our nanobody repertoire for antiviral activities (*Figure 4*). The lentiviral-based, single-round infection assay robustly measures the neutralization potential of a candidate nanobody and is a validated surrogate for replication competent SARS-CoV-2 (*Riepler et al., 2020*; *Schmidt et al., 2020*). Because measured IC50s are dependent on assay conditions and so cannot be readily compared across laboratories (*Cheng and Prusoff, 1973*), we included four other published nanobodies in this assay for comparison (*Xiang et al., 2020*; *Wrapp, 2020a*; *Figure 4G*). Overall, 36% of our monomeric nanobody repertoire neutralized with IC50s ≤ 100 nM, while 23% showed neutralization with IC50s < 50 nM and 17 potent neutralizers at 20 nM or lower (*Figure 4A*). Similarly, the four published nanobodies span the range of neutralization observed within our repertoire from potent (<20 nM) to relatively weak (between 1 and 10 μM). As a further comparison and validation of our IC50 values, we evaluated a subset of our nanobodies in a complementary neutralization assay (*Weisblum et al., 2020*), which revealed a strong correlation between these two assays with a Pearson's correlation coefficient of 0.98 and p-value < 0.0001 (*Figure 4—figure supplement 1*). Our most potent neutralizing nanobodies mapped to the RBD; neutralizing activity mapped to each of the major epitope bins of the RBD and were of similar efficacy to the most potent of the comparison nanobodies; importantly, nanobodies binding outside of the RBD also possess neutralizing activity (for example S1-64 and S1-65).

## Nanobody-based neutralization beyond the RBD

Notably, nanobodies mapping outside of the RBD on S1 (anti-S1, non-RBD) and to S2 also neutralized the pseudovirus in our assay, albeit with somewhat higher IC50s (*Figure 4B and C*). This is the first evidence of nanobody neutralization activity mapping outside of the RBD. As nanobodies are monomeric, the mechanism of this neutralization does not involve viral aggregation and likely reflects disruption of the virus binding or spike-driven fusion of viral and cellular membranes. Nanobodies, especially directed against relatively invariant regions of coronavirus spike proteins, may have broadly binding/neutralizing activities and are therefore important targets for optimization. Such optimization includes their use in cocktails and as oligomers.

**Table 3.** Nanobody binding activity against spike S1 variants; related to *Figure 2*.
Binding kinetics against wild-type spike S1 or two variants of concern were determined by surface plasmon resonance (SPR), with on rates, off rates, and $K_D$s determined by Langmuir fits to binding sensorgrams.

| ID | Spike S1 variant | $K_{on}$ (M$^{-1}$ s$^{-1}$) | $K_{off}$ (s$^{-1}$) | $K_D$ (M) |
|---|---|---|---|---|
| | WT (Wuhan 2019) | 4.14E+05 | 2.98E-05 | 7.20E-11 |
| | 20I/501Y.V1 | 2.71E+05 | 1.21E-05 | 4.44E-11 |
| S1-1 | 20H/501Y.V2 | 2.78E+05 | 1.25E-05 | 4.51E-11 |
| | WT (Wuhan 2019) | 1.02E+06 | 5.75E-04 | 5.65E-10 |
| | 20I/501Y.V1 | 3.55E+06 | 6.27E-04 | 1.77E-10 |
| S1-6 | 20H/501Y.V2 | 1.03E+06 | 3.29E-04 | 3.20E-10 |
| | WT (Wuhan 2019) | 2.82E+06 | 4.91E-05 | 1.74E-11 |
| | 20I/501Y.V1 | 5.96E+06 | 1.36E-03 | 2.29E-10 |
| S1-23 | 20H/501Y.V2 | NA | NA | NA |
| | WT (Wuhan 2019) | 2.85E+05 | 1.23E-04 | 4.30E-10 |
| | 20I/501Y.V1 | 4.84E+04 | 3.88E-05 | 8.01E-10 |
| S1-RBD-9 | 20H/501Y.V2 | 1.34E+05 | 9.55E-05 | 7.13E-10 |
| | WT (Wuhan 2019) | 2.22E+07 | 2.94E-04 | 1.32E-11 |
| | 20I/501Y.V1 | 3.84E+06 | 2.87E-04 | 7.46E-11 |
| S1-RBD-11 | 20H/501Y.V2 | 6.85E+06 | 1.10E-03 | 1.61E-10 |
| | WT (Wuhan 2019) | 5.37E+06 | 1.50E-04 | 2.79E-11 |
| | 20I/501Y.V1 | 2.99E+06 | 1.26E-04 | 4.22E-11 |
| S1-RBD-15 | 20H/501Y.V2 | 4.02E+06 | 2.22E-04 | 5.53E-11 |
| | WT (Wuhan 2019) | 8.01E+05 | 1.68E-04 | 2.10E-10 |
| | 20I/501Y.V1 | 1.33E+07 | 2.40E-03 | 1.80E-10 |
| S1-RBD-35 | 20H/501Y.V2 | 5.94E+06 | 2.65E-03 | 4.46E-10 |

NA, no activity.

## Oligomerization strongly enhances the affinity and neutralization activity of nanobodies

A distinct advantage of nanobodies is the facility by which oligomers can be produced to improve their affinities and avidities (*Fridy et al., 2014a*; *Schoof et al., 2020*; *Xiang et al., 2020*; *Koenig et al., 2021*). Oligomerization of most of our nanobodies significantly improved their IC50s and measured affinities (*Table 4*). For example, dimerization and trimerization of S1-RBD-35 improved neutralization activity from IC50s of ~12 nM to ~150 pM and ~70 pM, respectively. Similar results were found with S1-23, improving neutralization from ~6 nM to ~220 pM and ~90 pM, respectively (*Figure 4D*). Dimerization of the anti-S1 non-RBD nanobody S1-49 improved IC50s from ~350 nM to ~9 nM, and trimerization improved its activity an additional ~10-fold. Multimerization of some nanobodies directed against regions outside of the RBD on both S1 and S2 led to nanomolar range IC50s (*Figure 4E and F*). This includes S2-7, for which dimerization converted a nanobody that we considered to be a non-neutralizer to one having a respectable neutralizing activity (IC50 ~ 250 nM), with further potency achieved by trimerization and tetramerization, down to an IC50 of ~30 nM (*Figure 4F*). While these results show that multimerization can dramatically improve their activities, importantly this was not always the case (*Table 4*), indicating that enhancement by multimerization is not a given, but must be determined empirically.

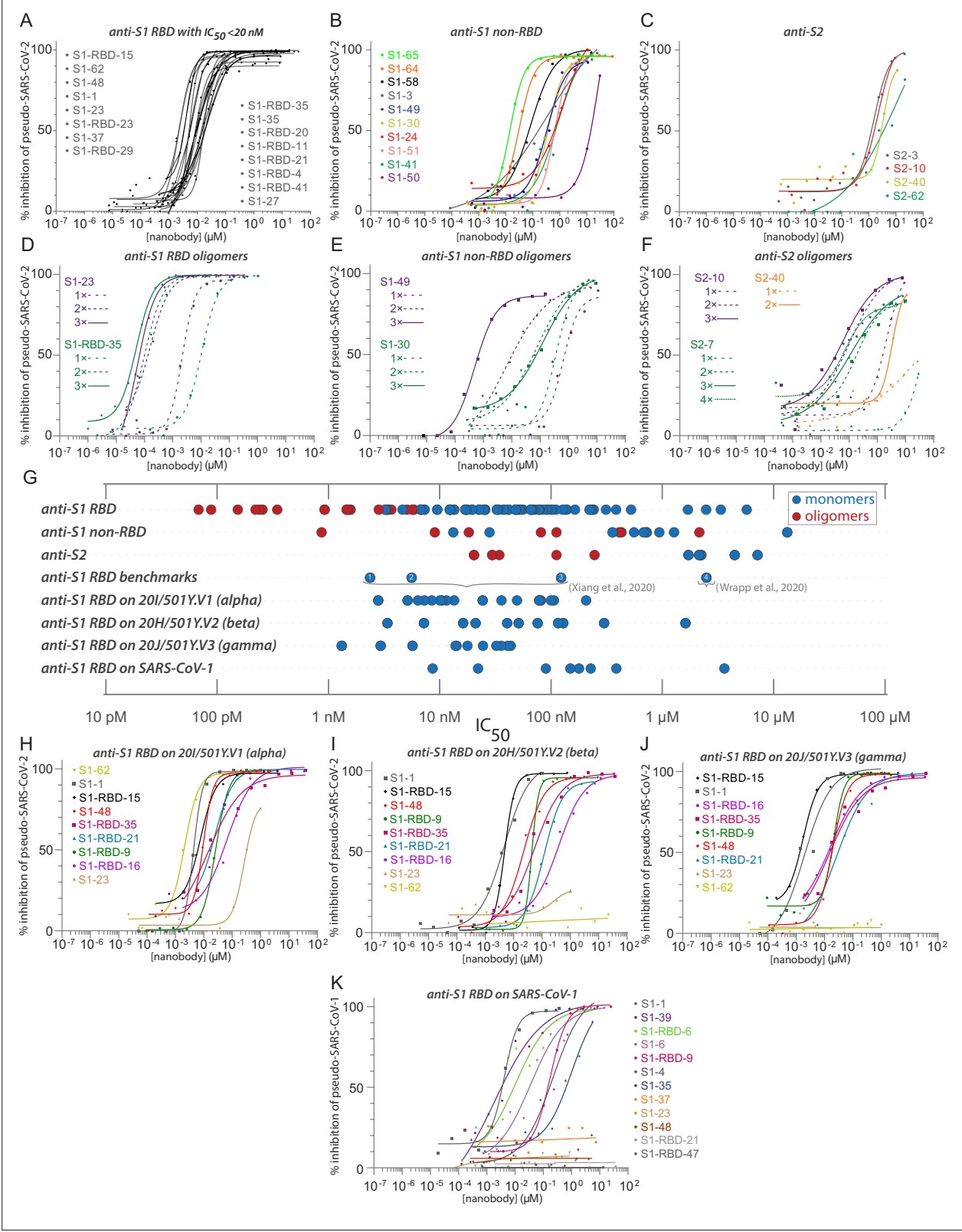

**Figure 4.** Diverse and potent nanobody-based neutralization of SARS-CoV-2. Nanobodies targeting the S1-RBD, S1 non-RBD, and S2 portions of spike effectively neutralize lentivirus pseudotyped with various SARS-CoV spikes and their variants from infecting ACE2 expressing HEK293T cells. (**A**) Of the 116 nanobodies, monomers that neutralize SARS-CoV-2 pseudovirus with IC50 values 20 nM and lower are displayed. (**B**) Representative nanobodies targeting the non-RBD portions of S1 and (**C**) the S2 domain of SARS-CoV-2 neutralize SARS-CoV-2 pseudovirus. (**D–F**) Oligomerization of RBD, S1

*Figure 4 continued on next page*

*Figure 4 continued*

non-RBD, and S2 nanobodies significantly increases neutralization potency. (**G**) Summary scatter plot of all nanobody IC50s across the major domains of SARS-CoV-2 spike and where tested, across SARS-CoV-2 variant 20H/501Y.V2 and SARS-CoV-1. Representative published nanobodies were also tested in our neutralization assays and show similar potency towards SARS-CoV-2 pseudovirus. From *Xiang et al., 2020*: (1) Nb-21 (IC50 2.4 nM); (2) Nb-34 (IC50 5.6 nM); and (3) Nb-93 (IC50 123 nM). From *Wrapp, 2020a*: (4) VHH-72 (IC50 2.5 µM). (**H–K**) Representative SARS-CoV-2 RBD targeting nanobodies cross-neutralize the 20I/501Y.V1/alpha variant with H69-, V70-, Y144- amino acid deletions and N501Y, A570D, D614G, P681H, T716I, S982A, and D1118H amino acid substitutions in spike (**H**); the 20H/501Y.V2/beta variant with L18F, D80A, K417N, E484K, and N501Y amino acid substitutions in spike; (**I**) 20J/501Y.V3/gamma variant with L18F, T20N, P26S, D138Y, R190S, K417T, E484K, N501Y, D614G, H655Y, T1027I, and V1176F amino acid substitutions in spike (**J**); and SARS-CoV-1 spike pseudotyped lentivirus (**K**). In all cases, n ≥ 2 biological replicates of each nanobody monomer/oligomer with a representative biological replicate with n = 4 technical replicates per dilution are displayed. See also *Table 1*, *Table 2*, *Table 4*, and *Table 5*.

The online version of this article includes the following figure supplement(s) for figure 4:

**Figure supplement 1.** Correlation between pseudovirus assays.

## Nanobodies neutralize SARS-CoV-2 variants and SARS-CoV-1

Certain mutations in spike, appearing in 'variants of concern' (VOC) associated with rapidly increasing case numbers in certain locales, have been demonstrated to reduce the neutralization potency of some monoclonal antibodies and polyclonal plasma, increase the frequency of serious illness, and are spreading rapidly (*Wang et al., 2021b*; *Wibmer et al., 2021*). We therefore tested a subset of our neutralizing nanobodies against pseudovirus carrying the spike protein of the alpha variant (B.1.1.7 /20I/501Y.V1) (*Figure 4H*); the beta variant (B.1.351/20H/501Y.V2) (*Tegally et al., 2021*; *Stamatatos et al., 2021*; *Figure 4I*); and the gamma variant (P.1/20J/501Y.V3) (*Figure 4J*, *Table 5*). These VOCs have mutations resulting in amino acid substitutions in spike, which impact the neutralization efficacy for some of the tested nanobodies. While all nanobodies neutralize the alpha variant, S1-23 showed an almost 14-fold drop in potency (*Figure 4A and I*). S1-23 and S1-62 failed to neutralize the beta and gamma variants, while S1-1 and S1-RBD-15 were as efficacious against all three VOCs as they were against wild-type spike (*Figure 4I*). Both S1-RBD-21 and -35 also remained effective neutralizers of spike VOC pseudotypes, albeit with reduced IC50s compared to the wild-type spike (*Figure 4I*). Remarkably, S1-RBD-9 showed *increased* neutralization activity against all three VOC, improving ~2-fold against alpha, ~6-fold against beta, and ~10-fold against gamma (*Figure 4H–J*). These results are in accord with our SPR studies, which showed that S1-1, S1-RBD-9, -15, and -35 retained very strong binding to the alpha and beta VOC, whereas binding of S1-23 was reduced against alpha and completely abolished against beta (*Figure 2G*). The B.1.617.2/21A/delta VOC has L452R and T478K as unique RBD amino acid substitutions (*Campbell et al., 2021*), which, based on our epitope binning and escape mutants (below), we would predict to impact neutralization of S1-RBD-11 and S1-RBD-35 (T478K) or S1-RBD-23 and S1-36 (L452F). However, the great majority of nanobodies in our repertoire would be predicted to show similar high neutralization efficacy against all these VOCs as compared to the wild-type virus. Overall, these data indicate that comprehensive mining of our repertoire and multimerization can lead to nanobody-based therapies that remain fully effective against common and even potentially yet-to-emerge variants of SARS-CoV-2 and with broad-spectrum coronavirus inhibition activities.

Both SARS-CoV-1 and SARS-CoV-2 share the same host receptor, ACE2, and the RBDs of the viruses share ~74% identity. As a result, some antibodies and nanobodies have been shown to be cross-neutralizing (*Liu et al., 2021b*; *Wrapp, 2020a*). We therefore tested the ability of our nanobodies to neutralize SARS-CoV-1 in the pseudovirus assay. Of the nanobodies tested in this assay, several (7 of 27 tested) of our anti-RBD monomer nanobodies also displayed excellent neutralizing activities against SARS-CoV-1 spike pseudotyped virus (*Figure 4K*). S1-1, S1-39 and S1-RBD-6 had similar IC50s against both pseudotypes, while S1-35 and S1-6 showed reduced activity against SARS-CoV-1 pseudotypes. Notably, S1-23, S1-37, and S1-48 showed no activity against SARS-CoV-1 spike pseudotypes. These three nanobodies are highly correlated with one another in our epitope binning analysis, indicating their binding to proximal epitopes on the RBD (*Figure 3A*). Beyond nanobodies that bind to the RBD, 3 of 11 nanobodies that bind to non-RBD regions of S1 and S2 also neutralized SARS-CoV-1 spike pseudotypes (*Table 5*).

**Table 4.** Characterization of oligomerized spike nanobodies; related to *Figure 4*.
Nanobody oligomers (1–4 nanobody repeats) were assayed for neutralization activity against a SARS-CoV-2 spike pseudotyped HIV-1 virus (PSV), with IC50s calculated from neutralization curves. Standard error of the mean (s.e.m.) is reported where replicates were available. Epitopes were determined by relative affinity for recombinant S1 or S1 RBD protein.

| ID | Epitope | SARS-CoV-2 PSV IC50 (s.e.m.) (nM) | SARS-CoV-1 PSV IC50 (s.e.m.) (nM) |
|---|---|---|---|
| S1-1 | RBD | 6.7 (1.0) | 8.6 (7.2) |
| S1-1$_{dimer}$ | RBD | 4.9 (0.1) | – |
| S1-1$_{trimer}$ | RBD | 5.7 (0.1) | – |
| S1-23 | RBD | 5.7 (2.2) | – |
| S1-23$_{dimer}$ | RBD | 0.22 (0.05) | NA |
| S1-23$_{trimer}$ | RBD | 0.089 (0.019) | NA |
| S1-RBD-35 | RBD | 12.3 (2.4) | NA |
| S1-RBD-35$_{dimer}$ | RBD | 0.15 (0.11) | – |
| S1-RBD-35$_{trimer}$ | RBD | 0.068 (0.043) | NA |
| S1-3 | S1 non-RBD | 1,030 (666) | 4161 |
| S1-3$_{dimer}$ | S1 non-RBD | 429 | 513 |
| S1-3$_{trimer}$ | S1 non-RBD | 411 | – |
| S1-30 | S1 non-RBD | 717 (388) | – |
| S1-30$_{dimer}$ | S1 non-RBD | 18.3 | 662 |
| S1-30$_{trimer}$ | S1 non-RBD | 77.5 (3.6) | – |
| S1-7 | S1 non-RBD | NA | – |
| S1-7$_{dimer}$ | S1 non-RBD | NA | – |
| S1-7$_{trimer}$ | S1 non-RBD | NA | – |
| S1-17 | S1 non-RBD | 1271 (888) | – |
| S1-17$_{dimer}$ | S1 non-RBD | 2144 | – |
| S1-49 | S1 non-RBD | 356 (32.8) | NA |
| S1-49$_{dimer}$ | S1 non-RBD | 9.1 (1.2) | NA |
| S1-49$_{trimer}$ | S1 non-RBD | 0.87 (0.08) | NA |
| S2-7 | S2 | NA | NA |
| S2-7$_{dimer}$ | S2 | 246 | 3516 |
| S2-7$_{trimer}$ | S2 | 112 | – |
| S2-7$_{tetramer}$ | S2 | 29.7 | – |
| S2-10 | S2 | 5269 (1418) | – |
| S2-10$_{dimer}$ | S2 | 48.0 (27.5) | – |
| S2-10$_{trimer}$ | S2 | 34.3 | – |

–, not determined; NA, no activity.

## Nanobodies effectively neutralize SARS-CoV-2 infection in human primary airway epithelium

Nanobody and antibody neutralizations have been reported to yield similar results when performed with pseudovirus versus authentic virus (*Schoof et al., 2020*; *Xiang et al., 2020*; *Schmidt et al.,*

**Table 5.** Nanobody neutralization activity against spike variants; related to *Figure 4*.
Nanobodies were assayed for neutralization activity against a pseudotyped HIV-1 virus (PSV)
expressing SARS-CoV-1 or SARS-CoV-2 wild-type or variant spike, with IC50s calculated from
neutralization curves. Standard error of the mean (s.e.m.) is reported where replicates were
available.

| ID | Epitope | SARS-CoV-2 PSV IC50 (s.e.m.) (nM) | SARS-CoV-1 PSV IC50 (s.e.m.) (nM) | SARS-CoV-2 20H/501Y. V2 PSV IC50 (s.e.m.) (nM) | SARS-CoV-2 20I/501Y. V1 PSV IC50 (s.e.m.) (nM) | SARS-CoV-2 20J/501Y. V3 PSV IC50 (s.e.m.) (nM) |
|---|---|---|---|---|---|---|
| S1-1 | RBD | 6.7 (1.0) | 8.6 (6.4) | 7.2 (1.6) | 8.5 (4.2) | 2.9 (0.2) |
| S1-3 | Non-RBD | 1030 (666) | 3598 (563) | – | – | – |
| S1-4 | RBD | 41.5 (3.7) | 179 | – | – | – |
| S1-6 | RBD | 56.1 (20.7) | 227 (205) | – | – | – |
| S1-17 | Non-RBD | 1271 (888) | NA | – | – | – |
| S1-20 | RBD | 51.8 (3.7) | NA | NA | 13.5 | NA |
| S1-23 | RBD | 5.7 (2.2) | NA | NA | 78.3 | NA |
| S1-24 | Non-RBD | 868 | NA | – | – | – |
| S1-27 | RBD | 19.5 (4.9) | NA | – | – | – |
| S1-30 | Non-RBD | 717.8 (387) | NA | – | – | – |
| S1-31 | RBD | 78.7 (3.5) | NA | – | – | – |
| S1-35 | RBD | 12.5 (0.1) | 386.8 (350) | – | – | – |
| S1-36 | RBD | 48.5 (21.1) | NA | – | – | – |
| S1-37 | RBD | 6.8 (0.7) | NA | NA | 13.5 | NA |
| S1-39 | RBD | 111 (4.0) | 22.1 (18.5) | – | – | – |
| S1-41 | Non-RBD | 679 (53.4) | NA | – | – | – |
| S1-48 | RBD | 5.82 (0.5) | NA | 20.9 (1.4) | 7.3 (1.5) | 14.2 (3.5) |
| S1-49 | Non-RBD | 356 (32.8) | NA | – | – | – |
| S1-51 | RBD | 555.8 (52.5) | NA | – | – | – |
| S1-58 | Non-RBD | 940 (795) | NA | – | – | – |
| S1-62 | RBD | 3.3 (0.8) | – | NA | 6.4 (4.5) | NA |
| S1-RBD-6 | RBD | 77.2 (21.8) | 89.7 (4.2) | 75.8 | 78.3 | 43.5 |
| S1-RBD-9 | RBD | 235 (97.5) | 149.4 (54.1) | 115.7 (18.9) | 35.9 (5.1) | 24.5 (5.8) |
| S1-RBD-11 | RBD | 13.5 (5.50) | NA | 40.4 (2.5) | 100.6 | 138.6 |
| S1-RBD-15 | RBD | 4.6 (1.2) | NA | 3.4 (1.4) | 5.2 (1.0) | 1.3 (0.03) |
| S1-RBD-16 | RBD | 79.2 (4.2) | – | 1612 (1303) | 81.2 (10.9) | 35.7 (23.1) |
| S1-RBD-20 | RBD | 12.4 (1.1) | NA | NA | 49.6 | NA |
| S1-RBD-21 | RBD | 17.3 (3.1) | NA | 128.5 (16.8) | 24.3 (0.7) | 40.7 (7.0) |
| S1-RBD-23 | RBD | 7.3 (0.4) | NA | 16.3 (1.9) | 2.8 | 5.7 |
| S1-RBD-27 | RBD | 163 (71.4) | NA | NA | 99.8 | NA |
| S1-RBD-29 | RBD | 9.5 (1.0) | NA | – | – | – |
| S1-RBD-35 | RBD | 12.3 (2.4) | NA | 51.2 (4.2) | 11.5 (1.1) | 17.7 (4.0) |
| S1-RBD-37 | RBD | 523 (93.4) | NA | NA | – | – |
| S1-RBD-40 | RBD | 25.6 (3.4) | NA | 299.8 (200) | 10.5 (0.2) | 32.2 (7.1) |

*Table 5 continued on next page*

*Table 5 continued*

| ID | Epitope | SARS-CoV-2 PSV IC50 (s.e.m.) (nM) | SARS-CoV-1 PSV IC50 (s.e.m.) (nM) | SARS-CoV-2 20H/501Y. V2 PSV IC50 (s.e.m.) (nM) | SARS-CoV-2 20I/501Y. V1 PSV IC50 (s.e.m.) (nM) | SARS-CoV-2 20J/501Y. V3 PSV IC50 (s.e.m.) (nM) |
|---|---|---|---|---|---|---|
| S1-RBD-47 | RBD | 127 (11.6) | NA | NA | 206 (123) | NA |
| S1-RBD-48 | RBD | 68.7 (14.2) | NA | NA | 106.6 (65.6) | NA |
| S2-2 | S2 | 4460 (901) | NA | – | – | – |
| S2-3 | S2 | 2234 (751) | 6277 | – | – | – |
| S2-40 | S2 | 1712 (828) | NA | – | – | – |
| S2-62 | S2 | 7177 (5801) | 1954 (364) | – | – | – |

–, not determined; NA, no activity.

*2020*). However, discrepancies have also been reported, particularly for antibodies targeting regions outside the RBD (*Chi et al., 2020*; *Huo et al., 2020b*). We therefore selected a panel of exemplar (monomeric) nanobodies, which target the RBD and domains outside of the RBD, to test for neutralization with authentic SARS-CoV-2. All nanobodies tested that neutralized pseudovirus also showed potent neutralization by plaque and focus reduction assays and correlated well with our pseudovirus assays (*Figure 5A*, *Table 6*).

To mimic human infection, we exploited human air-liquid interface (ALI) cultures of primary airway epithelium as an ex vivo model system of viral infection (*Barrow et al., 2021*). This system mimics the lung environment as it contains pseudostratified, ciliated, and mucous-secreting cells that express ACE2 (*Murphy et al., 2020*) and has several advantages over animal models including representing the relevant physiological site of initial SARS-CoV-2 infection in humans (and associated innate responses), while enabling experimental control over infection, nanobody delivery, and quantification of viral RNA at the site of infection. We thus tested a subset of our nanobodies for their ability to block SARS-CoV-2 infection and spread in this model (*Figure 5B*). We treated the air-exposed apical surface of the culture with serial dilutions of S1-1 and S1-23 and then challenged them with SARS-CoV-2 at an MOI of 0.5. ALI cultures were then treated with nanobodies at 24 hr intervals for an additional 3 days before harvesting the cells, extracting RNA, and measuring SARS-CoV-2 levels by qPCR (*Figure 5B*). S1-1 potently neutralized SARS-CoV-2 at each concentration tested while S1-23 inhibited SARS-CoV-2 in a dose-dependent manner (*Figure 5C*). The efficacy of the S1-23 nanobody was strongly enhanced when provided to cells as a trimer, potently inhibiting viral replication (*Figure 5C*). As an additional comparator and as a control, we determined the inhibition of replication upon addition of recombinant competitor, ACE2. Nanobodies inhibited at lower doses than recombinant ACE2, reflective of our measured low $K_D$ of nanobody interactions with spike (<1 nM) compared to a reported $K_D$ of 14.7 nM or greater for ACE2 with spike (*Huang and Chai, 2020*; *Shang et al., 2020*; *Chan et al., 2020*; *Rogers et al., 2020*; *Liu et al., 2020b*; *Cao et al., 2020*). These data highlight the potential for nanobodies to function as single-agent therapies against COVID-19, with efficacies comparable to monoclonal immunoglobulins.

## Escape-resistant nanobody cocktails

With the emerging variants of concern, our goal is to develop nanobody multimers and cocktails that are maximally refractory to escape by such variants. To do so, we used a previously employed method that drives the selection of antibody-resistant populations of rVSV/SARS-CoV-2 chimeric virus harboring variants of spike and measured the ability of the chimeric virus to escape nanobody-mediated neutralization (*Weisblum et al., 2020*). This approach simultaneously maps the escape potential of spike and the epitopes responsible for neutralization by nanobody binding (*Figure 6—figure supplement 1*), with the goal of discovering spike variants that resist the neutralizing activity of individual nanobodies. Based on this information, we could then predict pairs of nanobodies whose escape mutants do not map to the same region of spike, the combination of which would thus likely prevent escape. Specifically, we prepared large and diversified populations ($10^6$ infectious units) of a recombinant

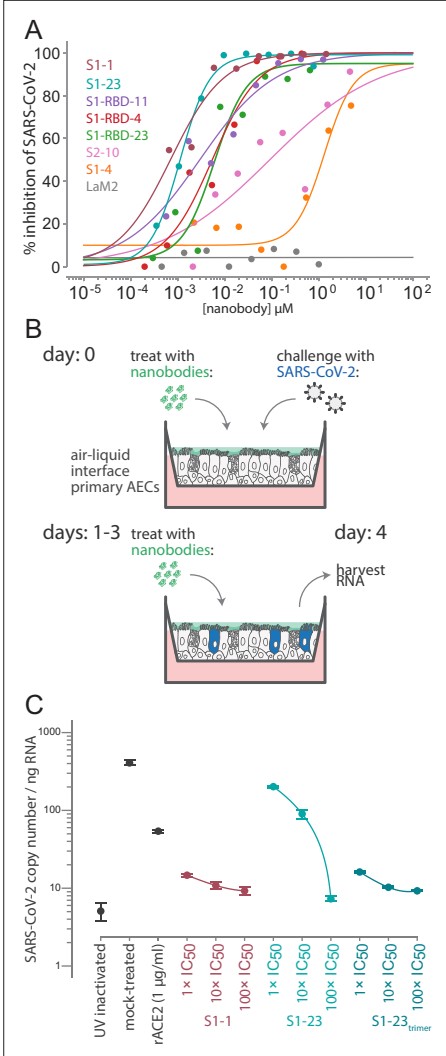

**Figure 5.** Authentic SARS-CoV-2 neutralization by anti-spike nanobodies. (**A**) Nanobodies neutralize the authentic SARS-CoV-2 virus with similar kinetics as the SARS-CoV-2 pseudovirus. Neutralization curves are plotted from the results of a focus-forming reduction neutralization assay with the indicated nanobodies. Serial dilutions of each nanobody were incubated with SARS-CoV-2 (MOI 0.5) for 60 min and then overlaid on a monolayer of Vero E6 cells and incubated for 24 hr. LaM2, an anti-mCherry nanobody (**Fridy et al., 2014a**), was used as a non-neutralizing control. After 24 hr, cells were collected and stained with anti-spike antibodies and the ratio of infected to uninfected cells was quantified by flow cytometry. (**B**) A schematic of an air-liquid interface (ALI) culture of primary human airway epithelial cells (AECs) as a model for SARS-CoV-2 infection. Cells were incubated with nanobodies and then challenged with SARS-CoV-2 (MOI 0.5). After daily treatment with nanobodies for three more days, the cultures are harvested to isolate RNA and quantify the extent of infection. (**C**) Potent neutralization of authentic SARS-CoV-2 in AECs. The AECs were infected

*Figure 5 continued on next page*

*Figure 5 continued*

with the indicated concentrations of anti-SARS-CoV-2 spike nanobodies. The infected cultures were maintained for 5 days with a daily 1 hr incubation of nanobodies before being harvested for RNA isolation and determination of the SARS-CoV-2 copy number by qPCR. SARS-CoV-2 copy number was normalized to total RNA measured by spectrophotometry. Mock-treated samples exposed to infectious and UV-inactivated SARS-CoV-2 virions served as positive and negative controls. Recombinant soluble angiotensin converting enzyme 2 (rACE2) was used as a positive treatment control. The indicated nanobodies were used at 1, 10, and 100× their IC50 values determined in pseudovirus neutralization assays (**Table 1** and **Table 4**).

The online version of this article includes the following source data for figure 5:

**Source data 1.** Neutralization data from authentic SARS-CoV-2 experiments.

VSV derivative (rVSV/SARS-CoV-2/GFP wt$_{2E1}$) that encodes SARS-CoV-2 spike protein in place of VSV-G, and recapitulates the neutralization properties of authentic SARS-CoV-2 (**Schmidt et al., 2020**). The rVSV/SARS-CoV-2/GFP wt$_{2E1}$ populations were incubated with each of the nanobodies at a nanobody concentration that was 10–100× the IC50, to neutralize susceptible variants. Then the nanobody-virus mixture was plated on 293T/ACE2cl.22 cells, and neutralization-resistant variants thereby selected and amplified by virus replication. Individual viral escape variants were then isolated by limiting dilution, amplified, and their sensitivity to neutralization by the selecting nanobody compared to the sensitivity of the starting

**Table 6.** Nanobody neutralization activity against SARS-CoV-2; related to **Figure 5**. Nanobodies were assayed for neutralization activity against authentic SARS-CoV-2, with IC50s calculated from neutralization curves.

| ID | Epitope | SARS-CoV-2 IC50 (nM) |
|---|---|---|
| S1-1 | RBD | 1.1 |
| S1-4 | RBD | 1310 |
| S1-23 | RBD | 0.7 |
| S1-RBD-4 | RBD | 5.4 |
| S1-RBD-11 | RBD | 3.0 |
| S1-RBD-23 | RBD | 6.1 |
| S2-10 | Non-RBD | 91.2 |
| LaM2 | Non-target ctrl | NA |

NA, no activity.

rVSV/SARS-CoV-2/GFP wt$_{2E1}$ virus. We thus identified 32 unique rVSV-SARs-CoV-2/GFP mutants that exhibited resistance to one or more of 22 representative neutralizing nanobodies against diverse spike epitopes (*Table 7*). For some of the non-RBD epitope nanobodies, we used dimeric or trimeric forms of the nanobodies to further enhance their activity, but in each case the selected viral isolates exhibited resistance to monomeric, dimeric, or trimeric forms. While some of the mutations that arose in the selection experiments were likely passenger mutations (*Table 7*), a number of mutations clustered on the spike surface close to each other on RBD (*Figure 6—figure supplement 1*; *Weisblum et al., 2020*; *Muecksch et al., 2021*; *Wang et al., 2021b*). Some of the most potently neutralizing nanobodies selected resistant mutations at the same positions (e.g., E484K) as those selected by potent neutralizing antibodies that have been cloned from SARS-CoV-2 convalescents and vaccine recipients, confirming that the ACE2 binding site is a point of particular vulnerability for potent neutralization. Additionally, however, other nanobodies selected mutations that have not previously been encountered in human antibody selection experiments (*Table 7*).

Nanobody cocktails are expected to be resistant to escape as they recognize multiple epitopes (*Baum et al., 2020*; *De Gasparo et al., 2021*; *Weisblum et al., 2020*). As proof of principle, we generated sets of two-nanobody cocktails by combining specific nanobodies that selected spatially distinct resistance mutations on the RBD (*Figure 3A*). When rVSV/SARS-CoV-2/GFP was passaged in the presence of the single nanobodies, resistant mutants were rapidly selected, as before. Indeed, the yield of infectious virus obtained after two passages in the presence of the single nanobody was nearly indistinguishable from that when rVSV/SARS-CoV-2/GFP was passaged in the absence of nanobodies. In contrast, when nanobodies were combined in cocktails containing two nanobodies, at the same total concentration as was used for the individual nanobodies, in eight out of nine cases, no infectious rVSV/SARS-CoV-2/GFP was recovered after two passages (*Figure 6—figure supplement 1E*). In the ninth case in which S1-48 and RBD-15 were combined and virus was still recovered, sequence analysis revealed that this virus contained two amino acid substitutions, F490V and Y508H, in the RBD. These substitutions were similar or identical to the individual substitutions found in the selection experiments with the single S1-48 and S1-RBD-15 nanobodies, which gave escape variants carrying the substitutions F490S and Y508H, respectively (*Table 7*). These results show that simply combining two nanobodies imposed the requirement for a minimum of two amino acid substitutions to confer resistance to the nanobody cocktail, greatly elevating the genetic barrier for escape. Such mixtures or derived multimers may represent powerful escape-resistant therapeutics, and even more escape resistance should be possible using three or more carefully chosen nanobodies in cocktails or multimers.

## Integrative structural modeling reveals that the nanobody repertoire explores the available spike epitopes

We have taken an integrative modeling approach to generate structural maps of representative nanobody-spike complexes from our repertoire, allowing us to infer likely mechanisms by which our different nanobodies and combinations inhibit the virus. We used the Integrative Modeling Platform (IMP) (*Webb et al., 2018*) to generate structures using multiple atomic resolution structures available for both spike and the invariant framework of nanobodies as our starting point. Spatial restraints for these calculations were based on our escape mutant data (*Table 7*) because for each nanobody its escape mutants cluster around a highly restricted area of its binding epitope on spike (*Garrett et al., 2021*); additional residue-specific distance restraints were generated by cross-linking with MS readout (XL-MS) using the amine-specific bifunctional cross-linkers DSS and BS3 (*Shi et al., 2015*; *Table 8*). We also incorporated our epitope binning and MP findings (*Figure 3*, *Figure 3—figure supplement 1*) to provide excluded volume validation data (*Webb and Sali, 2021*; *Webb et al., 2018*). We benchmarked this modeling approach using a published nanobody with escape mutant data and a solved cryo-EM structure (*Figure 6—figure supplement 2*; *Sun et al., 2021*). These models provide sufficient resolution to map the size and position of the epitopes bound by each nanobody; however, future higher-resolution studies using cryo-EM or crystallization are warranted for the highest priority nanobodies (*Schoof et al., 2020*; *Pymm et al., 2021*; *Xiang et al., 2020*; *Wrapp, 2020a*).

In sum, we solved integrative structures for 21 different neutralizing nanobodies that, based on our epitope binning data, appeared to collectively explore much of the spike surface, with 18 recognizing RBD, 1 recognizing the NTD of S1, and 2 recognizing S2 (*Figure 6*). It should be noted that these represent only a small fraction of our total repertoire and so total coverage is greater than what

**Table 7.** Nanobody neutralization of rVSV/SARS-CoV-2 and selected resistant mutants; related to *Figure 6*.

Neutralization assays were carried out using rVSV/SARS-CoV-2 and 293T/ACE2cl.22 target cells treated with the denoted nanobodies. Pseudovirus with either wild-type or variant spike (with escape mutants selected using the corresponding nanobody) was used. Escape mutants and IC50s are listed. Amino acid substitutions contributing to loss of neutralization activity are indicated in bold.

| Nanobody | Epitope | rVSV/SARS-CoV-2 variant | IC50 (nM)± s.e.m. |
|---|---|---|---|
| | | wt | 2.63 ± 0.23 |
| | | **Y369N** | 122 ± 3.0 |
| S1-1 | RBD | **G404E** | 40.8 ± 1.01 |
| | | wt | 13.0 ± 3.47 |
| | | D574N[*], Q792H, Q992H | 587 ± 31.1 |
| S1-6 | RBD | **S371P**, H66R, N969T | 202 ± 29.9 |
| | | wt | 0.58 ± 0.02 |
| | | **F490S, E484K, Q493K** | > 1000 |
| | | **Q493R**, G252R | > 1000 |
| S1-23 | RBD | **E484K**[†] | > 1000 |
| | | wt | 3.69 ± 0.14 |
| | | W64R,**L452F** | 262 ± 10.1 |
| | | W64R,**F490L**,I931G | 870 ± 202 |
| S1-36 | RBD | W64R, **F490S** | >1000 |
| | | wt | 1.83 ± 0.59 |
| S1-37 | RBD | W64R, **F490S** | >1000 |
| | | wt | 1.75 ± 0.43 |
| | | **Y449H, F490S**, Q787R | >1000 |
| S1-48 | RBD | **S494P** | >1000 |
| | | wt | 0.65 ± 0.16 |
| S1-62 | RBD | **E484K** | >1000 |
| | | wt | 60.0 |
| | | W64R, **Y170H**, V705M | >1000 |
| S1-3$_{trimer}$ | S1 non-RBD | W64R, **Y170H**, Q787H | >1000 |
| | | wt | 150 |
| S1-30$_{trimer}$ | S1 non-RBD | **T315I** | 2400 |
| | | wt | 146 ± 53.8 |
| S1-49 | S1 non-RBD | **S172G** | >1000 |
| | | wt | 3.38 ± 2.44 |
| S1-49$_{dimer}$ | S1 non-RBD | **S172G** | >1000 |
| | | wt | 0.47 ± 0.00 |
| S1-49$_{trimer}$ | S1 non-RBD | **S172G** | >1000 |
| S2-10 | S2 | wt | 6649 ± 2,545 |
| | | W64R, **S982R** | >100,000 |

*Table 7 continued on next page*

*Table 7 continued*

| Nanobody | Epitope | rVSV/SARS-CoV-2 variant | IC50 (nM)± s.e.m. |
|---|---|---|---|
| | | wt | 1015 ± 236 |
| S2-10$_{dimer}$ | S2 | W64R, **S982R** | >40,000 |
| | | wt | 30.2 ± 7.43 |
| | | T259K, **K378Q** | >1000 |
| | | W64R, **K378Q** | >1000 |
| S1-RBD-9 | RBD | **K378Q** | >1000 |
| | | wt | 1.44 ± 0.53 |
| | | **F486S** | >1000 |
| | | **T478R** | >1000 |
| S1-RBD-11 | RBD | **T478I** | >1000 |
| | | wt | 1.21 ± 0.06 |
| S1-RBD-15 | RBD | **Y508H** | 549 ± 36.9 |
| | | wt | 268 ± 162 |
| S1-RBD-16 | RBD | **N354S** | >1000 |
| | | wt | 9.61 ± 1.90 |
| | | **F486L** | >1000 |
| S1-RBD-21 | RBD | **Y489H** | >1000 |
| | | wt | 31.5 ± 11.8 |
| S1-RBD-22 | RBD | **K378Q** | >1000 |
| | | wt | 14.8 ± 3.55 |
| | | **L452R** | >1000 |
| S1-RBD-23 | RBD | H245R, **S349P**, H1083Y | >1000 |
| | | wt | 58.0 ± 0.00 |
| | | **P384Q** | >1000 |
| S1-RBD-24 | RBD | **K378Q**[†] | >1000 |
| | | wt | 18.0 ± 1.80 |
| | | **E484G** | >1000 |
| S1-RBD-29 | RBD | **E484K** | >1000 |
| | | wt | 1.80 ± 0.15 |
| | | **T478I** | >1000 |
| | | **F486L**[†] | 306 ± 17.2 |
| S1-RBD-35 | RBD | **Y489H**[†] | 57.4 ± 4.5 |
| | | wt | 38.9 ± 11.7 |
| S1-RBD-40 | RBD | W64R, **F490S** | > 500 |

[*]Residue 574 is outside the structurally covered region of the RBD (residues 333–526) and, therefore, was not used in the Integrative Modeling Platform modeling.
[†]Variant was separately identified by selection against a different nanobody.

**Table 8.** Cross-linked residues used in integrative modeling; related to *Figure 6*. The indicated nanobodies were bound to RBD, NTD, or the spike ectodomain and cross-linked with disuccinimidyl suberate (DSS). Cross-linked complexes were excised from SDS-PAGE gels, reduced, alkylated, and digested with either trypsin or chymotrypsin. Peptides were extracted and analyzed by mass spectrometry. Cross-linked residues (listed) were identified using pLink, and spectra were manually validated to eliminate false positives.

| Nanobody | Nanobody residue # | Spike construct | 7KRQ residue number |
|---|---|---|---|
| S1-49 | 49 | NTD | 187 |
| S1-49 | 70 | NTD | 187 |
| S1-49 | 70 | NTD | 41 |
| S1-49 | 70 | NTD | 182 |
| S1-49 | 81 | NTD | 97 |
| S1-49 | 81 | NTD | 187 |
| S1-49 | 81 | NTD | 182 |
| S1-49 | 81 | NTD | 41 |
| S1-1 | 47 | RBD | 458 |
| S1-1 | 47 | RBD | 462 |
| S1-1 | 47 | RBD | 417 |
| S1-1 | 69 | RBD | 458 |
| S1-1 | 69 | RBD | 444 |
| S1-1 | 69 | RBD | 417 |
| S1-1 | 80 | RBD | 417 |
| S1-1 | 80 | RBD | 386 |
| S1-1 | 80 | RBD | 444 |
| S1-1 | 80 | RBD | 458 |
| S1-1 | 80 | RBD | 417 |
| S1-1 | 91 | RBD | 444 |
| S1-1 | 105 | RBD | 386 |
| S1-23 | 47 | RBD | 458 |
| S1-23 | 47 | RBD | 444 |
| S1-23 | 47 | RBD | 462 |
| S1-23 | 69 | RBD | 444 |
| S1-23 | 69 | RBD | 462 |
| S1-23 | 91 | RBD | 417 |
| S1-23 | 91 | RBD | 444 |
| S1-23 | 91 | RBD | 417 |

*Table 8 continued on next page*

*Table 8 continued*

| Nanobody | Nanobody residue # | Spike construct | 7KRQ residue number |
|---|---|---|---|
| S1-46 | 47 | RBD | 458 |
| S1-46 | 69 | RBD | 386 |
| S1-46 | 69 | RBD | 458 |
| S1-46 | 80 | RBD | 458 |
| RBD-9 | 47 | RBD | 444 |
| RBD-9 | 69 | RBD | 386 |
| RBD-9 | 69 | RBD | 444 |
| RBD-9 | 80 | RBD | 458 |
| RBD-9 | 114 | RBD | 444 |
| RBD-35 | 47 | RBD | 458 |
| RBD-35 | 47 | RBD | 462 |
| RBD-35 | 62 | RBD | 417 |
| RBD-35 | 62 | RBD | 458 |
| RBD-35 | 68 | RBD | 458 |
| RBD-35 | 68 | RBD | 444 |
| S2-10 | 69 | Spike ectodomain | 964 |
| S2-10 | 80 | Spike ectodomain | 835 |
| S2-10 | 115 | Spike ectodomain | 854 |
| S2-10 | 115 | Spike ectodomain | 964 |
| S2-40 | 69 | Spike ectodomain | 814 |
| S2-40 | 69 | Spike ectodomain | 786 |
| S2-40 | 69 | Spike ectodomain | 790 |

is represented by these maps. Based on overlapping footprints, these 21 nanobodies are classified into 10 groups. *Figure 6* summarizes the position of binding and the relative neutralization activity of each of the 21 mapped nanobodies in a heatmap format. As expected, neutralizing nanobodies bind at sites that are complementary to sites of glycosylation, which entropically shield larger zones than represented (*Casalino et al., 2020*), and are instead concentrated at the largely glycan-free RBD. Indeed, among our entire repertoire, epitope binning shows that neutralization activity, corresponding escape mutants, and the mapped epitopes are heavily concentrated on the RBD (*Figures 3, 4 and 6*); ~80% of our anti-RBD

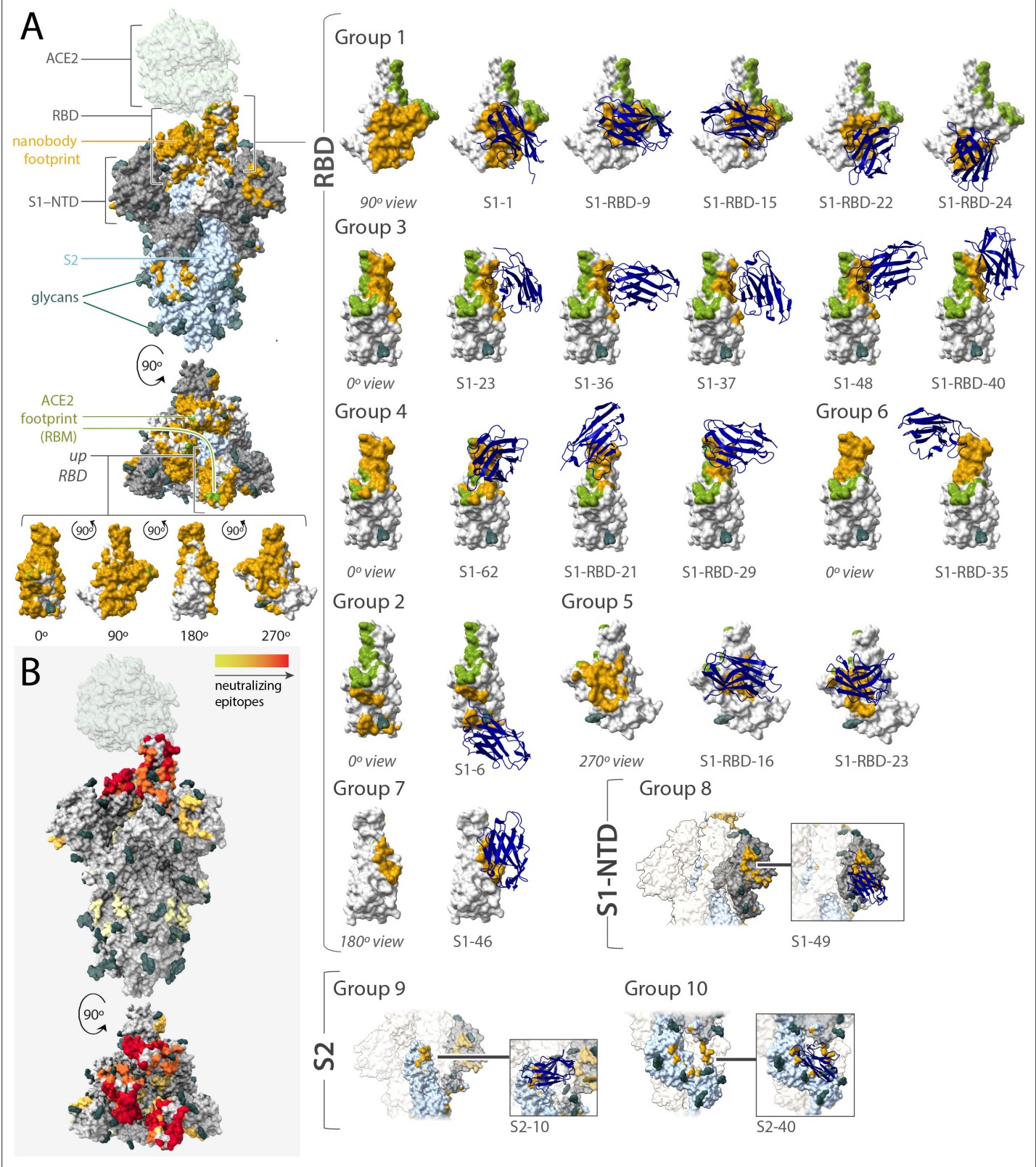

**Figure 6.** Epitope coverage of the 21 structural models of anti-spike SARS-CoV-2 nanobodies and neutralization potential of each epitope. (**A**) The structure of SARS-CoV-2 full spike (PDB ID: 6VYB) solved via cryo-EM with one RBD in the up position overlaid with the crystal structure of RBD bound to ACE2 (PDB ID: 6M0J). Key elements of spike are colored as follows: RBD (white), S1-NTD (gray), and S2 (light blue). All 21 modeled nanobody footprints are colored gold on full spike with the ACE2 footprint (RBM) colored green. Full coverage of the 18 anti-RBD modeled nanobody footprints on RBD is

*Figure 6 continued on next page*

*Figure 6 continued*

seen in four different orientations. All 21 nanobodies are categorized into 10 groups based on their footprint on spike where groups 1–7 are anti-RBD nanobodies; group 8 contains an anti-S1-NTD nanobody and groups 9 and 10 contain anti-S2 nanobodies. (**B**) Heatmap of neutralizing epitopes on the structure of SARS-CoV-2 full spike (PDB ID: 6VYB). Epitopes are colored from pale yellow (epitopes with weak neutralization against SARS-CoV-2) to dark red (strong neutralization against SARS-CoV-2).

The online version of this article includes the following source data and figure supplement(s) for figure 6:

**Source data 1.** PDB files of structural models of anti-spike SARS-CoV-2 nanobodies.

**Figure supplement 1.** Mapping of spike substitutions in rVSV/SARS-CoV-2/GFP escape mutants obtained in the presence of the corresponding nanobody.

**Figure supplement 2.** Comparison between RBD-Nb21 interface modeled with Integrative Modeling Platform (IMP) (red) with cryo-EM structure (7N9B) of the co-complex (indigo, taken from *Sun et al., 2021*).

**Figure supplement 3.** Nanobody groups resistant or predicted to be resistant against SARS-CoV-2 variants of concern (VOC).

nanobodies are neutralizing, with many escape mutants mapping adjacent to the receptor-binding motif (RBM), the region of RBD that interacts directly with ACE2 and is most lightly glycosylated (*Shajahan et al., 2020*; *Watanabe et al., 2020*), whereas ~20% of our anti-S2 nanobodies and ~60% of our non-RBD anti-S1 nanobodies are neutralizing. We note that, based on the fact that glycosylation obscures a considerable fraction of the spike surface (*Watanabe et al., 2020*; *Zhang et al., 2020*), our repertoire explores much of the remaining available epitope space. The neutralization bias that we observe also likely reflects the most obvious mechanism of viral inhibition, namely, blocking the binding of spike's RBD domain to ACE2 on host membranes to preclude viral fusion, but the non-RBD-based neutralization also underscores that other important mechanisms for viral inhibition exist.

Our RBD-binding nanobodies fall into at least seven groups (*Figure 6A*). Many of our mapped nanobodies bind epitopes that partially overlap with previously defined classes of IgG-binding epitopes, but are more compact due to the smaller nanobody paratopes (*Corti et al., 2021*; *Barnes et al., 2020*; *Xu et al., 2021*); however, many others define previously unreported binding sites. Overlapping with the RBM are groups 3 (S1-48, S1-RBD-40, S1-23, S1-37, S1-36) and 4 (S1-RBD-29, S1-RBD-21, S1-62); group 3 partially overlaps with previously defined site Ia/class I and group 4 partially overlaps with site Ib/class 2; however, S1-48 uniquely extends beyond the site Ia/class 1 sitting in the saddle of the RBM. Moving further out from the RBM, group 5 (S1-RBD-16 and S1-RBD-23) binds adjacent to the RBM (on the right of the RBD in *Figure 6*) and partially overlaps with site IV/class 3. Group 1 (S1-RBD-15, S1-1, S1-RBD-22, S1-RBD-24, S1-RBD-9) overlaps with site IIa/class 4 and partially overlaps with the opposite side of the RBM as groups 3 and 4. This site appears to be a common nanobody epitope and is shared by VHH-U, VHH-V, and WNb 10 (*Koenig et al., 2021*; *Pymm et al., 2021*). However, a number of our nanobodies map to epitopes that appear distinct from those previously described: group 6 is represented by a single nanobody (S1-RBD-35) and also binds the RBM at a site partially overlapping with site Ib/class 2 (*Corti et al., 2021*; *Tortorici et al., 2020*; *Dejnirattisai et al., 2021*), but is distinguished by its binding to the tip of the RBM at the left of this representation (*Figure 6*). S1-46 represents group 7 and defines a unique binding site, adjacent to site IIc/class 4, but higher on the RBD and closer to the RBM. Adjacent and to the right of group 1 is a unique binding site we define as group 2, represented by S1-6. Outside of the RBD, group 8 (represented by S1-49), like many IgGs, binds to the NTD of spike (*McCallum et al., 2021*). However, groups 9 (S2-10) and 10 (S2-40) are unique in binding to S2, with S2-10 binding to a region proximal to spike's heptad repeat 1 and S2-40 binding to a cleft between subunits on the stalk of S2 (*Wrapp et al., 2020b*; *Walls et al., 2020*). Of the 10 groups into which our nanobodies were classified, 7 (groups 1, 2, 5, and 7 targeting RBD; group 8 targeting the S1-NTD; groups 9 and 10 targeting S2) do not overlap with the mutations that distinguish alpha, beta, gamma, and delta SARS-CoV-2 VOCs (*Figure 6—figure supplement 3A*). Thus, to further validate our predictions for delta virus we tested two nanobodies against the virus in a plaque reduction neutralization assay. As predicted, the group 1 nanobody S1-1 effectively neutralized delta, whereas the group 6 nanobody S1-RBD-35 did not (*Figure 6—figure supplement 3B*). This confirms the potency of our repertoire against key VOC and underscoring the importance of generating large, diverse nanobody repertoires as presented here.

## Multiple modes of nanobody binding and neutralization

A subset of the 21 modeled nanobodies bind sites that interfere with ACE2 binding, preventing the virus from initial binding to its host cell (*Wrapp et al., 2020b*; *Walls et al., 2020*; *Starr et al., 2020*). Even here, more than one mechanism of inhibition can exist. The RBD is tethered to spike through a hinge, allowing it to fluctuate between either a 'down' conformation, hiding the vulnerable RBM from the host immune system, or an 'up' conformation, exposing the RBM for potential ACE2 binding and so spike activation/disassembly (*Corti et al., 2021*; *Cai et al., 2020*). Group 1, with overlapping and adjacent epitopes to the RBM, should only bind to the RBD when it is in the 'up' conformation (*Figure 6*). In contrast, groups 3, 4, and 6 also have overlapping and adjacent epitopes to the RBM, but appear agnostic to RBD conformation and may bind to both 'up' and 'down' positions of the RBD. For nanobodies whose epitopes actually overlap significantly with the RBM (e.g., S1-48, S1-62, S1-RBD-15, or S1-RBD-35) (*Figure 6*), neutralization may occur by directly competing with ACE2 and preventing its binding. For nanobodies within these groups with epitopes more adjacent to the RBM (e.g., S1-36, S1-1, or S1-RBD-9), neutralization may occur through sterically occluding ACE2 binding. However, interestingly, several nanobodies sharing similar epitope bins as S1-RBD-9 (group 1), such as S1-RBD-34, S1-RBD-19, S1-RBD-25, S1-RBD-32, and S1-RBD-36, do not neutralize spike (*Table 1*), suggesting that neutralization may occur via an additional mechanism. Alternatively, different nano-bodies can engage a shared epitope with different binding orientations that may or may not hinder ACE2 binding. Additionally, the binding of nanobodies in groups 1, 3, 4, and 6 may mimic ACE2 binding, thus trapping the RBD in its 'up' position to either catalyze the spike trimer rearrangements that prematurely convert spike into a post-fusion state, suppressing viral fusion, or destabilizing the trimer to cause its premature disassembly (*Huo et al., 2020b*; *Walls et al., 2019*; *Liu et al., 2020a*; *Turoňová et al., 2020*; *Benton et al., 2020*; *Cai et al., 2020*; *Koenig et al., 2021*).

Nanobodies in groups 2, 5, and 7 bind sites distal to the RBM, and therefore are unlikely to neutralize spike through direct ACE2 competition (*Figure 6*). However, we speculate that they neutralize spike via similar mechanisms. Group 5 (S1-RBD-16, S1-RBD-23) binds to the exposed face of the RBD, and its binding is likely to be agnostic to RBD conformation, and group 2 (S1-6) binds at a position adjacent to group 1, which is enough of a shift that it also may bind both 'up' and 'down' RBD conformations. Group 7 (S1-46) has a peculiar epitope that is only exposed in the 'up' conformation, so nanobody binding likely sterically blocks additional RBDs from accessing the 'up' position (although S1-46 should also be able to bind more than one simultaneously 'up' RBD). In each of these cases, nanobody binding is expected to stabilize the fluctuating RBD in its 'up,' ACE2-engaging, position, potentially destabilizing the trimer similar to mimics of ACE2.

S1-49, which is a member of group 8 (*Figure 6*), binds to the NTD for which neutralization activities remain unclear (*McCallum et al., 2021*). Human monoclonal antibodies specific to the NTD have been shown not to inhibit ACE2 binding and are instead proposed to inhibit viral infection by blocking membrane fusion, interaction with a different receptor, or proteolytic activation of spike (*McCallum et al., 2021*). It remains to be determined if these mechanisms of neutralization hold for our nano-bodies that bind non-RBD domains of S1, or if S1-49 suppresses ACE2 binding. The human monoclonals that neutralize the virus by binding outside of the RBD, and their yet to be discovered orthogonal mechanisms of neutralization, emphasize the potential and need for further characterization of our large repertoire of nanobodies.

The S2 domain is also a prime, but largely unexplored, therapeutic target (*Elshabrawy et al., 2012*; *Shah et al., 2021*). It is also not where the great majority of mutants in the current VOCs map, making it a particularly exciting target for potentially universal and VOC-resistant therapeutics. Here, we present the first neutralizing nanobodies that bind to S2 (groups 9 and 10) (*Figures 2, 4 and 6*, *Figure 6—figure supplement 1*). Some monoclonal antibodies that target S2 have been identified and shown to have neutralizing activity, but to our knowledge none have been structurally mapped (*Andreano et al., 2021*; *Wang et al., 2021c*; *Poh et al., 2020*; *Li et al., 2020*; *Song et al., 2021*). Because S2's function is primarily membrane fusion rather than receptor binding, the nanobodies' neutralization mechanisms must differ from those discussed above. For example, the S2-10 escape mutant S982R (*Figure 6—figure supplement 1*) indicates binding at S982 of spike, positioned at the end of the highly conserved heptad repeat 1, within a region of the S2 that undergoes large dynamic changes as the protein adopts a post fusion conformation; this suggests that S2-10 may restrict this conformational change, thereby inhibiting viral fusion (*Cai et al., 2020*; *Pierri, 2020*; *Turoňová et al.,*

*2020*; *Walls et al., 2020*). Notably, the region proximal to S982 appears accessible through an ~30 Å portal, even in the prefusion form with the RBDs in the 'up' position. This is a size not inconsistent with the binding face of a diminutive nanobody but likely inaccessible to conventional antibodies, as has been suggested by others (*Xu et al., 2021*). S2-40 uniquely sits at the interface between spike subunits, which raises fascinating possibilities for its neutralizing activity, perhaps involving the alteration of spike's quaternary structure or dynamics.

Nanobodies, as monomeric proteins, can provide a unique opportunity to define possible mechanisms of activity that may otherwise be difficult to distinguish. For example, the dimeric nature of conventional antibodies can introduce ambiguities regarding the mechanisms of neutralization because they can operate either as individual or pairwise binders. In the latter case, they may operate, for example, by aggregation (*Thomas et al., 1986*), increased avidity, enhanced steric hindrance via the larger binding entity, or by simultaneously binding and locking two separate moieties within a viral particle. In some cases, for example, S1-7 and S1-25 (which are non-neutralizing as monomers), dimerization did not convert them into neutralizers. In other cases, dimerization and trimerization can engender several folds to orders of magnitude increased neutralization potency (e.g., S1-RBD-35 and S1-23, respectively) (*Figure 4*, *Table 4*). We even have a curious case where a nanobody such as S2-7 that is essentially non-neutralizing as a monomer becomes strongly neutralizing upon dimerization (*Figure 4*). In this latter case, aggregation is a possible contributory mechanism, both between virions – which would lower effective virion concentration – or within a virion, with adjacent spike trimers being cross-linked to each other, inhibiting their function. Although a tremendous range in neutralization improvements by oligomerization is observed both by us and others, there is likely a limit to how much improvement can be induced by oligomerization as, for example, the trimers of S1-23 and S1-RBD-35 do not show a similar fold improvement as to what was observed for the monomer to dimer transition (*Figure 4D*, *Table 4*; *Schoof et al., 2020*; *Xiang et al., 2020*; *Koenig et al., 2021*; *Xu et al., 2021*; *Ma et al., 2021*).

## Synergistic activity with nanobody combinations

Drugs are often combined to improve single-agent therapies and dramatically enhance the therapeutic potential of either drug alone while reducing the drug concentrations to be administered. Synergy occurs when the combination of drugs has a greater effect than the sum of the individual effects of each drug. For example, tixagevimab and cilgavimab, two human monoclonal antibodies that target non-overlapping regions of the RBD, function synergistically and show promise as prophylactic and therapeutic agents against COVID-19 (*Dong et al., 2020*; *Zost et al., 2020*).

A major advantage of a large repertoire of nanobodies that bind to different epitopes on spike is their strong potential for cooperative activity among nanobody pairs (or higher-order combinations), leading to synergistic viral-neutralizing effects. The small size of nanobodies also provides a great advantage over much larger immunoglobulins in this context as the binding of a nanobody has a lower chance of sterically occluding the binding of a second nanobody to a distinct epitope and because they are monovalent. Moreover, as discussed above, nanobodies binding to the RBD may stabilize the otherwise 'up'-'down' fluctuating RBD in its 'up,' ACE2-engaging, position (*Figure 6*; *Xiang et al., 2020*; *Schoof et al., 2020*; *Bracken et al., 2021*). This can have three effects, all of which can potentially promote nanobody synergy: first, it will increase the effective on rate for spike trimer to that measured for monomer, and therefore, make it easier for complementary nanobodies to bind and inhibit; second, by stabilizing this 'up' for any one of the three RBDs in each spike trimer, it destabilizes the 'down' position for the remaining two RBDs, again making nanobody binding from the second class more likely; and third, the 'up' position exposes additional nanobody epitopes that would otherwise be buried (*Figure 6*; *Xiang et al., 2020*; *Sun et al., 2021*).

Using an automated platform, we titrated pairwise combinations of nanobodies in a 2D dilution format and measured their IC50s in the pseudovirus assay. IC50s were modeled using a multifaceted synergy framework (*Wooten et al., 2021*), including a parameterized version of the equivalent dose model (*Zimmer et al., 2016*), the Bivariate Response to Additive Interacting Doses (BRAID) model (*Twarog et al., 2016*), and the multidimensional synergy of combinations (MuSyC) model, which models a two-dimensional (2D) Hill equation and extends it to a 2D surface plot (*Meyer et al., 2019*). Synergy is evidenced by the parameters of the respective models (*Table 9*). To select nanobody pairs to test for synergy, we took advantage of epitope mapping, structural data, and biophysical

**Table 9.** Nanobody synergy of neutralization activity; related to *Figure 7*. Parameters from modeling the synergy observed for the indicated nanobody pairs. Multidimensional synergy of combinations (MuSyC), equivalent dose, and Bivariate Response to Additive Interacting Doses (BRAID) models were used to determine if statistically significant synergy was evident from the neutralization response in a 2D grid of nanobody concentrations.

| First nanobody | Second nanobody | (Nanobody #1) experimental range | (Nanobody #2) experimental range | h1 | h2 | C1 (nM) | C2 (nM) | alpha12 | alpha21 | a12 | a21 | Kappa |
|---|---|---|---|---|---|---|---|---|---|---|---|---|
| S1-23 | S1-27 | 4.1 pM to 717 nM, 0 µM | 3.4 pM to 600 nM, 0 µM | 1.80 | 1.46 | 25.0 | 20.0 | n.s. | n.s. | n.s. | n.s. | n.s |
| S1-1 | S1-23 | 2.2 pM to 387 nM, 0 µM | 4.1 pM to 717 nM, 0 µM | 0.93 | 1.51 | 4.2 | 18.6 | 21.4 | 31.8 | Synergy | Synergy | Synergy |
| S1-RBD-15 | S1-23 | 3 pM -to 527 nM, 0 µM | 4.1 pM to 717 nM, 0 µM | 1.32 | 1.42 | 3.7 | 10.3 | 300.0 | 10.2 | Synergy | Synergy | Synergy |
| S1-RBD-15 | S1-RBD-23 | 3 pM to 527 nM, 0 µM | 1.8 pM to 325 nM, 0 µM | 1.44 | 1.00 | 4.4 | 7.9 | 523 | 2.9 | Synergy | n.s. | Synergy |
| S1-23 | S1-46 | 4.1 pM to 717 nM, 0 µM | 7.55 pM to 1.34 µM, 0 µM | 1.76 | 0.75 | 3.3 | 273.0 | 50.1 | 1.0 | Synergy | Antagonism | Synergy |
| S1-RBD-15 | S1-46 | 39.7 pM to 7.04 µM, 0 µM | 7.55 pM to 1.34 µM, 0 µM | 1.10 | 0.75 | 4.9 | 164 | 10.7 | 0.9 | Synergy | Antagonism | Synergy |
| S1-23 | S2-10-dimer | 4.1 pM to 717 nM, 0 µM | 5.1 pM to 897 nM, 0 µM | 1.48 | 1.14 | 5.4 | 45.4 | 4232.0 | 2.8 | Synergy | n.s. | Synergy |
| S1-49 | S1-1 | 2.1 pM to 367 nM, 0 µM | 2.2 pM to 387 nM, 0 µM | 0.77 | 1.62 | 152.0 | 1.7 | 1,147 | 18.1 | Synergy | Synergy | Synergy |
| S1-49 | S1-RBD-15 | 2.1 pM to 367 nM, 0 µM | 39.7 pM to 7.04 µM, 0 µM | 0.85 | 1.20 | 629.0 | 2.2 | 243.5 | 20.3 | n.s. | Synergy | Synergy |
| S1-RBD-15 | S2-10-dimer | 3 pM to 527 nM, 0 µM | 5.1 pM to 897 nM, 0 µM | 1.57 | 1.00 | 1.9 | 56 | 4,110 | 2.2 | Synergy | Synergy | Synergy |

n.s., not significant; h1, h2, Hill slope; C1, C2, IC50 (nM);alpha12, alpha21, synergistic/antagonistic fold change of potency from MuSyC model; a12, a21, equivalent dose model synergy/antagonism; kappa, BRAID model synergy/antagonism.

characterization. We tested pairwise combinations of nanobodies that bind to similar epitopes, to different epitopes on RBD, and to regions outside and within the RBD (*Figure 7*, *Figure 7—figure supplement 1*, *Table 9*). Based on our structural mapping (*Figure 6*), we were able to infer some of the major molecular mechanisms by which these synergistic effects may occur.

Combinations of S1-27 and S1-23 showed simple additive effects (*Figure 7A*). These nanobodies belong to the same epitope bin (*Figure 3A*); their additive effect is as expected for two nanobodies accessing the same site on S1-RBD, but, for example, in equal concentrations, effectively doubling the concentration of a single nanobody. The potential for synergy resides instead in nanobodies that bind to different epitopes and can bind to spike monomers simultaneously. We therefore tested combinations that bind to different epitopes, first focusing on the RBD. Indeed, powerfully synergistic effects were observed between numerous nanobody pairs. For example, the combination of S1-23 and S1-1, which bind to opposite sides of the RBD, dramatically increased the potency of both nanobodies by ~32- and ~ 21-fold, respectively (*Figure 7B*, *Table 9*). S1-1 is expected to bind to RBD in its 'up' position, while S1-23 can bind to both 'up' and 'down' RBDs. We interpret this reciprocal synergy observed as S1-23 promoting the 'up' position and S1-1 stabilizing the 'up' position. S1-RBD-15, which binds to a similar epitope as S1-1, shows corresponding synergy with S1-23, suggesting that synergy may be predictable based on epitope mapping (*Figure 7C*). In this case, however, S1-RBD-15 had a greater influence on S1-23, promoting its potency by ~300-fold. S1-RBD-15 showed a comparable synergy profile against S1-RBD-23, which binds to the opposite side of the RBD, and adjacent to the site occupied by S1-23 (*Figures 3A, 6A and 7D*). The synergy profiles observed by these pairs of nanobodies highlight how stabilizing the RBD in the 'up' position can have a dramatic effect on their ability to neutralize spike activity. However, it should not be taken for granted that simply binding to distinct epitopes on RBD simultaneously will always be sufficient to generate a strongly synergistic response. For example, S1-46 failed to show synergy with either S1-23 or S1-RBD-15 (*Figure 7E and F*). Indeed, synergy modeling indicates that S1-46 actually mildly antagonizes both S1-23 and S1-RBD-15 (*Table 9*). S1-46 binds an epitope on RBD only in the 'up' position (*Figure 6*), but in this case binding hinders the movement of adjacent RBDs, and therefore reduces the activity of nanobodies depending on the dynamics of adjacent RBDs.

S1-49, which binds to the NTD of spike (*Figure 6*), substantially improved the neutralization potency of either S1-1 or S1-RBD-15 (*Figure 7G and H*). The synergy observed with S1-49 with S1-1 elicited a >1000-fold increase in potency. Interpretation of the mechanism underlying this remarkable synergistic effect will require a greater understanding of the mechanism of NTD activity on spike (*McCallum et al., 2021*).

We also tested for synergy between nanobodies targeting the S1-RBD and S2, which revealed remarkable results. In this case, we used a dimer of S2-10 to increase its potency to be closer to that of the nanobodies to which it was paired. Interestingly, among all the nanobody pairs that we tested, the synergy was greatest with the S2-10-dimer, which showed >4000-fold increase in potency when combined with either S1-23 or S1-RBD-15 (*Figure 7I and J*). S2-10 recognizes a site occluded by the S1-RBD when it is in its 'down'' position, but is revealed and becomes accessible when in its 'up' position. Thus, we interpret the mechanism of synergy as one of cooperativity where these RBD-binders promote the 'up' state and provide increased access for S2-10. The strong synergy observed may also reflect the distinct mechanisms by which either the RBD-binders or S2-10 operate individually.

Our repeated observation of strong synergistic effects between nanobodies is especially noteworthy, reflecting the unique properties of nanobodies such as their small size that are not shared by, for example, human monoclonal antibodies. For example, we structurally aligned human IgGs with the S1-23 and S1-1 paratopes and found the same binding characteristics would not be predicted to act synergistically because they could not bind the same monomer simultaneously as they would clash with other RBDs on the spike trimer. In the case of S1-23 and S1-RBD-15 epitopes, structural alignment of IgGs with nanobody paratopes suggested a strong inter-IgG steric clash in addition to a clash with other RBDs in the down position. While native nanobodies' small size and lack of an Fc domain affect their pharmacokinetic behavior in comparison to standard IgGs when used therapeutically, numerous studies have demonstrated that various modifications, such as albumin-binding domains or a synthetic Fc, are available to tune half-life and other behaviors upon either intravenous or intranasal delivery (*Tijink et al., 2008*; *Nambulli et al., 2021*; *Pymm et al., 2021*; *Shen et al., 2021*).

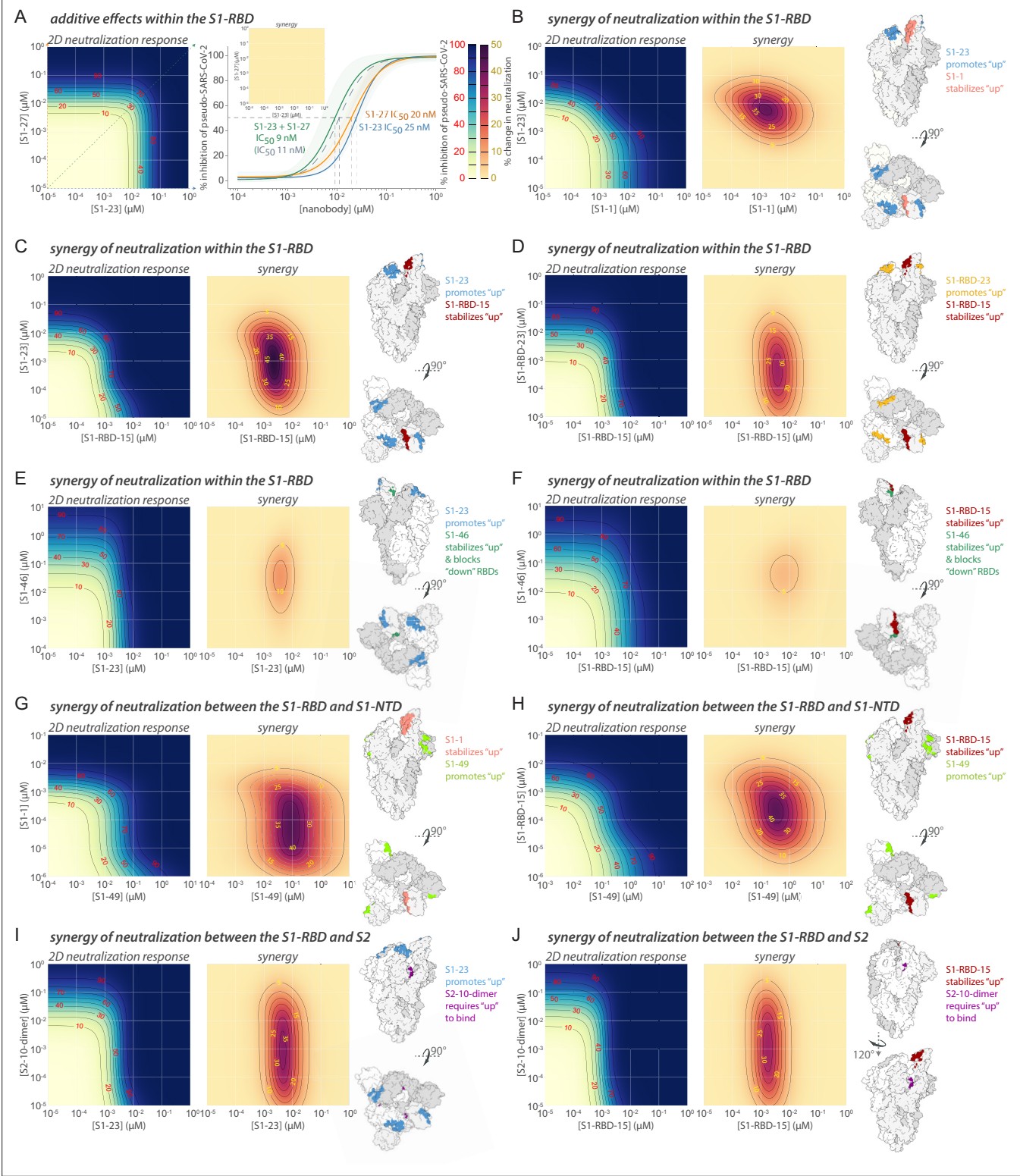

**Figure 7.** Synergistic neutralization of spike with nanobody cocktails. (**A**) An example of additive effects between two anti-SARS-CoV-2 spike nanobodies. S1-23 and S1-27 were prepared in a two-dimensional serial dilution matrix and then incubated with SARS-CoV-2 pseudovirus for 1 hr before adding the mixture to cells. After 56 hr, the expression of luciferase in each well was measured by addition of Steady-Glo reagent and read out on a spectrophotometer. The left panel shows a heatmap of pseudovirus neutralization by a two-dimensional serial dilution of combinations of S1-23 and S1-27. Lines and red numbers demarcate the % inhibition, that is, inhibitory concentration where X% of the virus is neutralized, e.g., IC50. Dark

*Figure 7 continued on next page*

*Figure 7 continued*

blue regions are concentrations that potently neutralize the pseudovirus, as per the heatmap legend. The right panel shows neutralization curves (with 90% confidence interval bands) and the calculated IC50 of each nanobody alone, or in a 1:1 combination was determined along with a calculated IC50 based on the theoretical additive mixture model of the pair (curve with dotted gray line). The inset shows a difference (synergy) map calculated as the difference between the parameterized 2D neutralization response and that expected in a null model of only additive effects. Here, no difference is observed. (**B**) S1-1 synergizes with S1-23 in neutralizing SARS-CoV-2 pseudovirus. The left panel shows the heatmap of pseudovirus neutralization observed by a two-dimensional serial dilution of combinations of S1-1 and S1-23. The middle panel shows a heatmap mapping the synergy of neutralization observed for this pair. The lines bounding the darker purple areas demarcate regions in the heatmap where the observed neutralization is greater than additive by the indicated percentages (yellow numbers), as per the heatmap legend. The right panel shows two representations of spike with the accessible S1-1 (salmon) and S1-23 (steel blue) epitopes (PDB ID: 6VYB). (**C–J**) Examples of synergy between nanobodies binding the S1-RBD, or between the S1-RBD and S1-NTD or S2 domains of spike. The layout is as found in (**B**), but comparing S1-RBD-15 with S1-23 (**C**), S1-RBD-15 with S1-RBD-23 (**D**), S1-23 with S1-46 (**E**), S1-RBD-15 with S1-46 (**F**), S1-49 with S1-1 (**G**), S1-49 with S1-RBD-15 (**H**), S1-23 with S2-10-dimer (**I**), and S1-RBD-15 with S2-10-dimer (**J**).

The online version of this article includes the following source data and figure supplement(s) for figure 7:

**Source data 1.** Neutralization data from synergy experiments.

**Figure supplement 1.** Heatmaps of nanobody synergy.

## Perspectives

The data presented here demonstrate the power of raising large and diverse repertoires of nanobodies against the entire ectodomain of SARS-CoV-2 spike to maximize the likelihood of generating potent reagents for prophylactics and therapeutics. Moreover, several neutralizing nanobodies in our current repertoire are, or are predicted to be, effective against current circulating variants (*Figure 6—figure supplement 3*). However, our escape experiments support the idea that the current circulating variants are not yet necessarily exploring the full potential of the virus to escape our current and emerging therapeutic arsenals, and that even if antibodies or nanobodies are resistant to the current variants, they will not necessarily be resistant to variants as they continually emerge. To counteract this eventuality, we show that judicious choice of nanobody combinations that can synergize and have orthogonal and complementary neutralization mechanisms have the potential to result in potent and broadly neutralizing reagents that are resistant to viral escape. Collectively, this unique and readily modifiable repertoire has the potential to complement vaccines, drugs and single epitope reagents, and guard against single-molecule failure in human trials even in the face of emerging variants. Most urgently, it paves the way to develop therapeutics for hospitalized patients with acute disease, and address the unmet needs of patients in the developing world, many of which will not see a COVID-19 vaccine before 2023 (*Padma, 2021*).

## Materials and methods
### Key resources table

See Appendix 1—key resources table.

### Summary of key improvements to nanobody generation pipeline

To maximize the purity of the serum HCAb sample, we explored different binding conditions to select for the tightest V$_H$H binders – a key step not generally available to display panning methods (*Fridy et al., 2014a*). We also used an additional HCAb purification step to deplete VH IgG by incubation with immobilized Protein M, a mycoplasma protein specific for IgG light chain (*Grover et al., 2014*). To further enrich the V$_H$H sample for MS analysis and remove Fc, we performed a digest with IdeS, a protease that cleaves the V$_H$H domain from the HCAb with higher specificity than conventionally used papain (*von Pawel-Rammingen et al., 2002*). Greater peptide coverage for LC-MS was attained by using complementary digestion with trypsin and chymotrypsin (*Xiang et al., 2021*), augmented by partial SDS-PAGE gel-based separation of different V$_H$Hs to reduce the V$_H$H complexity and to give more complete peptide coverage and candidate selection. We redesigned PCR primers to maximize coverage of V$_H$H sequences for our cDNA libraries. Also, to increase the reliability of the library, singletons were not considered as candidates and priority was given to sequences with high counts. Finally, we refined our Llama-Magic software package (*Fridy et al., 2014a*) to include

improved scoring functions, weighting the length, uniqueness, and quality of the MS data especially for complementarity-determining regions. This optimized protocol allowed us to identify 374 unique CDR3 sequences (from 847 unique $V_HH$ candidates). Details are provided below.

## Antigens

Recombinant Fc-tagged SARS-CoV-2 spike S1 and S2 proteins purified from HEK293 cells were used for llama immunization (The Native Antigen Company; REC31806 and REC31807). For affinity isolation, binding screens, SPR analysis, and MP, recombinant spike S1-His, untagged RBD, or S2-His proteins expressed in HEK293 (S1 and RBD), or insect cells (S2) were used (Sino Biological; 40591-V08H, 40592-VNAH, and 40590-V08B). Native mass spectrometry (*Olinares et al., 2016*; *Olinares et al., 2021*) was used to confirm the quality of these proteins and determine their glycosylation state, with S1 and S2 observed to be heavily glycosylated (at least 10 kDa of attached glycans). RBD was observed to be monomeric, S1-His likely monomeric, and S2-His, a mix of monomer and trimer.

## Immunization and isolation of $V_HH$ antibody fractions

We used a pre-screening protocol to select llamas with naturally strong immune responses, as determined by activity against standard animal vaccines (*Thompson et al., 2016*). Two llamas, Marley (9-year-old male) and Rocky (5-year-old male), were immunized with recombinant SARS-CoV-2 spike S1 and SARS-CoV-2 spike S2 expressed in HEK293 cells as Fc fusion proteins. Llamas were injected subcutaneously with 0.25 mg of each antigen with CFA, then boosted with the same amounts with IFA at intervals of 14, 7, 21, and 10 days. Serum bleeds and bone marrow aspirates were obtained 9 days after the final injection. From the production serum bleeds, HCAb fractions of IgG were obtained by purification with immobilized Protein A and Protein G as previously described (*Fridy et al., 2014a*). Residual light-chain-containing IgG was removed from this fraction by incubating with 25 µl of 10 mg/ml Protein M-Sepharose per mg of HCAb (*Grover et al., 2014*). After a 30 min incubation, the HCAb flow-through was collected. 12 mg of this HCAb fraction was then incubated with Sepharose-conjugated recombinant SARS-CoV-2 spike S1-His, RBD, or S2-His protein. This resin was washed with (1) 20 mM sodium phosphate, pH 7.4 + 500 mM NaCl; (2) 2 M $MgCl_2$ in 20 mM Tris, pH 7.5; (3) PBS + 0.5 % Triton X-100; and (4) PBS. The resin was then resuspended in a 200 µl solution of 2 U/µl IdeS enzyme (Genovis) in PBS and digested for 3.5 hr at 37°C on an orbital shaker. The resin was then washed with (1) PBS, (2) PBS plus 0.1% Tween-20, and (3) PBS. Bound protein was eluted by incubating 10 min at 72°C in 1× NuPAGE LDS sample buffer (Thermo Fisher). The samples were reduced with DTT and alkylated with iodoacetamide, then run on a 4–12% Bis-Tris gel. Bands at ~15 kDa and ~20 kDa corresponding to digested $V_HH$ region were then cut out and prepared for MS.

## RT-PCR and DNA sequencing

Bone marrow aspirates were obtained from immunized llamas concurrent with production serum bleeds. Bone marrow plasma cells were isolated on a Ficoll gradient using Ficoll-Paque (Cytiva). RNA was isolated from approximately $3–4 × 10^7$ cells using TRIzol reagent (Thermo Fisher), according to the manufacturer's instructions. cDNA was synthesized using SuperScript IV reverse transcriptase (Thermo Fisher). A PCR was then performed with $V_HH$ IgG specific primers and Deep Vent polymerase (New England Biolabs). Forward primers 6N_CALL001 5'-NNNNNNGTCCTGGCTGCTCTTCTACAAGG-3' and 6N_CALL001B 5'-NNNNNNGTCCTGGCTGCTCTTTTACAAGG-3' target the leader sequence (*Conrath et al., 2001*) while reverse primers 6N_VHH_SH_rev 5'-NNNNNNCTGGGGTCTTCGCT GTGGTGC-3' and 6N_VHH_LH_rev 5'-NNNNNNGTGGTTGTGGTTTTGGTGTCTTGGG-3' target short and long hinge sequences at the 3' side of $V_HH$. Primers included six random bases (N) to aid cluster identification. The approximately 350–450 bp product of this reaction was gel purified, then ligated to Illumina adaptors before library preparation using Illumina kits, before MiSeq sequencing using two 300 bp paired end reads.

## Identification of nanobodies by mass spectrometry

Trypsin (Roche) or chymotrypsin (Promega) solution was added to previously reduced, alkylated, diced, destained, and dehydrated gel pieces at ~1:4–3:1 enzyme to substrate mass ratios. Gel pieces were allowed to rehydrate with enzyme solution for 10 min on ice. 45 µl of digestion buffer (trypsin: 50 mM ammonium bicarbonate, 10% acetonitrile; chymotrypsin: 100 mM Tris pH 7.8, 10 mM $CaCl_2$)

were then added, and samples were incubated for 6 hr at 37°C (trypsin) or 25°C (chymotrypsin). Supernatant was then removed from gel pieces and transferred to a new tube. 150 µl of a 1.67% FA, 67% ACN, 0.05% TFA solution were added to gel pieces, and shaken at 4°C for ~6 hr. Supernatant was removed from gel pieces, transferred to the tube with previous supernatant, and evaporated in a speedvac until dry. Samples were resuspended in 5% formic acid, 0.1% TFA, and cleaned on StageTips (*Rappsilber, 2012*).

Samples were analyzed with a nano-LC 1200 (Thermo Fisher) using an EASYspray PepMap RSLC C18 3 µm, 100 Å, 75 µm × 15 cm column coupled to an Orbitrap Fusion Lumos Tribrid mass spectrometer (Thermo Fisher). An Active Background Ion Reduction Device (ABIRD, ESI Source Solutions) was used to reduce background. The Lumos was operated in data-dependent mode, and top intensity ions were fragmented by high-energy collisional dissociation (normalized collision energy 28). Ions with charge states 2–5 were selected for fragmentation. Orbitrap resolution was 120,000. The quadrupole isolation window was 1.4, and the MS/MS used a maximum injection time of 250 ms with one microscan.

The initial identification of nanobody sequences was performed as described (*Fridy et al., 2014a*) using the program Llama-Magic (https://github.com/FenyoLab/llama-magic) with a few added features (including being able to deal with chymotryptic proteolysis and to rank V$_H$Hs by corresponding read counts in high-throughput sequencing data), where 23 MS datasets (concatenated from all MS acquisition data according to antigens, animal individuals, gel band positions and proteases) were independently searched. The results were filtered with criteria including read counts, uniqueness score, and quality and coverage of MS/MS fragments to generate a collection of high-confidence nanobody sequences. A CDR3 network graph was created by connecting nodes (unique high-confidence CDR3 sequences) by edges where a CDR3 pair has a Damerau–Levenshtein distance of no more than three by using NetworkX 2.5 (https://networkx.org) and pyxDamerauLevenshtein (https://github.com/gfairchild/pyxDamerauLevenshtein; *Fairchild, 2013*). The diversity of nanobodies for screening was maximized by selecting CDR3 sequences from isolated components of the network graph, together with varying CDR3 lengths and animal individual origin.

## Cloning and purification of nanobodies

Nanobody sequences were codon-optimized for expression in *Escherichia coli* and synthesized as gene fragments (IDT), incorporating BamHI and XhoI restriction sites at 5′ and 3′ ends, respectively. Nanobody sequences were then subcloned into pET21-pelB using BamHI and XhoI restriction sites as previously described (*Fridy et al., 2014a*). pelB-fused nanobodies were expressed and purified using Arctic Express (DE3) cells (Agilent) as previously described using TALON metal affinity resin (Takara) (*Fridy et al., 2014a*).

Nanobodies to be oligomerized were ordered from IDT as minigenes incorporating at the 5′ end a SalI site followed by codon optimized sequence for the linker GGGGSGGGGSGGGGSGGGGS upstream of the start codon of the nanobody cDNA, and at the 3′ end of the nanobody the coding sequence a XhoI site was added. The minigene was cut with SalI and XhoI, the linker-nanobody insert was gel purified and ligated with the XhoI linearized recipient nanobody expression vector (pET21-pelB + nanobody). Restriction digests and sequencing was performed to identify two (dimer) and three (trimer) oligomers.

## Nanobody screening

To validate nanobody candidates, pelB-fused nanobodies were expressed in 50 ml cultures of Arctic Express (DE3) cells, and the periplasmic fractions were isolated by osmotic shock as previously described (*Fridy et al., 2014a*). Spike S1-His, RBD, or S2-His proteins (Sino Biological 40591-V08H, 40592-VNAH, and 40590-V08B) were conjugated to cyanogen bromide-activated Sepharose 4 Fast Flow resin (Cytiva) according to the manufacturer's instructions using 100 µg protein per mg of resin. Periplasm was incubated with 15 µl of the corresponding antigen-conjugated Sepharose for 30 min while rotating at room temperature (RT). The resin was then transferred to a spin column and washed twice with buffer TBT-100 (20 mM HEPES pH 7.4, 100 mM NaCl, 110 mM KOAc, 2 mM MgCl$_2$, 0.1% Tween 20). Bound protein was eluted with 1.2× NuPAGE LDS sample buffer (Thermo Fisher) for 10 min at 72°C, then reduced with 50 mM DTT (10 min at 72°C). Input and elution samples were separated by SDS-PAGE, and Coomassie-stained bands were quantified using ImageJ software.

## Surface plasmon resonance

$K_D$s were determined via SPR experiments. Measurements were either taken on a Proteon XPR36 Protein Interaction Array System (Bio-Rad) or a Biacore 8k (Cytiva). Recombinant spike S1, RBD, and spike S2 were immobilized at 5 µg/ml, 5 µg/ml, and 12.5 µg/ml, respectively, using the ProteOn Amine Coupling Kit (EDC/NHS coupling chemistry, Bio-Rad) according to the respective manufacturer's guidelines either on a ProteOn GLC sensor chip or a Series S CM5 sensor chip. All purified nanobodies in a final buffer containing 20 mM HEPES pH 7.4, 150 mM NaCl, 0.02% Tween, were prepared in 5–8 concentrations. For experiments performed on the Proteon XPR36, protein was then injected at 50 µl/min for 120 s, followed by a dissociation time of 600 s. Residual bound proteins were removed by regenerating the chip surface using glycine pH 3 + 1 M $MgCl_2$. Data were processed and analyzed using the ProteOn Manager software. For experiments performed on the Biacore 8k, protein was injected at 60 µl/min for 120 s, followed by a dissociation time of either 1200 s or 2400s. Residual bound proteins were removed by regenerating the chip surface using glycine pH 2.5 + 1 M $MgCl_2$. Data were processed and analyzed using the Biacore Insight Evaluation software.

## Differential scanning fluorimetry

Nanobody melting temperatures ($T_m$) were measured by DSF using a CFX96 Real-Time PCR Detection System (Bio-Rad, Hercules, CA). A 96-well thin-wall hard-shell PCR plate (Bio-Rad) was set up with each well containing 10–40 µM of protein in 20 mM HEPES, 150 mM NaCl buffer (pH 7.4), 5× SyproOrange Protein Gel Stain (Sigma-Aldrich). Fluorescence variation was measured from 25 to 95°C at a ramp rate of 0.5°C/5 s. Excitation was between 515 and 535 nm, and emission was monitored between 560 and 580 nm. $T_m$ was the transition midpoint value, calculated using the manufacturer's software (*Niesen et al., 2007*).

## Lyophilization

Nanobodies in 20 mM HEPES, 150 mM NaCl, pH 7.4 at concentrations between 0.33 mg/ml and 0.63 mg/ml were snap-frozen in liquid nitrogen and dried in a speed-vac to replicate lyophilization conditions. Nanobodies were then reconstituted in $_{dd}H_2O$ and characterized using SPR and DSF.

## Epitope mapping of nanobodies

### Biolayer interferometry for epitope binning anti-RBD nanobodies

Epitope mapping studies were carried out using the Octet system (ForteBio, USA, version 7) that measures biolayer interferometry (BLI). All steps were performed at 30°C with shaking at 1300 rpm in a black 96-well plate containing 300 µl kinetics buffer (PBS; 0.2% BSA; 0.02% sodium azide) in each well. AMC-coated biosensors were loaded with mFc tagged RBD (Sino Biological) at 40 µg/ml to reach >1 nm wavelength shift following binding and washing. The sensors were then reacted for ~300 s with reference nanobodies and then transferred to kinetics buffer-containing wells for another 180 s. A new baseline was set, sensors were then reacted for 180 s with analyte nanobodies (association phase), and then transferred to buffer-containing wells for another 180 s (dissociation phase). Binding and dissociation were measured as changes over time in light interference after subtraction of parallel measurements from unloaded biosensors. Sensorgrams of analyte association/dissociation responses were analyzed using the Octet data analysis software 7.1 (Fortebio, USA, 2015). Analyte binding to mFc RBD was also measured in parallel to get response levels in the absence of the reference nanobodies.

Octet response values were used to compute a Pearson's correlation coefficient for pairwise combinations of nanobodies using Pandas (*McKinney, 2010*) in Python 3.7.6 (https://www.python.org/). These coefficients were then used to hierarchically cluster the nanobodies and were visualized as a heatmap (*Pedregosa, 2011*).

The undirected unweighted network graph of Octet response values was constructed by treating each nanobody as a node, adding an edge to each measured pair of different nanobodies, and setting the maximum response value of a nanobody pair as an attribute to the edge, by using NetworkX 2.5 (https://networkx.org). The least responses of pairwise nanobodies within all fully measured nanobody subsets were computed by iterating through all network cliques of size 2–14 by using NetworkX's 'find_cliques' function, and taking the minimum value of edge attributes within each clique. Network coefficients (average shortest path length, average clustering coefficient, and small-world coefficient

sigma) were computed using NetworkX's 'average_shortest_path_length,' 'average_clustering,' and 'sigma' functions. Network visualization was created by using D3.js (https://d3js.org).

## Mass photometry

Select nanobodies were binned using MP. Experiments were performed on a Refeyn OneMP instrument (Refeyn Ltd). The instrument was calibrated with a mix of BSA (Sigma-Aldrich), thyroglobulin (Sigma-Aldrich), and beta-amylase (Sigma-Aldrich). Coverslips (Thorlabs) and gaskets (Grace Bio-Labs) were prepared by washing with 100% IPA followed by $_{dd}H_2O$, repeated three times, followed by drying with HEPA filtered air. 12 µl of buffer was added to each well to focus the instrument after which 8 µl of protein solution was added and pipetted up and down to briefly mix after which movies/frame acquisition was promptly started. The final concentration in each experiment of recombinant spike S1 monomer (Sino Biological) and each nanobody was 30 nM and between 25 and 40 nM, respectively. Movies were acquired for 60 s (6000 frames) using AcquireMP (version 2.3.0; Refeyn Ltd) using standard settings. All movies were processed, analyzed, and masses estimated by fitting a Gaussian distribution to the data using DiscoverMP (version 2.3.0; Refeyn Ltd).

## Epitope mapping of anti-S2 and non-RBD anti-S1 nanobodies

SPR was utilized to perform epitope binning experiments using a Biacore 8k (Cytiva) supplemented with the Biacore Insight Epitope Binning Extension. All nanobodies' concentrations were ≥20× the concentration of their $K_D$ for binning experiments, with the majority surpassing their $K_D$ by 50×. For non-RBD anti-S1 nanobodies, experiments were performed utilizing either the tandem method or dual-tandem method for epitope binning, whereas for anti-S2 nanobodies, only the tandem method was utilized. Series S CM5 sensor chips immobilized with spike S1 and spike S2 were used (see 'Surface plasmon resonance' section above for full details). For the tandem method, nanobody '1' was injected at 10 µl/min for 240 s, followed by a brief wash, after which nanobody '2' was injected at 10 µl/min for 240 s and dissociated for 30 s. Residual bound proteins were removed by washing the chip surface four times with 10 mM glycine pH 2 + 1 M $MgCl_2$ at 60 µl/min for 60 s. For the dual-tandem method, nanobody '1' was injected at 10 µl/min for 120 s, followed by nanobody '2,' which was injected at 10 µl/min for 150 s and dissociated for 30 s. Residual bound proteins were removed by washing the chip surface three times with 10 mM glycine pH 2 + 1 M $MgCl_2$ at 60 µl/min for 60 s. For the anti-S2 nanobody binning experiments, residual bound proteins were removed by washing the chip surface first with 0.1 M HCl at 60 µl/min for 60 s, followed by a second wash with 3 M $MgCl_2$ at 60 µl/min for 60 s. Data were processed and analyzed using the Biacore Insight Evaluation software utilizing the Epitope Binning Extension.

## Cell lines

Vero E6 cells (ATCC) were cultured at 30°C in the presence of 5% $CO_2$ in medium composed of high-glucose Dulbecco's modified Eagle's medium (DMEM, Gibco) supplemented with 5% (v/v) heat-inactivated fetal bovine serum (FBS) (VWR). TMPRSS expressing Vero E6 cells (gift from Rhea Coler) were cultured in DMEM supplemented with 5% (v/v) FBS and 1 mg/ml geneticin. 293T/17 and 293T-hACE2 (*Cawford et al., 2020*) cells (Life Technologies; Cat# R70007; RRID:CVCL_6911) were cultured in DMEM (Gibco) supplemented with 10% FBS, penicillin/streptomycin, 10 mM HEPES, and with 0.1 mM MEM non-essential amino acids (Thermo Fisher). All experiments were performed with cells passaged less than 15 times. The identities of cell lines were confirmed by chromosomal marker analysis and tested negative for mycoplasma using a MycoStrip (InvivoGen).

## Production of SARS-CoV-1, SARS-CoV-2, and SARS-CoV-2 variant pseudotyped lentiviral reporter particles

Pseudovirus stocks were prepared using a modified protocol published by *Cawford et al., 2020*; *Qing et al., 2020*. Briefly, pseudovirus stocks were prepared by cotransfecting 4.75 µg pHAGE-CMV-Luc2-IRES-ZsGreen-W (BEI Cat # NR-52516) (*Cawford et al., 2020*), 3.75 µg psPAX and 1.5 µg spike containing plasmid using lipofectamine 3000 (Thermo Fisher). $4 \times 10^6$ cells were plated 16–24 hr prior to transfection. 60 hr post transfection, pseudovirus containing media was collected, cleared by centrifugation at 1000 × $g$, and filtered through a 0.45 µm syringe filter to clear debris. 1 ml aliquots were frozen at –80°C. Pseudovirus was titered by threefold serial dilution on 293T-hACE2

cells (*Cawford et al., 2020*), treated with 2 μg/ml polybrene (Sigma). Infected cells were processed between 52 and 60 hr by adding equal volume of Steady-Glo (Promega), and firefly luciferase signal was measured using the Biotek Model N4 with integration at 0.5 ms.

## SARS-CoV-2 pseudovirus neutralization assay

All periplasmic purified nanobodies were treated with Triton X-114 to remove any residual endotoxins so as to not have endotoxins contribute to the effective neutralization (*Aida and Pabst, 1990*), and residual detergent was removed using Pierce Detergent Removal Resin according to the manufacturer's instructions (Thermo Fisher). 293-hACE2 cells were plated at 2500–4000 cells per well on 384 solid white TC-treated plates. Threefold serially diluted nanobodies (10 dilutions in total) were incubated with 40,000–60,000 RLU equivalents of pseudotyped SARS-CoV-2-Luc for 1 hr at 37°C. Mock treatment and a sham treatment with LaM2 nanobodies (*Fridy et al., 2014a*) that do not bind to spike were included as negative controls while untreated wells were used to monitor background levels. The nanobody-pseudovirus mixtures were then added in quadruplicate to 293T-hACE2 cells along with 2 μg/ml polybrene (Sigma). Cells were incubated at 37°C with 5% $CO_2$. Infected cells were processed between 52 and 60 hr as described above. Data were processed using Prism 7 (GraphPad), using four-parameter nonlinear regression (least-squares regression method without weighting). All nanobodies were tested at least two times and with more than one pseudovirus preparation.

## Nanobody synergy

Experiments were performed as per our pseudovirus neutralization assay. A robotic liquid handler was used to prepare 2D matrices of serial dilutions of two nanobodies and then mix these with SARS-CoV-2 pseudovirus for 1 hr. After incubation with the virus, the mixture was overlaid on a monolayer of 293-hACE2 cells and left to incubate for 56 hr. Luminescence was quantified as described above. Data were processed using synergy software (*Wooten et al., 2021*).

## Structural analysis

Integrative structural modeling proceeded through the standard four-stage protocol (*Russel et al., 2012*; *Kim et al., 2018*; *Rout and Sali, 2019*; *Saltzberg et al., 2021*), which was scripted using the *Python Modeling Interface* package, a library for modeling macromolecular complexes based on the *Integrative Modeling Platform* software (*Russel et al., 2012*), version develop-31a0ad09b4 (https://integrativemodeling.org). Separate models were computed for rigid-receptor-rigid ligand-type binary docking of (1) 18 nanobodies (S1-RBD-[9,15,16,21,22,23,24,29,35,40] and S1-[1,6,23,36,37,46,48,62]) on a monomeric S1-RBD domain, (2) the S1-49 nanobody on a monomeric S1-NTD domain, and (3) S2-10 and S2-40 nanobodies on the trimeric S2 domain of the spike protein. Monomeric S1-RBD, spanning amino acids T333-G526, was represented using the crystal structure of the co-complex of ACE2 bound RBD (PDB ID: 6M0J; *Lan et al., 2020*). Monomeric S1-NTD, spanning amino acids V16-S305, was represented using the crystal structure of the S2M28 Fab bound NTD (PDB ID: 7LY3; *McCallum et al., 2021*). Trimeric S2 was represented using the residues S689-P1162 (for each monomer) from a 2.9 Å cryo-EM structure with PDB ID: 6XR8 (*Cai et al., 2020*, *McCallum et al., 2021*). Comparative models of all 21 nanobodies were built from the crystal structure of the human Vsig4 targeting nanobody Nb119 (PDB ID: 5IML; *Wen et al., 2017*) as template using MODELLER (*Sali and Blundell, 1993*), and their CDR3 regions were refined using MODELLER's loop modeling algorithm (*Fiser et al., 2000*). To maximize the efficiency of structural sampling while avoiding too much information loss, the system was represented at a resolution of one bead per residue, and the receptors and all nanobodies were treated as rigid bodies. For each nanobody, alternate binding modes were scored using spatial restraints enforcing receptor-ligand shape complementarity, cross-link satisfaction and proximity of CDR3 loops on the nanobodies to escape mutant residues on the corresponding receptor. With the receptor fixed in space, 1,200,000 alternate docked nanobody models were produced through 20 independent runs of replica exchange Gibbs sampling based on the Metropolis Monte Carlo algorithm, where each Monte Carlo step consisted of a series of random rotations and translations of rigid nanobodies. The initial set of models was filtered to select a random subsample of 30,000 models, which were clustered by the similarity of their interfaces to the receptor. The fraction of common contacts (fcc) between receptor and nanobody was used to characterize interface similarity between alternate nanobody poses (*Fiser et al., 2000*; *Rodrigues et al., 2012*). Binding poses belonging to

only the most populated cluster were selected for further analysis. Five independent random subsamples of 30,000 models each were generated from the set of all models post-structural sampling, and the entire protocol of interface similarity-based clustering and top cluster selection was repeated each time. However, no significant changes were observed in the satisfaction of restraints. Integrative models of nanobody epitopes on the spike protein were computed on the Wynton HPC cluster at UCSF. Receptor epitopes were visualized in UCSF ChimeraX (*Pettersen et al., 2021*; *Rodrigues et al., 2012*). Files containing input data, scripts and output results are available at https://github.com/integrativemodeling/nbspike (*Sanyal, 2021*; copy archived at swh:1:rev:2607a97503e1d7641079641142734f4075d334e2).

## SARS-CoV-2 stocks and titers

SARS-related coronavirus 2, isolate USA-WA1/2020, NR-52281, was deposited by the Centers for Disease Control and Prevention and obtained through BEI Resources, NIAID, NIH. SARS-CoV-2, isolate USA (B.1.617.2), was a kind gift from Rhea Coler. Viral stocks were propagated in Vero E6 cells. All experimental work involving live SARS-CoV-2 was performed at Seattle Children's Research Institute (SCRI) in compliance with SCRI guidelines for BioSafety Level 3 (BSL-3) containment. An initial inoculum was diluted in Opti-MEM (Gibco) at 1:1000, overlaid on a monolayer of Vero E6 and incubated for 90 min. Following the incubation, the supernatant was removed and replaced with 2% (v/v) FBS in Opti-MEM medium. The cultures were inspected for cytopathic effects, which were prominent after 48 hr of infection. After 72 hr, infectious supernatants were collected, cleared of cellular debris by centrifugation, and stored at –80°C until use. Viral titers were determined by plaque assay using a liquid overlay and fixation-staining method, as described (*Mendoza et al., 2020*; *Case et al., 2020*). Briefly, serially diluted virus stocks were used to infect confluent monolayers of Vero E6 cells (~$1.2 \times 10^6$ cells per well) cultured in six-well plates. After a 90 min incubation, the virus was removed, and the cell monolayer overlaid with a medium composed of 3% (w/v) carboxymethylcellulose and 4% (v/v) FBS in phenyl-free Opti-MEM. 96 hr post infection, the viscous carboxymethylcellulose medium was removed and the cells were washed once with Dulbecco's phosphate buffered saline (DPBS; Gibco) before being fixed with 4% (w/v) paraformaldehyde in DPBS. After a 30 min incubation, the fixative was removed, and the cells were rinsed with DBPS before being stained with 1% (w/v) crystal violet in 20% (v/v) ethanol. Contrast was enhanced by successive washes with DPBS, and clear plaques representing individual viral infections were visualized as spots lacking the stain. Plaques were enumerated by first identifying the dilution factor of the well containing 10–100 plaques. After counting the plaques, the average number of plaque forming units (pfus) from three experiments was used to determine the viral titer by dividing the average by the dilution factor and volume of virus delivered per well.

## Focus forming reduction assay with authentic SARS-CoV-2

Nanobody neutralization of infectious SARS-CoV-2 was performed using a focus forming reduction assay. Briefly, eight threefold serial dilutions of nanobodies were incubated with ~$7.5 \times 10^4$ focus forming units of SARS-CoV-2 for 1 hr at RT. The mixture was then added to a confluent monolayer of Vero E6 cells or 293-ACE2 (*Cawford et al., 2020*) plated at ~$1.5 \times 10^5$ cells per well and seeded in 48-well plates. 24 hr post infection, the cells were washed once with DPBS, trypsinized with 0.05% trypsin (Gibco), and fixed for 30 min with 4% paraformaldehyde in DPBS. After fixation, the cells were permeabilized with 1% (w/v) Triton X-100 (Sigma Aldrich) for 30 min. After permeabilization, the cells were incubated with a blocking buffer (1% [w/v] bovine serum albumin [Calbiochem] and 0.5% [w/v] Triton X-100 in DBPS) for 60 min, and then stained with primary anti-spike CR3022 (Absolute Antibody) monoclonal antibodies (1:1000), and secondary anti-human IgG antibodies (1:2000) conjugated to Alexa Fluor 488 (Invitrogen). Cells staining positive for spike were measured by flow cytometry on a Becton Dickinson BD LSR II Special Order System Flow Cytometer With HTS Sampler. The percentage of spike-positive cells from triplicate wells for each dilution was used to determine the half maximal inhibitory concentrations (IC50) using a parametric 1D Hill fitting algorithm with synergy (*Wooten et al., 2021*). A mock treatment, sham treatment with LaM2 nanobodies (*Fridy et al., 2014a*), and untreated cells were used as controls.

## Plaque reduction neutralization test with authentic SARS-CoV-2

Nanobody neutralization of infectious SARS-CoV-2 was performed using a plaque reduction neutralization test (PRNT) assay. Briefly, 10 threefold serial dilutions of nanobodies were incubated with ~100–300 pfus of SARS-CoV-2 for 1 hr at RT. The mixture was then added to a confluent monolayer of TMPRSS2+ Vero E6 cells (~$6 \times 10^5$ cells per well) in 12-well plates. After a 90 min incubation, the virus was removed, and the cell monolayer overlaid with a medium composed of 3% (w/v) carboxymethyl-cellulose and 4% (v/v) FBS in phenyl-free Opti-MEM. 96 hr post infection, the viscous carboxymethyl-cellulose medium was removed and the cells were washed once with DPBS (Gibco) before being fixed with 4% (w/v) paraformaldehyde in DPBS. After a 30 min incubation, the fixative was removed, and the cells were rinsed with DBPS before being stained with 1% (w/v) crystal violet in 20% (v/v) ethanol. Contrast was enhanced by washing with DPBS, and clear plaques representing individual viral infections were visualized as spots lacking the stain. The number of plaques at each dilution was used to determine the IC50s of each nanobody.

## SARS-CoV-2 neutralization in primary airway epithelial cell (AEC) cultures

Assays with primary airway epithelial cell cultures were performed as described (*Barrow et al., 2021*). Briefly, bronchial AECs were obtained under study #12490 approved by the Seattle Children's Institutional Review Board, with investigations carried out following the rules of the Declaration of Helsinki of 1975. AECs were differentiated for 21 days at an ALI on 12-well collagen-coated Corning plates with permeable transwells in PneumaCult ALI media (Stemcell, Vancouver, BC, Canada). Differentiated AECs were treated with nanobodies diluted in PBS, or PBS alone for 60 min, the liquid was removed, and the AECs were then infected with SARS-CoV-2 at a multiplicity of infection (MOI) of 0.5. At 24 hr intervals, the cells were treated with nanobodies or PBS for 60 min. After 96 hr of infection, SARS-CoV-2 viral replication was measured in AEC cultures by quantitative PCR, with triplicate assays of harvested RNA from each SARS-CoV-2-infected AEC donor cell line (Genesig Coronavirus Strain 2019-nCoV Advanced PCR Kit, Primerdesign, Southampton, UK). The concentration of RNA harvested from AECs was used to normalize the qPCR data and was measured on a spectrophotometer (NanoDrop).

## rVSV/SARS-CoV-2 neutralization assays

Nanobodies were fivefold serially diluted and then incubated with rVSV/SARS-CoV-2/GFP wt$_{2E1}$ or plaque-purified selected variants for 1 hr at 37°C. The nanobody/recombinant virus mixture was then added to 293T/ACE2.cl22 cells. After 16 hr, cells were harvested, and GFP-positive cells quantified by flow cytometry. The percentage of GFP-positive cells was normalized to that derived from cells infected with rVSV/SARS-CoV-2 in the absence of nanobodies. The half-maximal inhibitory concentrations (IC50) for the nanobodies were determined using four-parameter nonlinear regression (least-squares regression method without weighting) (GraphPad Prism).

## Sequence analyses

To identify putative nanobody resistance mutations, RNA was isolated from aliquots of supernatant containing selected viral populations or individual plaque purified variants using NucleoSpin 96 Virus Core Kit (Macherey-Nagel). The purified RNA was subjected to reverse transcription using random hexamer primers and SuperScript VILO cDNA Synthesis Kit (Thermo Fisher Scientific). The cDNA was amplified using KOD Xtreme Hot Start DNA 396 Polymerase (MilliporeSigma) flanking the spike encoding sequences. The PCR products were gel-purified and sequenced using Sanger sequencing.

## Selection of virus variants in the presence of nanobodies

For selection of spike variants that were resistant to nanobodies, rVSV/SARS-CoV-2/GFP wt$_{2E1}$ was passaged to generate diversity and populations containing $10^6$ infectious particles were used. The rVSV/SARS-CoV-2/GFP wt$_{2E1}$ populations were incubated with dilutions of nanobodies (10× to 100× the IC50 excess) for 1 hr at 37°C. Then, the virus-nanobody mixtures were incubated with $5 \times 10^5$ 293T/ACE2.22 cells in six-well plates. Two days later, the cells were imaged and supernatant were harvested from cultures that showed evidence of viral replication (GFP-positive foci) or large numbers of GFP-positive cells. A 100 µl of the cleared supernatant was incubated with the same dilution of

nanobodies and then used to infect $5 \times 10^5$ 293T/ACE2.22 cells in six-well plates, as before. rVSV/SARS-CoV-2/GFP wt$_{2E1}$ were passaged in the present combination of nanobodies two times before complete escape was evaluated.

To isolate individual mutant viruses, selected rVSV/SARS-CoV-2/GFP wt$_{2E1}$ populations were serially diluted in medium without nanobodies and individual viral variants isolated by visualizing single GFP-positive plaques at limiting dilutions in 96-well plates containing $1 \times 10^4$ 293T/ACE2.22 cells. These plaque-purified viruses were expanded and further characterized using sequencing and nanobody neutralization assays.

## Cross-linking mass spectrometry

Nanobodies and antigens were incubated together at a 2× molar excess of nanobody at RT for 10 min in 20 mM HEPES pH 7.4 and 150 mM NaCl. Cross-linker was then added to a final concentration of 5 mM bissulfosuccinimidyl suberate (BS3) or 1 mM disuccinimidyl suberate (DSS), and samples were cross-linked for 30 min (RBD, NTD) or 18 min (ectodomain trimer) at RT. Reactions were quenched, reduced, and alkylated, and run on an SDS-PAGE gel. The band corresponding to the cross-linked nanobody-antigen complex was then excised from the gel and subjected to in-gel digestion at 37°C with trypsin (Roche, 1 µg, 4 hr) or chymotrypsin (Roche, 0.5 µg, 1.5 hr).

Peptides were extracted and analyzed with a nano-LC 1200 (Thermo Fisher) with an EASYspray PepMap RSLC C18 3 µm, 100 Å, 75 µm × 15 cm column coupled to an Orbitrap Fusion Lumos Tribrid mass spectrometer (Thermo Fisher). An ABIRD (ESI Source Solutions) was used to reduce background. The Lumos was operated in a data-dependent mode, and ions were fragmented by high-energy collisional dissociation (normalized collision energy 28). Separate LC runs were used to analyze the +3 and the +4 through +7 charge states. Higher charge species were prioritized for selection for fragmentation when analyzing the 4–7 species. Orbitrap resolution was 30,000 for MS and 15,000 for MS/MS analyses. The quadrupole isolation window was 1.4, and the MS/MS used a maximum injection time of 800 ms with four microscans. Data were then searched by pLink 2.3 (*Chen et al., 2019*) to identify cross-linked peptides. The mass accuracy in pLink was set to 10 ppm for MS and 20 ppm for MS/MS. Cysteine carbamidomethylation was included as a fixed modification and methionine oxidation as a variable modification. For trypsin, up to three missed cleavages were permitted. For chymotrypsin, the enzyme setting was 'nonspecific.' Spectra were manually checked to ensure correct identifications of cross-linked peptides.

## Acknowledgements

We are very grateful to Sonya Paske and the staff at Capralogics, Inc for raising antisera; the High-Throughput and Spectroscopy Resource Center (HTSRC), Rockefeller University, and especially Lavoisier Ramos-Espiritu for technical and data analysis support; Brenda Watt from Refeyn Ltd for technical and data analysis support; Jason Wendler and Maxwell Neal for help with statistical analyses; Thomas Gallagher and Enya Qing, Loyola University Chicago, for the kind gift of the SARS-CoV-1 and SARS-CoV-2 spike expression plasmids; Andrew McGuire and Leonidas Stamatatos (Fred Hutchinson Cancer Research Center) for their kind gift of the SARS-CoV-2 20H/501Y.V2 spike variant expression plasmid; Rhea Coler (Seattle Children's Research Institute) for the kind gift of the SARS-CoV-2 delta variant and the TMPRSS2+ Vero E6 cells; and the rest of the Aitchison, Chait and Rout laboratories, as well as Fred Cross for technical and intellectual support.

## Additional information

### Competing interests

Fred D Mast, Peter C Fridy, Natalia E Ketaren, Junjie Wang, Erica Y Jacobs, Jean Paul Olivier, John D Aitchison, Brian T Chait, Michael P Rout: Inventor on a provisional patent describing the anti-spike nanobodies described in this manuscript. Louis Herlands: Louis Herlands is affiliated with AbOde Therapeutics Inc. The author has no financial interests to declare. The other authors declare that no competing interests exist.

## Funding

| Funder | Grant reference number | Author |
|---|---|---|
| Mathers Foundation | | John D Aitchison<br>Brian T Chait<br>Michael P Rout |
| Robertson Foundation | Robertson Therapeutic Development Fund | John D Aitchison<br>Brian T Chait<br>Michael P Rout |
| Jain Foundation | | Brian T Chait<br>Michael P Rout |
| National Institutes of Health | P41GM109824 | Andrej Sali<br>John D Aitchison<br>Brian T Chait<br>Michael P Rout |
| National Institutes of Health | R01AI0501111 | Paul D Bieniasz |
| National Institutes of Health | R01AI078788 | Theodora Hatziioannou |
| National Institutes of Health | U19AI125378 | Jason S Debley |
| National Institutes of Health | K24AI150991 | Jason S Debley |
| National Institutes of Health | R01GM112108 | Andrej Sali<br>John D Aitchison<br>Michael P Rout |

The funders had no role in study design, data collection and interpretation, or the decision to submit the work for publication.

## Author contributions

Fred D Mast, Junjie Wang, Conceptualization, Data curation, Formal analysis, Investigation, Methodology, Software, Validation, Visualization, Writing – original draft, Writing – review and editing; Peter C Fridy, Natalia E Ketaren, Erica Y Jacobs, Jean Paul Olivier, Conceptualization, Data curation, Formal analysis, Investigation, Methodology, Validation, Visualization, Writing – original draft, Writing – review and editing; Tanmoy Sanyal, Investigation, Methodology, Software, Visualization, Writing – original draft, Writing – review and editing; Kelly R Molloy, Fabian Schmidt, Magdalena Rutkowska, Yiska Weisblum, Lucille M Rich, Elizabeth R Vanderwall, Paul Dominic B Olinares, Investigation, Methodology; Nicholas Dambrauskas, Vladimir Vigdorovich, Methodology, Resources; Sarah Keegan, Methodology, Software; Jacob B Jiler, Milana E Stein, Investigation; Louis Herlands, Conceptualization, Writing – review and editing; Theodora Hatziioannou, Funding acquisition, Methodology, Supervision, Writing – review and editing; D Noah Sather, Funding acquisition, Methodology, Resources, Supervision; Jason S Debley, Paul D Bieniasz, Funding acquisition, Methodology, Resources, Supervision, Writing – review and editing; David Fenyö, Software, Supervision; Andrej Sali, Funding acquisition, Methodology, Resources, Software, Supervision; John D Aitchison, Brian T Chait, Michael P Rout, Conceptualization, Funding acquisition, Methodology, Project administration, Resources, Supervision, Visualization, Writing – original draft, Writing – review and editing

## Author ORCIDs

Fred D Mast ⓘ http://orcid.org/0000-0002-2177-6647
Peter C Fridy ⓘ http://orcid.org/0000-0002-8208-9154
Natalia E Ketaren ⓘ http://orcid.org/0000-0002-7869-6162
Erica Y Jacobs ⓘ http://orcid.org/0000-0002-5018-0711
Jean Paul Olivier ⓘ http://orcid.org/0000-0002-2197-271X
Tanmoy Sanyal ⓘ http://orcid.org/0000-0002-6009-9431
Yiska Weisblum ⓘ http://orcid.org/0000-0002-9249-1745
Vladimir Vigdorovich ⓘ http://orcid.org/0000-0003-4195-4858
Jacob B Jiler ⓘ http://orcid.org/0000-0002-4566-6098
Paul Dominic B Olinares ⓘ http://orcid.org/0000-0002-3429-6618

D Noah Sather http://orcid.org/0000-0002-9128-172X
David Fenyö http://orcid.org/0000-0001-5049-3825
Andrej Sali http://orcid.org/0000-0003-0435-6197
Paul D Bieniasz http://orcid.org/0000-0002-2368-3719
John D Aitchison http://orcid.org/0000-0002-9153-6497
Brian T Chait http://orcid.org/0000-0003-3524-557X
Michael P Rout http://orcid.org/0000-0003-2010-706X

## Ethics

Human subjects: Bronchial airway epithelial cells were obtained under study #12490 approved by the Seattle Children's Institutional Review Board, with investigations carried out following the rules of the Declaration of Helsinki of 1975.

## Decision letter and Author response

Decision letter https://doi.org/10.7554/eLife.73027.sa1
Author response https://doi.org/10.7554/eLife.73027.sa2

## Additional files

### Supplementary files
• Transparent reporting form

### Data availability
The data generated or analyzed during this study are included in the manuscript.

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

# Appendix 1

## Appendix 1—key resources table

| Reagent type (species) or resource | Designation | Source or reference | Identifiers | Additional information |
|---|---|---|---|---|
| Strain, strain background (*Escherichia coli*) | ArticExpress(DE3) | Agilent | Cat# 230192 | Competent cells, enabling efficient high-level expression of heterologous proteins. |
| Strain, strain background (vesicular stomatitis virus) | rVSV/SARS-CoV-2/ GFP; WT$_{2E1}$ | *Schmidt et al., 2020* | | Recombinant chimeric VSV/SARS-CoV-2 reporter virus. Inquiries should be addressed to P. Bieniasz. |
| Strain, strain background (vesicular stomatitis virus) | 2E1-Y369N | This study | | Mutant rVSV/SARS-CoV-2/GFP derivative. Inquiries should be addressed to P. Bieniasz. |
| Strain, strain background (vesicular stomatitis virus) | 2E1-G404E | This study | | Mutant rVSV/SARS-CoV-2/GFP derivative. Inquiries should be addressed to P. Bieniasz. |
| Strain, strain background (vesicular stomatitis virus) | 2E1-D574N, E484K, Q493K | This study | | Mutant rVSV/SARS-CoV-2/GFP derivative. Inquiries should be addressed to P. Bieniasz. |
| Strain, strain background (vesicular stomatitis virus) | 2E1-S371P, H66R, N969T | This study | | Mutant rVSV/SARS-CoV-2/GFP derivative. Inquiries should be addressed to P. Bieniasz. |
| Strain, strain background (vesicular stomatitis virus) | 2E1-F490S, E484K, Q493K | This study | | Mutant rVSV/SARS-CoV-2/GFP derivative. Inquiries should be addressed to P. Bieniasz. |
| Strain, strain background (vesicular stomatitis virus) | 2E1-Q493R, G252R | This study | | Mutant rVSV/SARS-CoV-2/GFP derivative. Inquiries should be addressed to P. Bieniasz. |
| Strain, strain background (vesicular stomatitis virus) | 2E1-W64R, L452F | This study | | Mutant rVSV/SARS-CoV-2/GFP derivative. Inquiries should be addressed to P. Bieniasz. |
| Strain, strain background (vesicular stomatitis virus) | 2E1-H245R, H1083Y | This study | | Mutant rVSV/SARS-CoV-2/GFP derivative. Inquiries should be addressed to P. Bieniasz. |
| Strain, strain background (vesicular stomatitis virus) | 2E1-W64R, F490L, I931G | This study | | Mutant rVSV/SARS-CoV-2/GFP derivative. Inquiries should be addressed to P. Bieniasz. |
| Strain, strain background (vesicular stomatitis virus) | 2E1-W64R, F490S | This study | | Mutant rVSV/SARS-CoV-2/GFP derivative. Inquiries should be addressed to P. Bieniasz. |
| Strain, strain background (vesicular stomatitis virus) | 2E1-Y449H, F490S, Q787R | This study | | Mutant rVSV/SARS-CoV-2/GFP derivative. Inquiries should be addressed to P. Bieniasz. |
| Strain, strain background (vesicular stomatitis virus) | 2E1-S494P | This study | | Mutant rVSV/SARS-CoV-2/GFP derivative. Inquiries should be addressed to P. Bieniasz. |
| Strain, strain background (vesicular stomatitis virus) | 2E1-S172G | This study | | Mutant rVSV/SARS-CoV-2/GFP derivative. Inquiries should be addressed to P. Bieniasz. |
| Strain, strain background (vesicular stomatitis virus) | 2E1-E484K | *Schmidt et al., 2020* | | Mutant rVSV/SARS-CoV-2/GFP derivative. Inquiries should be addressed to P. Bieniasz. |
| Strain, strain background (vesicular stomatitis virus) | 2E1-W64R, S982R | This study | | Mutant rVSV/SARS-CoV-2/GFP derivative. Inquiries should be addressed to P. Bieniasz. |
| Strain, strain background (vesicular stomatitis virus) | 2E1-T259K, K378Q | This study | | Mutant rVSV/SARS-CoV-2/GFP derivative. Inquiries should be addressed to P. Bieniasz. |
| Strain, strain background (vesicular stomatitis virus) | 2E1-W64R, K378Q | This study | | Mutant rVSV/SARS-CoV-2/GFP derivative. Inquiries should be addressed to P. Bieniasz. |
| Strain, strain background (vesicular stomatitis virus) | 2E1-F486S | This study | | Mutant rVSV/SARS-CoV-2/GFP derivative. Inquiries should be addressed to P. Bieniasz. |

*Appendix 1 Continued on next page*

*Appendix 1 Continued*

| Reagent type (species) or resource | Designation | Source or reference | Identifiers | Additional information |
|---|---|---|---|---|
| Strain, strain background (vesicular stomatitis virus) | 2E1-T478R | This study | | Mutant rVSV/SARS-CoV-2/GFP derivative. Inquiries should be addressed to P. Bieniasz. |
| Strain, strain background (vesicular stomatitis virus) | 2E1-T478I | This study | | Mutant rVSV/SARS-CoV-2/GFP derivative. Inquiries should be addressed to P. Bieniasz. |
| Strain, strain background (vesicular stomatitis virus) | 2E1-Y508H | This study | | Mutant rVSV/SARS-CoV-2/GFP derivative. Inquiries should be addressed to P. Bieniasz. |
| Strain, strain background (vesicular stomatitis virus) | 2E1-N354S | This study | | Mutant rVSV/SARS-CoV-2/GFP derivative. Inquiries should be addressed to P. Bieniasz. |
| Strain, strain background (vesicular stomatitis virus) | 2E1-F486L | This study | | Mutant rVSV/SARS-CoV-2/GFP derivative. Inquiries should be addressed to P. Bieniasz. |
| Strain, strain background (vesicular stomatitis virus) | 2E1-Y489H | This study | | Mutant rVSV/SARS-CoV-2/GFP derivative. Inquiries should be addressed to P. Bieniasz. |
| Strain, strain background (vesicular stomatitis virus) | 2E1-K378Q | This study | | Mutant rVSV/SARS-CoV-2/GFP derivative. Inquiries should be addressed to P. Bieniasz. |
| Strain, strain background (vesicular stomatitis virus) | 2E1-L452R | This study | | Mutant rVSV/SARS-CoV-2/GFP derivative. Inquiries should be addressed to P. Bieniasz. |
| Strain, strain background (vesicular stomatitis virus) | 2E1-H245R, S349P, H1083Y | This study | | Mutant rVSV/SARS-CoV-2/GFP derivative. Inquiries should be addressed to P. Bieniasz. |
| Strain, strain background (vesicular stomatitis virus) | 2E1-P384Q | This study | | Mutant rVSV/SARS-CoV-2/GFP derivative. Inquiries should be addressed to P. Bieniasz. |
| Strain, strain background (vesicular stomatitis virus) | 2E1-E484G | This study | | Mutant rVSV/SARS-CoV-2/GFP derivative. Inquiries should be addressed to P. Bieniasz. |
| Strain, strain background (vesicular stomatitis virus) | 2E1-W64R, Y170H, V705M | This study | | Mutant rVSV/SARS-CoV-2/GFP derivative. Inquiries should be addressed to P. Bieniasz. |
| Strain, strain background (vesicular stomatitis virus) | 2E1-W64R, Y170H, Q787H | This study | | Mutant rVSV/SARS-CoV-2/GFP derivative. Inquiries should be addressed to P. Bieniasz. |
| Strain, strain background (vesicular stomatitis virus) | 2E1-T315I | This study | | Mutant rVSV/SARS-CoV-2/GFP derivative. Inquiries should be addressed to P. Bieniasz. |
| Strain, strain background (betacoronavirus) | SARS-CoV-2, Isolate USA-WA1/2020 | BEI Resources | NR-52281 | Wild-type SARS-CoV-2. |
| Strain, strain background (betacoronavirus) | SARS-CoV-2, Isolate USA-WA (B.1.617.2) | R.Coler | Delta | SARS-CoV-2 delta variant of concern. |
| Biological sample (*Lama glama*) | Bone marrow aspirates | Capralogics | | From two male llamas immunized with SARS-CoV-2 spike S1, RBD, and S2. |
| Biological sample (*L. glama*) | Sera | Capralogics | | From two male llamas immunized with SARS-CoV-2 spike S1, RBD, and S2. |
| Cell line (*Homo sapiens*) | 293T/17 | ATCC | CRL-11268 | Human kidney epithelial cells. |
| Cell line (*H. sapiens*) | 293/ACE2cl.22 | *Schmidt et al., 2020* | | 293T cells expressing human ACE2 (single-cell clone). |
| Cell line (*H. sapiens*) | 293T-ACE2 | *Cawford et al., 2020* | BEI NR-52511 | 293T cells expressing human ACE2 (single-cell clone). |
| Cell line (*H. sapiens*) | Primary human airway epithelial cells | This study | | Air-liquid interface culture system. Inquiries should be addressed to J. Debley. |

*Appendix 1 Continued on next page*

*Appendix 1 Continued*

| Reagent type (species) or resource | Designation | Source or reference | Identifiers | Additional information |
|---|---|---|---|---|
| Cell line (*Cercopithecus aethiops*) | VERO C1008 [Vero 76, clone E6, Vero E6] | ATCC | CRL-1586 | Monkey kidney epithelial cells. |
| Cell line (*C. aethiops*) | TMPRSS2+ Vero E6 | R.Coler | | Vero E6 cells expressing human TMPRSS2. |
| Antibody | Anti-COVID-19 and SARS-CoV S glycoprotein [CR3022] (human monoclonal) | Absolute Antibody | Cat# Ab01680-10.0 | Flow cytometry (1:1000). |
| Antibody | Anti-human IgG (H+L) Cross-Adsorbed Secondary Antibody, Alexa Fluor 488 (goat polyclonal) | Invitrogen | Cat# A-11013 | Flow cytometry (1:2000). |
| Recombinant DNA reagent | pET21-pelB | *Fridy et al., 2014a* | | Expression plasmid for expressing nanobodies. |
| Recombinant DNA reagent | pET21-pelB-SARS-CoV-2 Nanobody | This study; *Fridy et al., 2014a* | See *Table 1* and *Table 2* | SARS-CoV-2 nanobody expression plasmids. Inquiries should be addressed to M. Rout. |
| Recombinant DNA reagent | pcDNA3.1+ SARS-1-S-C9 | T.Gallagher | | SARS-CoV-1 spike expression plasmid. |
| Recombinant DNA reagent | pcDNA3.1+ SARS-2-S-C9 WUHAN-1 | T.Gallagher | | SARS-CoV-2 spike expression plasmid. |
| Recombinant DNA reagent | pcDNA3.1+ SARS-2-B.1.1.7 | NIAID | | SARS-CoV-2 spike expression plasmid for alpha variant. |
| Recombinant DNA reagent | pHDM-SARS-CoV-2-Spike-B.1.351 | L.Stamatatos | | SARS-CoV-2 spike expression plasmid for beta variant. |
| Recombinant DNA reagent | pcDNA3.1+ SARS-2 P.1 | NIAID | | SARS-CoV-2 spike expression plasmid for gamma variant. |
| Recombinant DNA reagent | psPAX2 | D.Trono/ Addgene | Plasmid #12260 | Second-generation lentiviral packaging plasmid. |
| Recombinant DNA reagent | pHAGE-CMV-Luc2-IRES-ZsGreen-W | J.Bloom / *Cawford et al., 2020* | NR-52516 | Lentiviral backbone plasmid that uses a CMV promoter to express luciferase followed by an IRES and ZsGreen. |
| Sequence-based reagent (primer) | 6N_CALL001 | This study | PCR and sequencing primer | NNNNNNGTCCTGGCTGCTCTTCTACAAGG |
| Sequence-based reagent (primer) | 6N_CALL001B | This study | PCR and sequencing primer | NNNNNNGTCCTGGCTGCTCTTTTACAAGG |
| Sequence-based reagent (primer) | 6N_VHH_SH_rev | This study | PCR and sequencing primer | NNNNNNCTGGGGTCTTCGCTGTGGTGC |
| Sequence-based reagent (primer) | 6N_VHH_LH_rev | This study | PCR and sequencing primer | NNNNNNGTGGTTGTGGTTTTGGTGTCTTGGG |
| Peptide, recombinant protein | Spike S1 (Wuhan Str.) | Sino Biological | Cat# 40591-V08H | For determining $K_D$s. |
| Peptide, recombinant protein | Spike RBD | Sino Biological | Cat# 40592-VNAH | For determining $K_D$s. |
| Peptide, recombinant protein | SARS-CoV-2 (2019-nCoV) Spike RBD-mFc Recombinant Protein | Sino Biological | Cat# 40592-V05H | For epitope mapping. |
| Peptide, recombinant protein | Spike S2 | Sino Biological | Cat# 40590-V08B | For immunization and for determining $K_D$s. |
| Peptide, recombinant protein | Spike S1 (501Y.V1) | Sino Biological | Cat# 40591-V08H12 | For determining $K_D$s. |
| Peptide, recombinant protein | Spike S1 (501Y.V2) | Sino Biological | Cat# 40591-V08H10 | For determining $K_D$s. |
| Peptide, recombinant protein | SARS-CoV-2 Spike S1, Sheep Fc-Tag | The Native Antigen Co. | Cat# REC31806 | For immunization. |

*Appendix 1 Continued on next page*

*Appendix 1 Continued*

| Reagent type (species) or resource | Designation | Source or reference | Identifiers | Additional information |
| --- | --- | --- | --- | --- |
| Peptide, recombinant protein | SARS-CoV-2 Spike S2, Sheep Fc-Tag | The Native Antigen Co. | Cat# REC31807 | For immunization. |
| Peptide, recombinant protein | Protein M | *Grover et al., 2014* | | Used to deplete light-chain containing IgGs. |
| Peptide, recombinant protein | Thyroglobulin | Sigma-Aldrich | Cat# A8531-1V | Used to calibrate mass photometer. |
| Peptide, recombinant protein | Bovine serum albumin | Sigma-Aldrich | Cat# T9145-1VL | Used to calibrate mass photometer. |
| Peptide, recombinant protein | Beta-amylase | Sigma-Aldrich | Cat# A8781-1VL | Used to calibrate mass photometer. |
| Peptide, recombinant protein | FabRICATOR (IdeS) | Genovis | Cat# A0-FR1-050 | Protease to cleave $V_HH$ domain from the HCAb. |
| Commercial assay, kit | ProteOn Amine Coupling Kit | Bio-Rad | Cat# 1762410 | Used to couple $V_HH$ domain to beads. |
| Commercial assay, kit | TruSeq Nano DNA Low Throughput Library Prep Kit | Illumina | Cat# 20015964 | Used to sequence $V_HH$. |
| Commercial assay, kit | Steady-GLO | Promega | Cat# E2520 | Used in pseudovirus assay. |
| Software, algorithm | Llama-Magic | *Fridy et al., 2014a*, | https://github.com/FenyoLab/llama-magic | For identifying nanobody sequences. |
| Software, algorithm | IMP, the Integrative Modeling Platform | *Russel et al., 2012* | https://integrativemodeling.org | For integrative structural modeling. |
| Software, algorithm | UCSF ChimeraX | *Pettersen et al., 2021* | https://www.rbvi.ucsf.edu/chimerax/download.html | For visualizing structural models. |
| Software, algorithm | synergy v0.4 | *Wooten et al., 2021* | https://pypi.org/project/synergy/ | For observing synergy. |
| Software, algorithm | matplotlib v3.4.1 | *Hunter, 2007* | https://pypi.org/project/matplotlib/ | For preparing figures. |
| Software, algorithm | seaborn v0.11.0 | *Waskom, 2021* | https://seaborn.pydata.org/index.html | For preparing figures. |
| Software, algorithm | plotly v4.12.0 | 2019 Plotly, Inc | https://plotly.com/python/ | For preparing figures. |
| Software, algorithm | numpy v1.19.2 | *Huo et al., 2020a* | https://numpy.org | For data analysis. |
| Software, algorithm | pandas v1.1.2 | *The pandas development team, 2020* | https://pandas.pydata.org | For data analysis. |
| Software, algorithm | scipy v1.5.0 | *Virtanen et al., 2020* | https://www.scipy.org | For data analysis. |
| Software, algorithm | scikit-learn v0.23.2 | *Pedregosa, 2011* | https://scikit-learn.org/stable/ | For data analysis. |
| Software, algorithm | python v3.8 | *Van Rossum and Drake, 2009* | https://www.python.org | For data analysis and preparing figures. |
| Software, algorithm | Prism 9 | GraphPad | https://www.graphpad.com | For data analysis and preparing figures. |
| Software, algorithm | OCTET | Sartorius | | For data analysis. |
| Other | ProteOn GLC Sensor Chip | Bio-Rad | Cat# 176-5011 | For protein interaction analysis. |
| Other | Series S Sensor Chip CM5 | Cytiva | Cat# BR100530 | For protein interaction analysis. |
| Other | Hard-shell PCR plates, 96-well, thin-wall | Bio-Rad | Cat# HSP9661 | For differential scanning fluorimetry. |
| Other | Microseal 'B' seal | Bio-Rad | Cat# MSB1001 | For differential scanning fluorimetry. |
| Other | Precision Coverslips | Thorlabs | Cat# CG15KH1 | For mass photometry. |

*Appendix 1 Continued on next page*

*Appendix 1 Continued*

| Reagent type (species) or resource | Designation | Source or reference | Identifiers | Additional information |
|---|---|---|---|---|
| Other | CultureWell Reusable Gasket | Grace Bio-Labs | Cat# 103250 | For mass photometry. |
| Other | Protein A Sepharose 4B | Thermo Fisher | Cat# 101042 | For protein purification. |
| Other | Recombinant Protein G Sepharose 4B | Thermo Fisher | Cat# 101243 | For protein purification. |
| Other | CNBr-activated Sepharose 4 Fast Flow | Cytiva | Cat# 17098101 | For nanobody screening. |
| Other | SuperScript VILO Master Mix | Thermo Fisher | Cat# 11755250 | For viral escape analysis. |
| Other | Anti-mouse IgG Fc Capture Biosensors | Sartorius | Cat# 18-5088 | For epitope mapping. |
| Other | KOD Xtreme Hot Start DNA Polymerase | Sigma-Aldrich | Cat# 71975 | For viral escape analysis. |

