## [Editor Report]

The paper describes an impressive collection of hundreds of new nanobodies binding SARS-CoV-2 spike by combining in vivo antibody affinity maturation and proteomics. It provides a comprehensive characterization of a repertoire of the spike nanobodies and their combinations by complementary biophysical, structural modeling, and functional assays. It also identifies non-receptor binding domain nanobodies, includes extensive bioengineering to substantially improve potency and resistance to escaping variants, and demonstrates synergistic activities using nanobody cocktails. This work thus provides significant impacts on SARS-CoV-2 research and therapeutics.

---

## [Decision Letter]

**Decision letter after peer review:**

Thank you for submitting your article "Highly synergistic combinations of nanobodies that target SARS-CoV-2 and are resistant to escape" for consideration by *eLife*. Your article has been reviewed by 3 peer reviewers, including Wai-Hong Tham as the Reviewing Editor and Reviewer #1, and the evaluation has been overseen by Miles Davenport as the Senior Editor.

Essential revisions:

1) To fully benefit from the publication of this latest collection of nanobodies, the authors should publish all the sequences and have to make the best efforts to provide comparisons with existing nanobodies described in the literature as requested by the reviewers.

2) The authors benchmarked their neutralization assay using several previously reported nanobodies. However, their identities are missing. It would be useful to provide some insight into the consistency and variations among different assays with published literature.

*Reviewer #1 (Recommendations for the authors):*

1. Page 6, Lines 1-4: Could the authors please describe how nanobody concentration in periplasmic extracts was normalised when assessing binding to immobilized antigen? If no normalisation took place in this assay, could the amount of nanobody bound vary considerably due to yield rather than affinity?

2. Page 14, Line 12: Could the authors please include a table of neutralisation values for SARS-CoV-2 live virus neutralisation that corresponds to the data shown in figure 5A.

3. Page 14, Line 20-23: In the ALI neutralisation model, cells were incubated with serial dilutions of nanobodies both prior to and subsequent to viral exposure. This therefore does not "mimic a treatment regimen" which would necessarily only begin once infection has occurred. Please could the authors rephrase this sentence.

4. Page 14, Lines 26-28: The references given here do not contain similar ALI experiments to those described in the manuscript and yet we are asked to compare neutralisation values in this assay to those in these papers. Given that the authors earlier (Page 12, Lines 19-21) state that neutralisation values cannot be compared between labs or assays, could the authors justify/remove this comparison, or alternatively include experimental data for benchmark antibodies in this assay?

5. Page 16, Lines 20-33: The identification of nanobody binding epitopes through the IMP modelling is relied upon in this manuscript to identify nanobodies with unique binding sites, and to identify nanobodies that may be resistant to mutations present in VOC. Particularly for the Delta variant, there are no other assays carried out here to validate nanobodies identified as likely to be resistant to the Delta strain RBD mutations.

Given the reliance upon this data, it would be extremely useful to include a nanobody which has been structurally characterised as a control in the IMP approach, to determine whether an accurate binding site is reproduced. Please could the authors include this data or comment on why this would not be appropriate?

6. Page 18, Line 22: The S1-RBD-35 site seems to overlap with the epitope commonly targeted by the anti RBD IgHV 1-58 class of antibodies (Tortorici et al. 2020, Dejnirattisai et al. 2021). Could the authors please describe how the S1-RBD-35 site differs from the epitopes of this commonly elicited anti-SARS-CoV-2 antibody class?

7. Page 18, Lines 24-25:

a. The authors describe the S1-6 epitope as a unique site, however, looking at the data given in tables S6 and S7, it seems that the RBD residues important for S1-6 binding are 501 and 371. Including residues identified in table S6 for nanobodies within the same small epitope bin (Figure 3A), 369, 378 and 404, this still gives a binding site that largely overlaps that of VHH-U and VHH-V (Koenig et al. 2021) and WNb 10 (Pymm et al. 2021). Could the authors please compare the S1-6 epitope with that of these nanobodies and describe the overlap? And also include the papers in the reference list?

b. Page 18, Table S6: The authors describe the data shown in this table as being an input for epitope modelling (through IMP) for the nanobody epitopes shown in Figure 6. For nanobody S1-6, an escape mutation outside of the RBD (residue 574) is identified as a driver of escape. Please could the authors explain how this is incorporated into the IMP modelling to define a binding epitope, and how this mutation is expected impact the RBD directed nanobody?

8. Page 19, Lines 1-4: While additional neutralisation mechanisms are certainly possible, it is also possible that despite sharing a similar epitope, there are a variety of binding orientations to the RBD among these nanobodies, some of which will sterically hinder ACE2 interaction while others will not, despite the group having similar epitopes.

a. Could the authors please include this as a possible explanation here?

b. This could be tested by measuring the ability of these nanobodies to block interaction of ACE2 with the RBD. Given that stated aim in the abstract to "identify novel mechanisms of viral neutralization" could the authors explain why this has this not been carried out on the repertoire?

9. Pages 18-19, Multiple modes of RBD Binding and Neutralization: While this section contains well-reasoned, valid speculation on mechanisms of neutralisation, novel mechanisms of neutralisation are not actually identified here.

a. There are a number of experimental approaches such as receptor interaction assays and structural characterisation that could have been undertaken to determine these mechanisms. Could the authors briefly describe experimental approaches to address these outstanding questions.

b. Could the authors remove statements from the abstract that claim identification of novel mechanisms of neutralisation in the manuscript.

c. The section also discusses nanobodies directed against S1 non-RBD epitopes and S2, could the title please be adapted to reflect this.

10. Page 23, Lines 8-14: This is an extremely interesting section, and highlights differences between nanobody and antibody binding. However, nanobodies are cleared rapidly in vivo, limiting their use as therapeutics without modification to increase half-life. This often involves fusion with an Fc domain that results in a dimeric, antibody-like construct.

In light of this could the authors please comment here on how these findings could remain applicable in a therapeutic setting? Has similar synergy been demonstrated for nanobody-Fc fusion constructs for these nanobodies or elsewhere?

*Reviewer #2 (Recommendations for the authors):*

– As the data of this manuscript is substantial, I would mostly recommend to adjust the text to acknowledge other work on SARS-CoV-2 nanobodies that is not discussed much in the current version of the manuscript. It would be desirable, however, if at least some data on the now dominating Delta variant was included.

Exemplary suggestions for the text of the manuscript:

– Page 4, line 6:

'To date, there are a limited number of nanobodies available and those that are available recognize regions of RBD that are subject to escape variation (Niu et al., 2021)'

There are at least 187 if not many more neutralizing SARS-CoV-2 nanobodies described (http://opig.stats.ox.ac.uk/webapps/covabdab/). Although this is a 'limited' number, this statement is misleading as the number is not small. While escape mutations can arise in any of the targeted epitopes, the relevant emerging variants do not exhibit mutations in the epitope bound e.g. by VHH72 (Wrapp et al.).

– This is not the first work that defines epitopes on the RBD or spike of SARS-CoV-2, although the level of detail provided is impressive. It might still be helpful to compare the defined epitopes and classes to those identified elsewhere, e.g. Sun et al. (Nature Communication 12, 2021) or Koenig et al. (Science 371, 2021). Throughout the manuscript (and in particular in figure 6), it may be helpful to mention which (structurally) defined nanobodies fall into the classes the authors define. The authors included some published nanobodies into their functional characterization (Xiang et al., Wrapp et al., 2020), but do not specify the nanobodies in their data set, and do not comment on whether the obtained values agreed with the previous description – this information should be included/discussed. In case the same nanobodies were used in any of the other systematic assessments (e.g. epitope binning in figure 3A), it would be helpful to also include this data.

– Page 12, line 40:

The authors cite their previous work (Fridy et al., 2014) for enhanced binding of multimeric nanobodies. Perhaps it would be fair to mention that this concept has also been successfully applied to SARS-CoV-2 neutralizing nanobodies (Schoof et. al, Xiang et al., Koenig et al.).

– Page 14, line 41:

The authors correctly cite work of one of the authors (Weisblum et al., 2020) for their in vitro evolution studies, but it would perhaps also be fair to reference the first study that undertook such evolution experiments with a similar chimeric virus (Baum et al., Science 369, 2020).

– Page 19, line 4

'Additionally, the binding of nanobodies in Groups 1, 3, 4, and 6 may mimic ACE2 binding, thus trapping the RBD in its "up" position to either catalyze the spike trimer rearrangements that prematurely convert spike into a post-fusion state, suppressing viral fusion (Cai et al., 2020), or destabilizing the trimer to cause its premature disassembly.' This mode of action of nanobodies is postulated/described in several publications. Please refer to this work as well.

– Page 23, line 1:

‘Interestingly, among all the nanobody pairs that we tested, the synergy was greatest with the S2-10-dimer, which showed >4000 fold increase in potency when combined with either S1-23 or S1-RBD-15 (Figure 7I,J).' It is not clear if the data provided in figure 7I/J refers to monomers or dimers of S2-10 (the >4000 fold increase is not apparent from the figure).

*Reviewer #3 (Recommendations for the authors):*

Figure 2B: an S2 binder (S2-47) was included in the cluster dominated by RBD nanobodies. Any idea what is going on here?

Figure 4: the authors benchmarked their neutralization assay using several previously reported nanobodies. However, their identities are missing. For example, did the authors evaluate epitope I neutralizers such as Nb20 and Nb21 reported by Xiang et al.? It would be useful to provide some insight into the consistency and variations among different assays.

Regarding nanobody nebulization (Page 3 line 42): Indeed, Nambulli et al. reported the high preclinical efficacy of a nanobody construct (PiN-21) for inhalation therapy of SARS-CoV-2 infection in a sensitive COVID-19 animal model.

The synergy between different epitope binders (e.g., RBD and S2) is intriguing. It is unfortunate that the authors have not been able to follow up by characterizing their binding to the spike by high-resolution cryoEM, which may unravel the mechanism(s) of cooperativity to substantially strengthen the paper. The authors state that the strong synergy is unique to nanobodies, but not to human IgG antibodies (Page 23 line 9), which seems to contradict some literature (see below). Moreover, since the 2D response assay has not been well adopted in antibody literature (for practical reasons, the molar ratio between two antibodies within a cocktail is fixed and stoichiometric), it would be difficult to draw a strong conclusion here.

ter Meulen J, van den Brink EN, Poon LLM, Marissen WE, Leung CSW, Cox F, et al. (2006) Human Monoclonal Antibody Combination against SARS Coronavirus: Synergy and Coverage of Escape Mutants. PLoS Med 3(7): e237.

Page 23 lines 23-24 (perspective and discussion): it would be useful if the authors could provide some thoughts to suggest potential combinatory/ synergistic examples as lead candidates for future preclinical development.

Page 26 line 10: complementary digestion with trypsin and chymotrypsin (Xiang et al. 2020). It seems that the paper was incorrectly cited (Xiang, Y., et al. (2021). Integrative proteomics identifies thousands of distinct, multi-epitope, and high-affinity nanobodies. Cell Syst 12, 220-234 e229.)

---

## [Author Response]

Essential revisions:1) To fully benefit from the publication of this latest collection of nanobodies, the authors should publish all the sequences and have to make the best efforts to provide comparisons with existing nanobodies described in the literature as requested by the reviewers.

We provide the sequences for all nanobodies in the repertoire as source data for Figure 1. Comparisons to existing nanobodies have been included in the revised text as requested.

2) The authors benchmarked their neutralization assay using several previously reported nanobodies. However, their identities are missing. It would be useful to provide some insight into the consistency and variations among different assays with published literature.

We have now identified these nanobodies in the plot of Figure 4 by the numbers 1, 2, 3, and 4, which are detailed explicitly in the figure legend using their original names and the references in which they were described. Instead of comparing IC50s between labs, we used other labs’ nanobodies as benchmarks. This avoids confusion and disputes that could otherwise arise. We emphasize that we made these comparisons because as we stated in the manuscript, “measured IC50s are dependent on assay conditions and so cannot be readily compared across laboratories (Cheng and Prusoff, 1973)”, hence, we can only compare values for our nanobodies vs benchmark nanobodies when they are measured in the same assay, performed in parallel (as we have done here).

To test the reliability of our neutralization assay (and so the fairness of the comparison with these benchmarks), we have also used two different, orthogonal neutralization assays, performed by different investigators in different labs (Aitchison Lab, Seattle and Bieniasz Lab, RU) to characterize our nanobodies, as shown now in Figure 4—figure supplement 1. Note the high degree of correlation and significance, underscoring the consistency of the values that we obtained for our nanobodies. We are, to our knowledge, the only group to have performed such broad orthogonal verifications. Therefore, we can with confidence compare the neutralization values we obtain for nanobodies from other groups with our nanobodies as measured by our neutralization assay(s).

Reviewer #1 (Recommendations for the authors):1. Page 6, Lines 1-4: Could the authors please describe how nanobody concentration in extracts was normalised when assessing binding to immobilized antigen? If no periplasmic normalisation took place in this assay, could the amount of nanobody bound vary considerably due to yield rather than affinity?

While the nanobody concentration in periplasmic extracts was not normalized and does indeed vary due to expression level, a large excess of periplasmic input (and so nanobody) was used in these experiments. Thus the yield of bound nanobody would only be significantly affected by very modestly expressed candidates, which we aimed to eliminate in this screen along with non-binders. This has been clarified in the text.

2. Page 14, Line 12: Could the authors please include a table of neutralisation values for SARS-CoV-2 live virus neutralisation that corresponds to the data shown in figure 5A.

A new Table 6 as now been added containing those values.

3. Page 14, Line 20-23: In the ALI neutralisation model, cells were incubated with serial dilutions of nanobodies both prior to and subsequent to viral exposure. This therefore does not "mimic a treatment regimen" which would necessarily only begin once infection has occurred. Please could the authors rephrase this sentence.

The sentence has been rephrased:

“AlI cultures were then treated with nanobodies at 24 h intervals for an additional 3 days before harvesting the cells, extracting RNA, and measuring SARS-CoV-2 levels by qPCR (Figure 5B).”

4. Page 14, Lines 26-28: The references given here do not contain similar ALI experiments to those described in the manuscript and yet we are asked to compare neutralisation values in this assay to those in these papers. Given that the authors earlier (Page 12, Lines 19-21) state that neutralisation values cannot be compared between labs or assays, could the authors justify/remove this comparison, or alternatively include experimental data for benchmark antibodies in this assay?

Agreed, and so we have rephrased this sentence to remove the comparison:

“The efficacy of the S1-23 nanobody was strongly enhanced when provided to cells as a trimer, potently inhibiting viral replication (Figure 5C).”

5. Page 16, Lines 20-33: The identification of nanobody binding epitopes through the IMP modelling is relied upon in this manuscript to identify nanobodies with unique binding sites, and to identify nanobodies that may be resistant to mutations present in VOC. Particularly for the Delta variant, there are no other assays carried out here to validate nanobodies identified as likely to be resistant to the Delta strain RBD mutations.Given the reliance upon this data, it would be extremely useful to include a nanobody which has been structurally characterised as a control in the IMP approach, to determine whether an accurate binding site is reproduced. Please could the authors include this data or comment on why this would not be appropriate?

We relied heavily on the escape data as inputs to IMP, which is not available for other structurally characterized nanobodies; therefore, we cannot perform the comparison requested, by simply starting from a nanobody with an experimentally determined structure. However, Sun at. al. 2021 identify the RBD point mutation E484K/Q as critical to the high affinity of the nanobody Nb21, for which, they also report cryo-EM structures (in addition to other nanobodies). This provides an indirect way to benchmark our modeling protocol in IMP by using the cryo-EM derived structure of Nb21 (source from the PDB id 7N9B.D) and the residue E484 as an “effective” escape mutant for Nb21.

Figure 6—figure supplement 2 compares the IMP-modeled binding pose (after clustering 1,200,000 alternate models produced from structural sampling) and epitope of Nb21 to that in the cryo-EM structure (chain D), by aligning the RBD structure used in modeling (6m0j.E:333-526) onto the corresponding region in the down-RBD from 7N9b.A. Clearly, both the relative orientation of Nb21 as well as its epitope on the RBD (spread around E484) are in extremely good agreement with experiment, with a backbone RMSD of 2.1 Å. Spatial restraints implemented in IMP only ensured receptor-nanobody shape complementarity around the escape mutation, and did not explicitly enforce the orientation of the framework regions to the receptor. Thus, Figure 6—figure supplement 2 confirms that our computational epitope prediction model developed in IMP is reliable, when supplied with appropriate escape mutation(s) as input(s).

Further, we now include assays that provide direct evidence for the accuracy of our epitope mapping by IMP, using binding to the delta as a test of this accuracy. As predicted by the model S1-1 neutralizes delta as whereas S1-RBD-35 does not. These data are now included as Figure 6—figure supplement 3B and we have made the following additions to the text:

“Of the 10 groups into which our nanobodies were classified, 7 (Groups 1, 2, 5, and 7 target RBD; Group 8 targets the S1-NTD; Groups 9 and 10 target S2) do not overlap with the mutations that distinguish alpha, beta, gamma and delta SARS-CoV-2 VOCs (Figure 6—figure supplement 6A). […] This confirms the potency of our repertoire against key VOC and underscoring the importance of generating large, diverse nanobody repertoires as presented here.”

6. Page 18, Line 22: The S1-RBD-35 site seems to overlap with the epitope commonly targeted by the anti RBD IgHV 1-58 class of antibodies (Tortorici et al. 2020, Dejnirattisai et al. 2021). Could the authors please describe how the S1-RBD-35 site differs from the epitopes of this commonly elicited anti-SARS-CoV-2 antibody class?

We thank the reviewer for pointing this out. Indeed, IgHV 1-58 class antibodies partially overlap with the S1-RBD-35 epitope in sharing F486 and N487 of the RBM. However, while S1-RBD-35 epitope is restricted to this point on the RBD, the IgHV 1-58 (Site Ib/class 2) binding site continues down the RBM engaging spike residues 455-458. We have modified the text to now read:

“Group 6 is represented by a single nanobody (S1-RBD-35) and also binds the RBM at a site partially overlapping with Site Ib/class2 (Corti et al., 2021; Tortorici et al., 2020; Dejnirattisai et al., 2021), but is distinguished by its binding to the tip of the RBM at the left of this representation (Figure 6).”

7. Page 18, Lines 24-25:a. The authors describe the S1-6 epitope as a unique site, however, looking at the data given in tables S6 and S7, it seems that the RBD residues important for S1-6 binding are 501 and 371. Including residues identified in table S6 for nanobodies within the same small epitope bin (Figure 3A), 369, 378 and 404, this still gives a binding site that largely overlaps that of VHH-U and VHH-V (Koenig et al. 2021) and WNb 10 (Pymm et al. 2021). Could the authors please compare the S1-6 epitope with that of these nanobodies and describe the overlap? And also include the papers in the reference list?

The binding sites of VHH-U, VHH-V and WNb 10 more closely overlap with our Group 1 nanobodies, particularly S1-RBD-22 and S1-RBD-24. The S1-6 binding site is adjacent to this binding site as shown in Figure 6B of the manuscript. We have modified the discussion of our Group 1 nanobodies to note the shared binding site with VHH-U, VHH-V and WNb 10 as follows:

“Group 1 (S1-RBD-15, S1-1, S1-RBD-22, S1-RBD-24, S1-RBD-9) overlaps with Site Iia / Class 4 and partially overlaps with the opposite side of the RBM as Groups 3 and 4. This site appears to be a common nanobody epitope and is shared by VHH-U, VHH-V and WNb 10 (Koenig et al., 2021; Pymm et al., 2021).”

b. Page 18, Table S6: The authors describe the data shown in this table as being an input for epitope modelling (through IMP) for the nanobody epitopes shown in Figure 6. For nanobody S1-6, an escape mutation outside of the RBD (residue 574) is identified as a driver of escape. Please could the authors explain how this is incorporated into the IMP modelling to define a binding epitope, and how this mutation is expected impact the RBD directed nanobody?

We thank the reviewer for pointing this out. Indeed residue 574 is outside the structurally covered region of the RBD (residues 333-526) and therefore, was not used in the IMP modeling. We have added an appropriate note at the end of Table 7.

8. Page 19, Lines 1-4: While additional neutralisation mechanisms are certainly possible, it is also possible that despite sharing a similar epitope, there are a variety of binding orientations to the RBD among these nanobodies, some of which will sterically hinder ACE2 interaction while others will not, despite the group having similar epitopes.a. Could the authors please include this as a possible explanation here?

This is an excellent point and we have modified the text as follows:

“However, interestingly, several nanobodies sharing similar epitope bins as S1-RBD-9 (Group 1), such as S1-RBD-34, S1-RBD-19, S1-RBD-25, S1-RBD-32, and S1-RBD-36 do not neutralize spike (Table 1) suggesting that neutralization may occur via an additional mechanism. Alternatively, different nanobodies can engage a shared epitope with different binding orientations that may or may not hinder ACE2 binding.”

b. This could be tested by measuring the ability of these nanobodies to block interaction of ACE2 with the RBD. Given that stated aim in the abstract to "identify novel mechanisms of viral neutralization" could the authors explain why this has this not been carried out on the repertoire?

Blocking Spike’s interaction with spike – while a common mechanism – is only one of the many possible mechanisms by which these Nbs may inhibit spike. While we discuss some of these, based on the epitope positions, a comprehensive test of all these possible mechanisms (including systematically testing all our nanobodies for inhibition of Ace2 binding) represents a large body of work that is well beyond the scope of this already large manuscript and we believe more suitable for follow-up studies. Indeed, preventing interactions with ACE2 can occur through multiple mechanisms. We contend that based on the fact that nanobodies bind to novel sites on spike, this likely reflects previously unreported mechanisms of inhibition. This perspective is now reflected in the revised text.

9. Pages 18-19, Multiple modes of RBD Binding and Neutralization: While this section contains well-reasoned, valid speculation on mechanisms of neutralisation, novel mechanisms of neutralisation are not actually identified here.a. There are a number of experimental approaches such as receptor interaction assays and structural characterisation that could have been undertaken to determine these mechanisms. Could the authors briefly describe experimental approaches to address these outstanding questions.

There are many assays that could have been performed but it does not seem appropriate to describe assays not performed as part of this study. Nonetheless we do highlight a few key structural methods such as crystallization and cryoEM that are now mentioned in the manuscript as validation to IMP deduced structures. We have included the following modifications to the text:

“These models provide sufficient resolution to map the size and position of the epitopes bound by each nanobody; however, however, future higher resolution studies using cryoEM or crystallization are warranted for the highest priority nanobodies (Schoof et al., 2020, Pymm et al., 2021, Xiang et al., 2020, Wrapp et al., 2020a)”

b. Could the authors remove statements from the abstract that claim identification of novel mechanisms of neutralisation in the manuscript.

The abstract has been rephrased to reflect the fact that nanobodies binding to unique sites likely neutralize through novel mechanisms. It now reads:

“… to identify new nanobody binding sites that may reflect novel mechanisms of viral neutralization”.

c. The section also discusses nanobodies directed against S1 non-RBD epitopes and S2, could the title please be adapted to reflect this.

Header has been changed.

10. Page 23, Lines 8-14: This is an extremely interesting section, and highlights differences between nanobody and antibody binding. However, nanobodies are cleared rapidly in vivo, limiting their use as therapeutics without modification to increase half-life. This often involves fusion with an Fc domain that results in a dimeric, antibody-like construct.In light of this could the authors please comment here on how these findings could remain applicable in a therapeutic setting? Has similar synergy been demonstrated for nanobody-Fc fusion constructs for these nanobodies or elsewhere?

The tunable half-life in vivo of nanobodies is considered one of their potential advantages. Although nanobody fusions are indeed often required to tune pharmacokinetic behavior as therapeutics, many options are available for such optimization, and we have added clarification and references in the text to this effect. To our knowledge the type of synergy that we have observed has not been demonstrated in nanobody fusion constructs. The text has been changed to reflect this point.

Reviewer #2 (Recommendations for the authors):– As the data of this manuscript is substantial, I would mostly recommend to adjust the text to acknowledge other work on SARS-CoV-2 nanobodies that is not discussed much in the current version of the manuscript. It would be desirable, however, if at least some data on the now dominating Delta variant was included.

We have added citations as requested and have generated new data on the delta variant, presented in Figure 6-Figure supplement 3B.

Exemplary suggestions for the text of the manuscript:– Page 4, line 6:'To date, there are a limited number of nanobodies available and those that are available recognize regions of RBD that are subject to escape variation (Niu et al., 2021)'There are at least 187 if not many more neutralizing SARS-CoV-2 nanobodies described (http://opig.stats.ox.ac.uk/webapps/covabdab/). Although this is a 'limited' number, this statement is misleading as the number is not small. While escape mutations can arise in any of the targeted epitopes, the relevant emerging variants do not exhibit mutations in the epitope bound e.g. by VHH72 (Wrapp et al.).

We have added citations as requested and have generated new data on the delta variant, presented in Figure 6-Figure supplement 3B.

– This is not the first work that defines epitopes on the RBD or spike of SARS-CoV-2, although the level of detail provided is impressive. It might still be helpful to compare the defined epitopes and classes to those identified elsewhere, e.g. Sun et al. (Nature Communication 12, 2021) or Koenig et al. (Science 371, 2021). Throughout the manuscript (and in particular in figure 6), it may be helpful to mention which (structurally) defined nanobodies fall into the classes the authors define. The authors included some published nanobodies into their functional characterization (Xiang et al., Wrapp et al., 2020), but do not specify the nanobodies in their data set, and do not comment on whether the obtained values agreed with the previous description – this information should be included/discussed. In case the same nanobodies were used in any of the other systematic assessments (e.g. epitope binning in figure 3A), it would be helpful to also include this data.

We now include the identities and IC50 values obtained for the comparator nanobodies in the Figure 4 legend.

“From Xiang, et al., 2020: 1. Nb-21 (IC50 2.4 nM) 2. Nb-34 (IC50 5.6 nM) and 3. Nb-93 (IC50 123 nM). From Wrapp et al., 2020: 4. VHH-72 (IC50 2.5 μM).”

As mentioned in our response to Reviewer 1 we made these comparisons because as we stated in the manuscript: “measured IC50s are dependent on assay conditions and so cannot be readily compared across laboratories (Cheng and Prusoff, 1973)”, hence, we can only compare values for our nanobodies vs benchmark nanobodies when they are measured in the same assay, performed in parallel (as we have done here).

– Page 12, line 40:The authors cite their previous work (Fridy et al., 2014) for enhanced binding of multimeric nanobodies. Perhaps it would be fair to mention that this concept has also been successfully applied to SARS-CoV-2 neutralizing nanobodies (Schoof et. al, Xiang et al., Koenig et al.).

Done.

– Page 14, line 41:The authors correctly cite work of one of the authors (Weisblum et al., 2020) for their in vitro evolution studies, but it would perhaps also be fair to reference the first study that undertook such evolution experiments with a similar chimeric virus (Baum et al., Science 369, 2020).

We appreciate the Reviewer’s point but feel that in the context cited, reference to Weisblum et al., 2020 is correct. We provide citation to Baum et al., elsewhere in the text, for example:

“Nanobody cocktails are expected to be resistant to escape, as they recognize multiple epitopes (Baum et al., 2020, De Gasparo et al., 2021, Weisblum et al., 2020).”

– Page 19, line 4'Additionally, the binding of nanobodies in Groups 1, 3, 4, and 6 may mimic ACE2 binding, thus trapping the RBD in its "up" position to either catalyze the spike trimer rearrangements that prematurely convert spike into a post-fusion state, suppressing viral fusion (Cai et al., 2020), or destabilizing the trimer to cause its premature disassembly.' This mode of action of nanobodies is postulated/described in several publications. Please refer to this work as well.

We have added citations as requested. The text now reads:

“Additionally, the binding of nanobodies in Groups 1, 3, 4, and 6 may mimic ACE2 binding, thus trapping the RBD in its “up” position to either catalyze the spike trimer rearrangements that prematurely convert spike into a post-fusion state, suppressing viral fusion, or destabilizing the trimer to cause its premature disassembly (Huo et al., 2020b, Walls et al., 2019, Liu et al., 2020a, Turonova et al., 2020, Benton et al., 2020, Cai et al., 2020, Koenig et al., 2021).”

– Page 23, line 1:‘Interestingly, among all the nanobody pairs that we tested, the synergy was greatest with the S2-10-dimer, which showed >4000 fold increase in potency when combined with either S1-23 or S1-RBD-15 (Figure 7I,J).' It is not clear if the data provided in figure 7I/J refers to monomers or dimers of S2-10 (the >4000 fold increase is not apparent from the figure).

The data provided refer to the dimer of S2-10 and we have modified Figure 7 to make this clearer. The data are indeed apparent from Figure 7I,J which show an extended asymmetric synergy profile spanning 4 orders of magnitude along the y-axis. Such an extended profile shows >4000 fold increase in potency for the S2-10-dimer, but not for S1-23 or S1-RBD-15, which had relatively modest 45- or 56-fold improvements in potency.

Reviewer #3 (Recommendations for the authors):Figure 2B: an S2 binder (S2-47) was included in the cluster dominated by RBD nanobodies. Any idea what is going on here?

The CDR3s in this particular cluster happen to be very short, at lengths of 6-7, appearing in the lower range of the length distribution of all 374 unique CDR3s (see following plot). A CDR3 pair with short length is more likely to feature a short Damerau-Levenshtein distance, and thus more likely to be categorized into the same cluster.

**Author response image 1. sa2fig1:** 

Figure 4: the authors benchmarked their neutralization assay using several previously reported nanobodies. However, their identities are missing. For example, did the authors evaluate epitope I neutralizers such as Nb20 and Nb21 reported by Xiang et al.? It would be useful to provide some insight into the consistency and variations among different assays.

We thank the Reviewer for the suggestion and the changes have been made as noted in response to Reviewer 1.

Regarding nanobody nebulization (Page 3 line 42): Indeed, Nambulli et al. reported the high preclinical efficacy of a nanobody construct (PiN-21) for inhalation therapy of SARS-CoV-2 infection in a sensitive COVID-19 animal model.

We thank the reviewer for bringing this to our attention and we now include the citation.

The synergy between different epitope binders (e.g., RBD and S2) is intriguing. It is unfortunate that the authors have not been able to follow up by characterizing their binding to the spike by high-resolution cryoEM, which may unravel the mechanism(s) of cooperativity to substantially strengthen the paper.

Although we have structurally mapped the binding epitope, we completely agree that this is a direction that we are urgently pursuing – however, given the ongoing nature of the pandemic and likely transformation of this virus into a major endemic problem for years to come, and the potential usefulness of all the nanobodies produced by our group and those of other groups, we feel it is of the utmost importance to publish these nanobodies, their sequences and characterization, as quickly as possible.

The authors state that the strong synergy is unique to nanobodies, but not to human IgG antibodies (Page 23 line 9), which seems to contradict some literature (see below). Moreover, since the 2D response assay has not been well adopted in antibody literature (for practical reasons, the molar ratio between two antibodies within a cocktail is fixed and stoichiometric), it would be difficult to draw a strong conclusion here.ter Meulen J, van den Brink EN, Poon LLM, Marissen WE, Leung CSW, Cox F, et al. (2006) Human Monoclonal Antibody Combination against SARS Coronavirus: Synergy and Coverage of Escape Mutants. PLoS Med 3(7): e237.

We were careful in our wording not to say that monoclonals cannot synergize: “our repeated observation of strong synergistic effects between nanobodies is especially noteworthy, reflecting the unique properties of nanobodies [such as their small size] that are not shared by, for example, human monoclonal antibodies”, i.e., we are stating that the property of being 10 times smaller than a monoclonal antibody greatly increases their likelihood of binding multiple epitopes in a synergistic fashion on any given target; but we have added the “such as their small size” to further emphasize this point.

We also note that the synergy we discovered against SARS-CoV-2 involves combinations of *monovalent* nanobodies binding to spike, whereas as typified by, Lui et al. 2020 IgG synergy typically occurs through aggregation of spike with poorly neutralizing bivalent antibodies. Their trivial example is certainly achievable by increasing the valency of nanobodies. Emphasizing the distinction, they show with Fab fragments of their antibodies, that the *neutralization and synergy* entirely depend on these reagents being delivered as bivalent IgGs. As is well recognized in the field, the mAbs in the Lui et al. paper cannot bind in a 1:1 molar ratio on a spike trimer due to steric clash. In striking contrast, we show that at least three nanobodies can bind to the same RBD domain alone, which could result in nine nanobodies binding a single spike timer! This *intra-spike synergy* of independent neutralizing mechanisms is a broad conceptual advance unique to our manuscript. Indeed, many of our nanobody combinations are synergistic, which is a far cry from the *very limited* examples of synergy among mAbs. We now also include a reference to two SARS-CoV-2 mabs that have demonstrated synergy:

“For example, tixagevimab and cilgavimab, two human monoclonal antibodies that target non-overlapping regions of the RBD, function synergistically and show promise as prophylactic and therapeutic agents against COVID-19 (Dong et al., 2020, Zost et al., 2020).”

Page 23 lines 23-24 (perspective and discussion): it would be useful if the authors could provide some thoughts to suggest potential combinatory/ synergistic examples as lead candidates for future preclinical development.

We completely agree that this very much represents urgent follow-up work, but we feel it is premature to speculate about combinatorial examples as specific kinds of therapeutics at this time without also being able to present accompanying pertinent data.

Page 26 line 10: complementary digestion with trypsin and chymotrypsin (Xiang et al. 2020). It seems that the paper was incorrectly cited (Xiang, Y., et al. (2021). Integrative proteomics identifies thousands of distinct, multi-epitope, and high-affinity nanobodies. Cell Syst 12, 220-234 e229.)

Many thanks, we have updated the reference.